# Vascular endothelium deploys caveolin-1 to regulate oligodendrogenesis after chronic cerebral ischemia in mice

Ying Zhao [1,5], Wusheng Zhu[1,5], Ting Wan [2,5], Xiaohao Zhang [3,5], Yunzi Li[1], Zhenqian Huang [1], Pengfei Xu[4], Kangmo Huang[1], Ruidong Ye [1,6] ✉, Yi Xie [1,6] ✉ & Xinfeng Liu[1,4,6] ✉

Oligovascular coupling contributes to white matter vascular homeostasis. However, little is known about the effects of oligovascular interaction on oligodendrocyte precursor cell (OPC) changes in chronic cerebral ischemia. Here, using a mouse of bilateral carotid artery stenosis, we show a gradual accumulation of OPCs on vasculature with impaired oligodendrogenesis. Mechanistically, chronic ischemia induces a substantial loss of endothelial caveolin-1 (Cav-1), leading to vascular secretion of heat shock protein 90α (HSP90α). Endothelial-specific over-expression of Cav-1 or genetic knockdown of vascular HSP90α restores normal vascular-OPC interaction, promotes oligodendrogenesis and attenuates ischemic myelin damage. miR-3074(−1)−3p is identified as a direct inducer of Cav-1 reduction in mice and humans. Endothelial uptake of nanoparticle-antagomir improves myelin damage and cognitive deficits dependent on Cav-1. In summary, our findings demonstrate that vascular abnormality may compromise oligodendrogenesis and myelin regeneration through endothelial Cav-1, which may provide an intercellular mechanism in ischemic demyelination.

With lower blood flow from distal parts of long deep arteries, cerebral white matter is susceptible to ischemia[1,2]. In the hypoxia/ischemia, damages, such as oxidative stress, inflammation, and excitotoxicity, to oligodendrocyte precursor cells (OPCs) can trigger rapid and profound demyelination[2]. Adult resident OPCs constitute approximately 6% of the total cell number in the CNS[3], providing a source for myelin sheath renewal[4,5]. However, OPC maturation is often arrested in hypoxia/ischemia[6]. Our previous study, consistently, found robust OPC proliferation after chronic cerebral ischemia, but it was not durable and failed to restore myelination[7]. Thus, attempts to promote oligodendrogenesis may provide clues to ameliorate demyelination therapeutically.

The coordination of oligovascular coupling is important for white matter maintenance[8]. Oligodendroglial cells can regulate CNS endothelial cell proliferation and angiogenesis[9,10]. Vascular endothelial cells, in turn, are documented to secrete factors to favor OPC survival and promote oligovascular remodeling[11–13]. In cerebral ischemic injury, dysfunctional OPC-vascular interaction was detected as OPC might mediate early endothelial barrier opening[14]. To the opposite, OPC in both human and animal models of hypoxic-ischemic encephalopathy was reported to promote white matter vascularization[15]. However, whether this aberrant interaction would induce OPC changes is not fully understood. As endothelial dysfunction was detected even before demyelination occurred[14,16], it then prompts us to consider the

[1]Department of Neurology, Affiliated Jinling Hospital, Medical School of Nanjing University, Nanjing, Jiangsu 210000, China. [2]Department of Neurology, Xijing Hospital, Air Force Medical University, Xi'an, Shanxi 710032, China. [3]Department of Neurology, Nanjing First Hospital, Nanjing Medical University, Nanjing, Jiangsu 210000, China. [4]Stroke Center & Department of Neurology, The Affiliated Hospital of USTC, Division of Life Sciences and Medicine, University of Science and Technology of China, Hefei 230036 Anhui, China. [5]These authors contributed equally: Ying Zhao, Wusheng Zhu, Ting Wan, Xiaohao Zhang. [6]These authors jointly supervised this work: Ruidong Ye, Yi Xie, Xinfeng Liu. ✉e-mail: yeruid@gmail.com; xy_307@126.com; xfliu2@vip.163.com

contributing effects of vascular damage to the pathogenesis of ischemic demyelination based on OPC-endothelial association.

Caveolin-1 (Cav-1), a component of caveolae, is highly expressed on endothelial cells, orchestrating signal transduction and vesicular trafficking[17,18]. Decreased level of endothelial Cav-1 was implicated in blood-brain barrier (BBB) breakdown and neuroinflammation after ischemic stroke[19,20]. Clinical data revealed that, in patients with ischemic stroke, a reduced level of serum Cav-1 was associated with cerebral microbleeds[21] and symptomatic bleeding[22], suggesting an important role of Cav-1 in vessel stability. A recent paper has indicated that arteriolar Cav-1 has an active role in mediating neurovascular coupling, including neural activity and vascular dynamics[23]. As such, we aim to investigate whether endothelial Cav-1 could participate in oligovascular coupling in chronic cerebral ischemia.

In this study, we investigated the potential role of endothelial Cav-1 linking vasculature with ischemic demyelination in the mouse bilateral carotid artery stenosis (BCAS) model and leukoaraiosis patients. Endothelial Cav-1 was remarkably reduced, which was responsible for the aberrant relationship between the OPCs and the vasculature, following hypoxic-ischemic injury. We identified both an upstream cause of endothelial Cav-1 reduction and a resultant secreted downstream effector affecting OPC differentiation. The treatment to stabilize oligovascular interactions restored myelination in chronic cerebral ischemia, suggesting a new therapeutic strategy.

## Results

### Endothelial dysfunction precedes demyelination in the corpus callosum (CC) during chronic hypoperfusion

The time-course changes in cerebral blood flow (CBF) and myelin damage were analyzed in the CC at 1, 2 and 4 weeks after BCAS surgery. At 1-week post BCAS, CBF significantly declined to nearly 49% of the control group (Fig. S1a, b, $P < 0.001$). Despite a slow recovery thereafter, the CBF remained at a level reduced by 23% 4 weeks after BCAS (Fig. S1a, b, $P = 0.0002$). Luxol fast blue (LFB) and black-gold II staining both confirmed myelin loss since BCAS_2w (Fig. S1c–e). Consistently, the fluorescent intensity of myelin-related proteins, myelin-associated glycoprotein (MAG), and myelin basic protein (MBP), was not reduced until 2 weeks after the insult (Fig. S1f, g).

We next probed whether endothelial dysfunction was an initial change before demyelination under chronic ischemia. By Transmission Electron Microscopy (TEM), as early as 1 week after BCAS, the morphology of tight junctions (TJs) was significantly disrupted, characterized by an increased gap area between endothelial cells (Fig. S2a, b). To assess vascular permeability, the 3-kDa dextran was utilized as a fluorescent tracer. In the CC of BCAS mice, the extravasated dyes were observed around the CD31+ vessels at 1 and 2 weeks after BCAS, which was minimized at 4 weeks post-BCAS (Fig. S2c, d). The vessel regression, quantified by the number of collagen IV+ basement membranes without CD31+ endothelial cells (empty sleeves), increased after BCAS with a peak at 1 week (Fig. S2e, f). Moreover, the number of vessels with degraded collagen IV gradually increased from 1 week to 4 weeks post-BCAS (Fig. S2e, g), suggesting vessel instability during hypoperfusion.

### OPCs are increasingly accumulated as clusters on vasculature with impaired differentiation after BCAS

We then asked whether early endothelial dysfunction was associated with OPC changes in BCAS mice. By immunostaining of PDGFRα, oligodendrocyte transcription factor 2 (Olig2), and CD31, we found accumulative OPCs interacted with blood vessels 2 weeks after BCAS. Both frequency and size of clusters were increased on the vasculature (Fig. 1a–c; compared to controls, both $P < 0.001$). This association was more frequent at 4 weeks post-BCAS, suggesting a gradual strengthened association of OPCs with blood vessels (Fig. 1a–c). Immunoelectron microscopy also showed the process of Olig2 immunogold+ oligodendroglial cells contacting the vessels (Fig. 1d). Breast

carcinoma amplified sequence 1 (BCAS1)[24] and ectonucleotide pyrophosphatase/phosphodiesterase 6 (ENPP6)[25] have been recognized as markers of newly-formed immature oligodendrocytes, representing a status of ongoing myelination and remyelination. Interestingly, these markers were greatly increased at 2 weeks followed by a reduction at 4 weeks of chronic hypoperfusion (Fig. 1e–g; compared with BCAS_2w, $P = 0.0017$ and 0.0062, respectively). Mature oligodendrocytes, which were stained for CC1 and Olig2, were gradually decreased at both 2 and 4 weeks after BCAS (Fig. 1h, i). These data suggest that OPCs tended to differentiate to generate new oligodendrocytes. However, this compensation, possibly with defective maturation and increased oligodendrocyte death, were not capable of supporting remyelination at 4 weeks of hypoperfusion.

### Dysfunctional endothelial cells in hypoxia compromise OPC maturation

In vitro, we utilized CoCl$_2$ (10 μM, 5 days) and oxygen-glucose deprivation (OGD, 4 h) to induce hypoxia. We performed PCR for the expression of classic hypoxia-related genes in endothelial cells. CoCl$_2$ and OGD treatment both increased the mRNA level of hypoxic markers, including HIF-1α, HIF-2α, PDK-1, BNIP3, and VEGF, representing significant hypoxia (Fig. 2a). After exposure to CoCl$_2$ or OGD, the proliferation of brain microvascular endothelial cells (BMECs) was increased with enhanced apoptosis by the quantification of EdU-positive and TUNEL-positive cells (Fig. 2b, c). ZO-1 and Claudin-5 in the immunostaining were both fractured at the boundary of CoCl$_2$- and OGD-treated BMECs compared to controls (Fig. 2d). A consistent decrease was observed in the protein levels of TJs (Fig. 2e, f). To address the effects of dysfunctional BMECs on normal OPCs, we conducted conditioned media (CM) transfer assay. CM from either CoCl$_2$- or OGD-treated BMECs was collected and then incubated with OPCs. When cultured in the CoCl$_2$-CM or OGD-CM, the proliferation of OPCs was preferred, as Ki67+PDGFRα+Olig2+ cells were significantly increased compared to those in control CM (Fig. 2g, h; $P < 0.001$ and $P = 0.004$, respectively). The generation of MBP+ oligodendrocytes with highly developed ramified processes was inhibited by CoCl$_2$-CM and OGD-CM (Fig. 2i–k; both $P < 0.001$). Immunoblotting displayed similar changes in that the expression of PDGFRα was increased while mature oligodendrocytes markers, MBP and 2′, 3′-cyclic nucleotide 3′-phosphodiesterase (CNPase), were reduced to less than half of control protein levels (Fig. 2l, m). These observations verified that BMECs, who suffered from hypoxia, could release detrimental factors to suppress OPC maturation.

### Hypoxic-ischemic injury induces endothelial Cav-1 reduction and HSP90α secretion

Given that Cav-1 plays a vital role in endothelial function, we explored the expression of Cav-1 in BMECs exposed to CoCl$_2$ or OGD. Immunostaining for Cav-1 displayed that the intensity of Cav-1, which normalized to CD31 intensity, was decreased compared to BMECs under normoxic status (Fig. 3a, b; both $P < 0.001$). Besides, immunoblotting revealed that CoCl$_2$ and OGD exposure reduced Cav-1 protein to 43.2% and 27.8% of control cells respectively (Fig. 3c, d; $P = 0.0288$ and 0.0068, respectively). To probe the extracellular engagement, we utilized an antibody array covering more than 1300 antibodies related to several signaling pathways to measure the changes in soluble factors secreted by BMECs. Compared to control CM, secreted HSP90α was the most significantly expressed functional protein in CoCl$_2$-CM (Fig. 3e), which was further confirmed by ELISA (Fig. 3f). In vivo, reduced endothelial Cav-1, accompanied by elevated extravascular HSP90α, was detected in the CC after BCAS (Fig. 3g–i). Immunoblotting of extracted microvascular segments of the CC consistently demonstrated that the vascular Cav-1 was reduced while HSP90α was increased (Fig. 3j–m). As for the correlation between Cav-1 and HSP90α, we found the Cav-1 antibody specifically coprecipitated

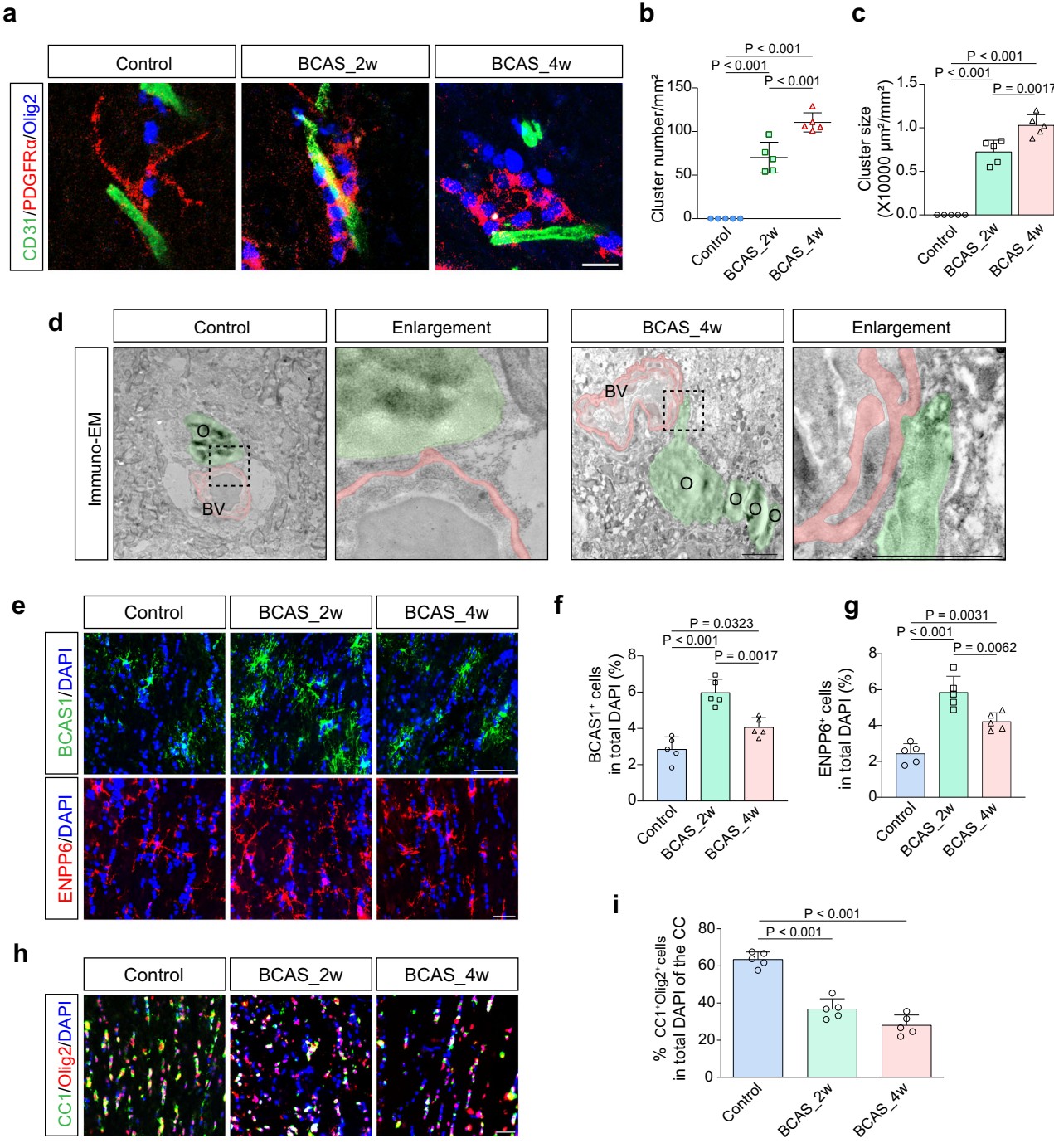

**Fig. 1 | Perivascular clusters of OPCs with impaired differentiation in cerebral hypoperfusion. a–c** Representative confocal images and quantitative analyses of cluster number and size of PDGFRα⁺Olig2⁺ cells on CD31⁺ blood vessels (*n* = 5 mice/group). Scale bar, 10 μm. **d** Ultrastructural images displaying the interactions between Olig2 immunogold labeled cells (pseudocolor-green) and the vessel (pseudocolor-red). Magnified regions showing the oligodendroglial cell process attached to the vessel in the right panel (*n* = 5 mice/group). Scale bar, 2 μm.

**e–g** Representative immunofluorescence images and quantifications showing the changes in the percentage of BCAS1⁺ or ENPP6⁺ newly formed immature oligodendrocytes (*n* = 5 mice/group). **h, i** Representative images and quantifications showing the changes in CC1⁺Olig2⁺ mature oligodendrocytes (*n* = 5 mice/group). Scale bar, 10 μm. All data are presented as the mean ± SD. The data were compared by one-way ANOVA with Tukey post hoc test. BV blood vessel, O oligodendroglial cells. Source data are provided as a Source Data file.

HSP90α in the co-immunoprecipitation assay (Fig. S3a). HSP90α was also be detected in a reverse co-immunoprecipitation analysis (Fig. S3b). Therefore, after hypoxic treatment, reduced Cav-1 could also be co-localized with HSP90α in BMECs. The Cav-1/HSP90α co-immunostaining further revealed that, under normoxic status, Cav-1 was co-localized with about 45.6% of total HSP90α at the endothelial membrane (Fig. S3c). However, during chronic hypoxia, the co-localization was reduced to 5.5% (Fig. S3c, d; *P* = 0.0028).

Of particular interest, we evaluated Cav-1 expression among different neural cells, including endothelial cells, pericytes, oligodendroglial lineage cells, and astrocytes. The results showed that about 84% CD31⁺ endothelial cells expressed Cav-1, whose level was at least 2.5 times higher than other neural cells (Fig. S4). It thus indicated that the function of Cav-1 may be endothelial cell-autonomous. To determine whether Cav-1 reduction would affect the caveolae level after BCAS, we used TEM to directly assess caveolar change. We found

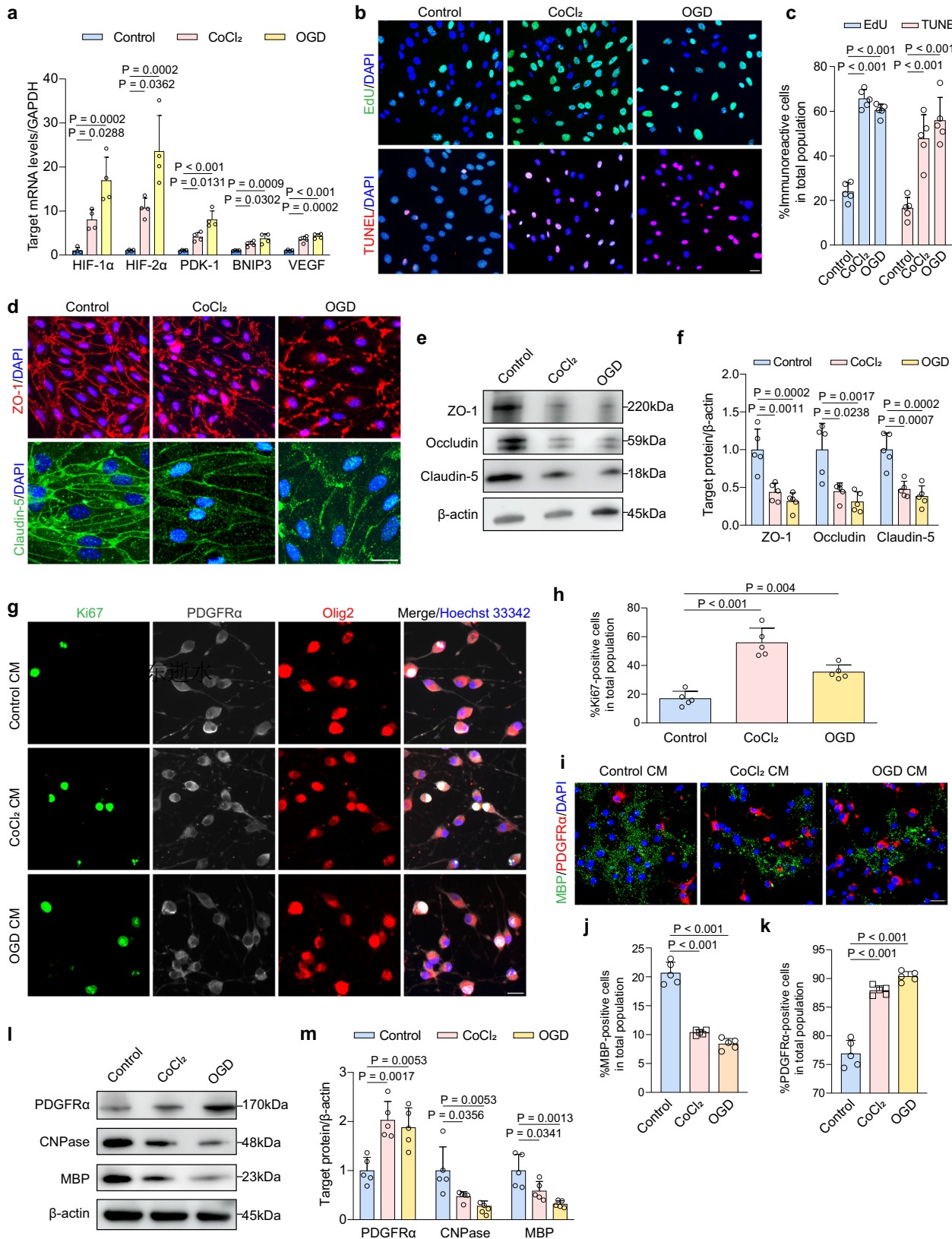

capillary and arteriolar caveolae number did not change between control and BCAS mice (Fig. S5a–c). Co-immunostaining of Cavin-1, a caveolae marker, with vascular markers, including MFSD2A for capillary and α-SMA for arteriole, consistently showed no differences in vascular Cavin-1 expression between control and BCAS mice (Fig. S5d–f). In in vitro experiments, the immunostaining and the

quantification demonstrated that chronic hypoxia did not affect the co-localization of Cav-1 and Cavin-1 in the membrane (Fig. S5g, h), further suggesting that hypoxia might not affect caveolar Cav-1.

To identify the origin of HSP90α secretion, we performed combined fluorescent in situ hybridization (FISH) and immunofluorescence. We used the double-FAM-labeled probes to detect

**Fig. 2 | Dysfunctional BMECs in hypoxia block OPC differentiation. a** The mRNA level of classic hypoxia-related genes of control, CoCl$_2$- and OGD-treated BMECs (*n* = 4 independent primary cell cultures/group). **b**, **c** Representative images and quantifications of endothelial proliferation (EdU$^+$) and apoptosis (TUNEL$^+$) (*n* = 5 independent primary cell cultures/group). Scale bar, 20 μm. **d** Immunofluorescent staining of ZO-1 or Claudin-5 in BMECs (*n* = 5 independent primary cell cultures/group). Scale bar, 20 μm. **e**, **f** Immunoblotting showing ZO-1, Claudin-5, Occludin expression of BMECs with or without ischemia. Results represent the mean for the relative band intensity of five replicates. **g**, **h** Proliferating OPCs (Ki67$^+$PDGFRα$^+$Olig2$^+$) grown in control CM and CoCl$_2$−, OGD-treated CM and its quantification (*n* = 5 independent primary cell cultures/group). Scale bar, 20 μm. **i–k** Representative immunofluorescent images and quantifications showing the percentage of OPC differentiation (*n* = 5 independent primary cell cultures/group). Scale bar, 20 μm. **l**, **m** Immunoblotting analyses of PDGFRα, MBP, CNPase expression in OPCs of 3 groups. Results represent the mean for the relative band intensity of five replicates. For immunoblotting experiments, protein samples derived from the same experiment and gels/blots were processed in parallel. All data are presented as the mean ± SD. The data were compared by one-way ANOVA with Tukey post hoc test. Source data are provided as a Source Data file.

*Hsp90α* mRNA and different cell marker antibodies to locate different cells, including CD31 for endothelial cells, PDGFRβ for pericytes, GFAP for astrocytes, and Olig2 for oligodendroglial lineage cells. The results showed that *Hsp90α* mRNA was mainly expressed in endothelial cells and significantly increased under chronic ischemic conditions (Fig. S6). Thus, the secreted HSP90α close to or associated with the vessel might be mostly derived from vascular endothelial cells.

## Specific expression of endothelial Cav-1 attenuates ischemic myelin damage

We employed a genetic approach to increase endothelial Cav-1 using an Adeno-associated virus (AAV) carrying the *Cav-1* gene under the promoter of *Tie1*. The viral vectors were stereotaxically injected into the CC 3 weeks before BCAS surgery (Fig. 4a). The GFP signal was found within endothelial cells in the CC. AAV-*Tie1-Cav-1* successfully elevated endothelial Cav-1 expression (Fig. 4b, c; *t* = 18.41, *P* < 0.001). This 1.98-fold upregulation of endothelial Cav-1 led to 46.7% downregulation of endothelial HSP90α and 48.9% downregulation of secreted HSP90α in the perivascular region (Fig. 4d–f; *P* = 0.0111 and 0.0008, respectively), indicating HSP90α might be a downstream secretory factor of Cav-1. Besides, in the CC receiving AAV-*Tie1-Cav-1*, OPC clustering around the microvessels was significantly attenuated with decreased cluster numbers and occupied area (Fig. 4g–i). BCAS1$^+$ cells as well as CC1$^+$Olig2$^+$ mature oligodendrocytes were remarkably increased in AAV-*Tie1-Cav-1*-transfected mice (Fig. 4j–l). As expected, chronic ischemia-induced myelin proteins reduction was rescued in the *Cav-1* overexpression group (Fig. 4m, n), suggesting that endothelial Cav-1 may be an important intermediary in ischemic demyelination.

## Knockdown of HSP90α ameliorates myelin abnormality without changing the Cav-1 level

To examine the role of HSP90α in myelin damage and its relation with Cav-1, we intravenously injected negative control (N.C.) or HSP90α siRNA packed with polyetherimide (PEI) into wild-type and *Cav-1*$^{-/-}$ mice after BCAS surgery at intervals of three days for 4 weeks (Fig. 5a). The capture of Cy5-labeled N.C. or HSP90α siRNA by microvessels was identified in the CC (Fig. 5b). HSP90α siRNA-induced inhibition on secretory HSP90α was significantly higher than the suppression of endothelial HSP90α in BCAS mice of two genotypes (Fig. 5c). As expected, the interference of HSP90α did not affect the endothelial Cav-1 level (Fig. S7). Compared to mice injected with N.C. siRNA, both wild-type and *Cav-1*$^{-/-}$ mice that received HSP90α siRNA exhibited less frequent OPC clusters attached to the vessels with decreased size (Fig. 5d–f). More BCAS1- and ENPP6-positive pre-myelinating oligodendrocytes were newly formed in HSP90α siRNA-treated BCAS mice (Fig. 5g–i), together with more mature oligodendrocytes (Fig. 5j, k).

## miR-3074-1-3p directly regulates endothelial Cav-1 expression

To explore the upstream mediator of Cav-1, high-throughput sequencing of miRNAs was carried out using the CC tissue of control and BCAS_4w mice. Among the miRNA sequences obtained, we filtered 30 significantly up-regulated miRNAs with fold change >= 2 (Fig. 6a). After cross-compared to the predicted miRNAs in the TargetScan, 13

potential miRNAs were screened out (Fig. 6a, b). We found that miR-3074-1-3p was one of the most significantly upregulated genes at BCAS_4w (Fig. 6c). Validated by real-time PCR, miR-3074-1-3p reached 10.2-fold higher expression in mice subjected to BCAS (Fig. 6d). Most importantly, to define the cell-autonomous nature of miR-3074-1-3p during chronic ischemia, real-time PCR was employed in the BMECs under chronic hypoxia and miR-3074-1-3p had the highest increase (Fig. 6e). miRNA fluorescent in situ hybridization further displayed the elevation of miR-3074-1-3p in CoCl$_2$-treated BMECs (Fig. 6f, g).

To determine whether miR-3074-(1)-3p could specifically recognize the 3′ untranslated regions (UTR) of *Cav-1*, we conducted the luciferase reporter assay in 293 T cells. Mouse miR-3074-1-3p and human homolog miR-3074-3p share conserved predicted binding sites in the 3′ UTR of mouse and human *Cav-1* (Fig. 6h). miR-3074-(1)-3p mimics, instead of N.C., inhibited the luciferase activities of wild-type *Cav-1* 3′ UTR. However, when introduced to the mutated *Cav-1* 3′ UTR, the mimics could not induce any interference in the luciferase activities (Fig. 6i), implying that Cav-1 may be a direct target of miR-3074-(1)-3p in both mice and human. To examine whether miR-3074-1-3p could regulate Cav-1 at the protein level in BMECs, we analyzed endothelial Cav-1 expression under the transfection with miR-3074-1-3p agomir and antagomir. As depicted in Fig. 6j, k, miR-3074-1-3p agomir substantially suppressed the level of Cav-1 while antagomir enhanced the Cav-1 expression in BMECs.

## Serum levels of homolog miR-3074-3p and Cav-1 in clinical leukoaraiosis

We included a total of 154 patients (54.5% male; mean age, 67.0 ± 9.5 years) in the present study. Eighty-eight subjects (57.1%) had imaging manifestation with leukoaraiosis. The comparison of baseline data was demonstrated in table S1 based on patients with or without leukoaraiosis. Patients with leukoaraiosis were more likely to have a higher level of hsa-miR-3074-3p and a lower level of Cav-1 (Table S1, both *P* = 0.001). After adjusting for potential confounders, increased hsa-miR-3074-3p [odds ratio (OR), 1.41; 95% confidence interval (CI), 1.17–1.69; *P* = 0.001] and reduced Cav-1 level (OR, 0.29; 95% CI, 0.16–0.53; *P* = 0.001) were also significantly associated with a higher risk of leukoaraiosis (Fig. 7a). Furthermore, linear regression analysis indicated that hsa-miR-3074-3p level was negatively associated with Cav-1 (Fig. 7b, *β* = −0.261, *P* = 0.027). Among the 88 patients with leukoaraiosis, 37 patients (42.0%) were diagnosed with severe leukoaraiosis. Compared to patients with mild leukoaraiosis, patients with severe leukoaraiosis tended to have a higher level of hsa-miR-3074-3p (Fig. 7c, *P* = 0.1309) but a lower level of Cav-1 (Fig. 7d, *P* = 0.0159).

## Antagomir ameliorates endothelial abnormality and reverses OPC changes via Cav-1

To ask whether the regulation of miR-3074-1-3p may exert effects on BMECs and OPCs and whether Cav-1 was involved, we treated wild-type and *Cav-1*$^{-/-}$ BMECs with miR-3074-1-3p antagomir respectively. Eight hours before exposure to hypoxia, BMECs were transfected with N.C. or antagomir. The Cy5-labeled antagomir was localized around the nucleus of BMECs, verifying the successful uptake of antagomir (Fig. 8a). By both immunostaining and immunoblotting, Cav-1 was

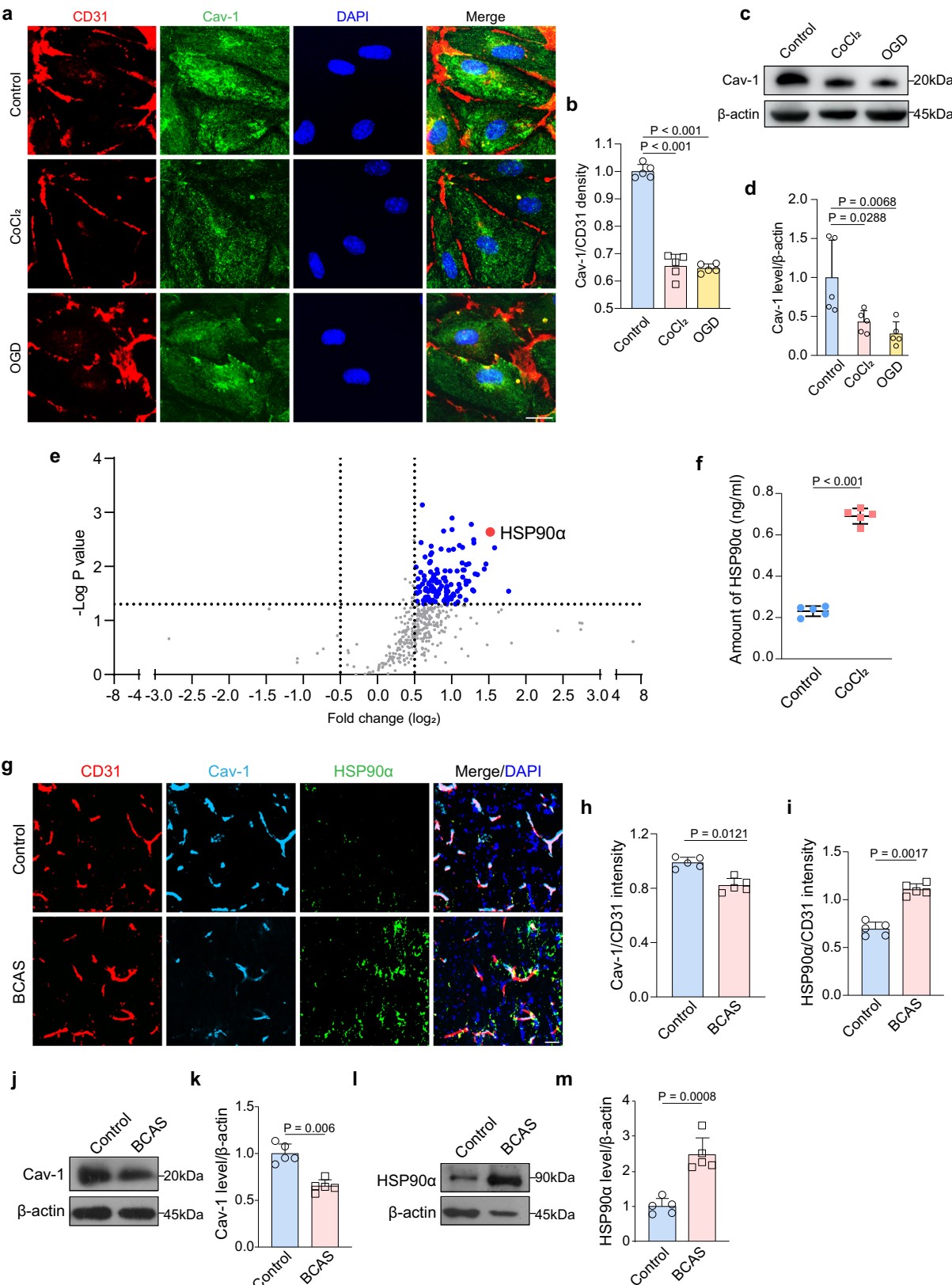

remarkably increased in wild-type BMECs transfected with antagomir (Fig. 8a–c). Antagomir alleviated the loss of endothelial TJs almost to the basal conditions in CoCl₂-treated wild-type BMECs (Fig. 8b, c). This restoration was counteracted by the ablation of Cav-1 (Fig. 8b, c). In wild-type BMECs, co-immunostaining revealed that antagomir could increase the percent of co-localized HSP90α with Cav-1 in the hypoxic

environment (Fig. 8d, e). Correspondingly, the secreted HSP90α was strikingly decreased in the culture medium (Fig. 8f), which thereby rescued the defective maturation of OPC (Fig. 8g–j). On the contrary, under the deletion of Cav-1, antagomir could not suppress HSP90α secretion in response to CoCl₂, which subsequently limited OPC differentiation, as illustrated by immunostaining and immunoblotting for

**Fig. 3 | Reduced endothelial Cav-1 facilitates vascular HSP90α secretion. a–d** Immunofluorescent images, immunoblotting and quantifications showing the changes of Cav-1 in BMECs of control and CoCl₂ group. Results represent the mean for the relative band intensity of five replicates. Scale bar, 20 μm. (**e**) Volcano plot showing highly expressed secretory factors by endothelial cells treated with chronic hypoxia. HSP90α (red spot) is shown as a highly expressed secretory protein compared to control. Spots of $p$ value <0.05 with $\log_2$ [fold change] >0.5 or < −0.5 were considered as potential proteins to be chosen and represented in blue. Grey spots showing no differences between two groups ($n = 4$ independent primary cell cultures/group). **f** ELISA displaying the amount of HSP90α in control

CM and CoCl₂ CM ($n = 5$ independent primary cell cultures/group). **g–i** Immunostaining of Cav-1, HSP90α, and CD31 and quantifications showing the decreased Cav-1 expression and elevated HSP90α secretion in BCAS mice ($n = 5$ mice/group). Scale bar, 20 μm. **j–m** Immunoblotting and quantifications of Cav-1 level in microvascular segments and HSP90α level in the CC brain tissue. Results represent the mean for the relative band intensity of five replicates. For immunoblotting experiments, protein samples derived from the same experiment and gels/blots were processed in parallel. All data are presented as the mean ± SD. The data in **b**, **d** were compared by one-way ANOVA with Tukey post hoc test. The data in others were compared by paired $t$-test. Source data are provided as a Source Data file.

---

myelin-related protein levels (Fig. 8f–j). Thus, our data suggested endothelial miR-3074-1-3p inhibition could rescue the hypoxia-induced endothelial dysfunction and subsequent defective OPC differentiation via Cav-1.

### Cav-1 is required in PEI-antagomir-induced attenuation of OPC perivascular clustering

We next examined whether antagomir could exert protection in vivo. The time-course expression of miR-3074-1-3p showed that it started to increase at 2 weeks after BCAS (Fig. 9a). Thus, continuous delivery of in vivo-*jetPEI*™-formulated Cy5-antagomir was started at BCAS_2w and lasted for another 2 weeks through the osmotic pumps into the CC (Fig. 9b). By live imaging and immunostaining, antagomir was located within the microvessels (Fig. 9c, d). It increased endothelial Cav-1 level and inhibited HSP90α secretion (Fig. 9d–g; Fig. S8a, b). The TJs level and structure in wild-type BCAS mice were significantly recovered by the antagomir (Fig. 9g; Fig. S8c–f). Nonetheless, antagomir could not affect the HSP90α increment and TJs loss in *Cav-1*⁻/⁻ BCAS mice (Fig. 9e–g; Fig. S8). After treatment with antagomir, a lower frequency of OPC cluster was detected in wild-type mice of BCAS. Conversely, regardless of receiving N.C. or antagomir, the Cav-1 null mice manifested a parallel number and size of perivascular OPC clusters after BCAS (Fig. 10a–f). In addition, newly-formed oligodendrocytes were found in antagomir-treated wild-type CC, while equally treated *Cav-1*⁻/⁻ BCAS mice displayed defective remyelination as fewer BCAS1⁺ and ENPP6⁺ cells were present in the CC (Fig. 10g–i).

### Cav-1 is required in PEI-antagomir-induced attenuation of demyelination and cognitive deficits

By the analysis of immunostaining, TEM, and immunoblotting, the antagomir significantly restored the white matter integrity in wild-type BCAS mice, while it was unable to induce detective attenuation without Cav-1 (Fig. S9). Cognitive function was evaluated by Morris water maze (MWM) and novel object recognition (NOR). From the 3rd day to the 5th day of the spatial trial in MWM, wild-type BCAS mice subjected to antagomir spent less time achieving platform (Fig. S10a, $P = 0.0161$, $P = 0.0002$ and $P < 0.001$ on each day) and swam shorter distances than the N.C.-treated group (Fig. S10b, c, $P = 0.0276$, P = 0.0184 and $P = 0.014$ on each day). In contrast, antagomir-administered *Cav-1*⁻/⁻ BCAS mice did not acquire promotion in either escape latency or path length (Fig. S10a–c, both $P < 0.01$, on each day). For probe trial without the platform, antagomir treatment ameliorated chronic ischemia-induced impairment in the crossovers of platform location as well as the time spent in the target quadrant in wild-type mice (Fig. S10c–e), which was not seen within antagomir-administered *Cav-1*⁻/⁻ BCAS mice. In the NOR with 1-hour interval, wild-type BCAS mice that received antagomir took a longer time to explore new objects (Fig. S10f, compared to the N.C. group of wild-type mice, $P = 0.0005$). Another NOR task with 24-hour interval consistently showed increased interactions with the novel object in the antagomir-administered wild-type BCAS mice (Fig. S10g, compared to the N.C. group of wild-type mice, $P = 0.0028$). Animals in other groups could not distinguish the new object from the old one. Therefore, Cav-1 was necessary for the

protection of white matter and cognitive function induced by miR-3074-1-3p antagomir.

## Discussion

Endothelial Cav-1 has been well-characterized in maintaining BBB homeostasis. Here, we extended insights into ischemic demyelination and identified a role of Cav-1 in regulating oligovascular coupling. Chronic hypoxia and ischemia led to endothelial dysfunction and diminished Cav-1 expression, triggering HSP90α secretion from endothelial cells. Extracellular HSP90α exerted remarkable paracrine function by inhibiting oligodendrogenesis and myelin regeneration. Endothelial miR-3074-1-3p was screened out and acted as an upstream regulator of Cav-1. In leukoaraiosis patients, human serum miR-3074-3p, which shared a conserved seed region with mouse miR-3074-1-3p, was increased while serum Cav-1 was decreased. This reciprocal change was in accord with the results from the animal study, indicating the translation of laboratory findings to human biology. Inhibition of miR-3074-1-3p by nanoparticle-antagomir, dependent on Cav-1, could ameliorate OPC clustering and promote oligodendrogenesis, which was ultimately beneficial for cognitive performance. However, as the hippocampus is important for both spatial memory and recognition memory[26], the results might be confounded by the chronic ischemia-induced changes in the hippocampal region. Therefore, the conclusion should be treated with caution.

Structural and functional changes in the vasculature may promote hypoperfusion[27]. The damage and chronic remodeling of microvessels, such as narrowing of the arteriolar lumen and thickening of the vessel wall[28], can impair CBF regulation and cause ischemia in distal territories[29]. We found significant loss of endothelial junction proteins and early vascular hyper-permeability in the CC of BCAS mice, which coincided with the findings in patients and animal models with chronic cerebral ischemia[14,16,30]. Despite angiogenesis reported in response to chronic hypoxia[31], we found vessel regression and BMECs apoptosis under hypoxic injury, reflecting endothelial instability in long-term ischemic injury. Vascular dysfunction has been identified as an early prelude to white matter abnormalities in various pathological settings. Clinical studies found increasingly destroyed BBB in normal-appearing white matter at the proximity of white matter lesions[29,32], a predilection site for the expansion of hyperintensity[33,34]. A recent animal study confirmed that endothelial dysfunction was an upstream pathogenic factor in oligodendroglial pathologies[35]. We also discovered that endothelial impairments occurred before ischemic demyelination, indicating that myelin loss may be secondary to vascular dysfunction in ischemic settings.

Disrupted OPC-vascular interaction could promote early BBB opening in chronic cerebral ischemia[14] and multiple sclerosis[36]. Conversely, increased OPC could interact with endothelium and promote white matter vascularization[10,15]. In contrast to these studies, we focused on the effects of altered vascular-OPC association on oligodendroglial injury under the context of chronic cerebral ischemia. We found increased OPCs accumulating on the vasculature at 2 weeks post-BCAS in the CC. Interestingly, newly-formed oligodendrocytes of BCAS1⁺ and ENPP6⁺ cells significantly increased at the same time. We

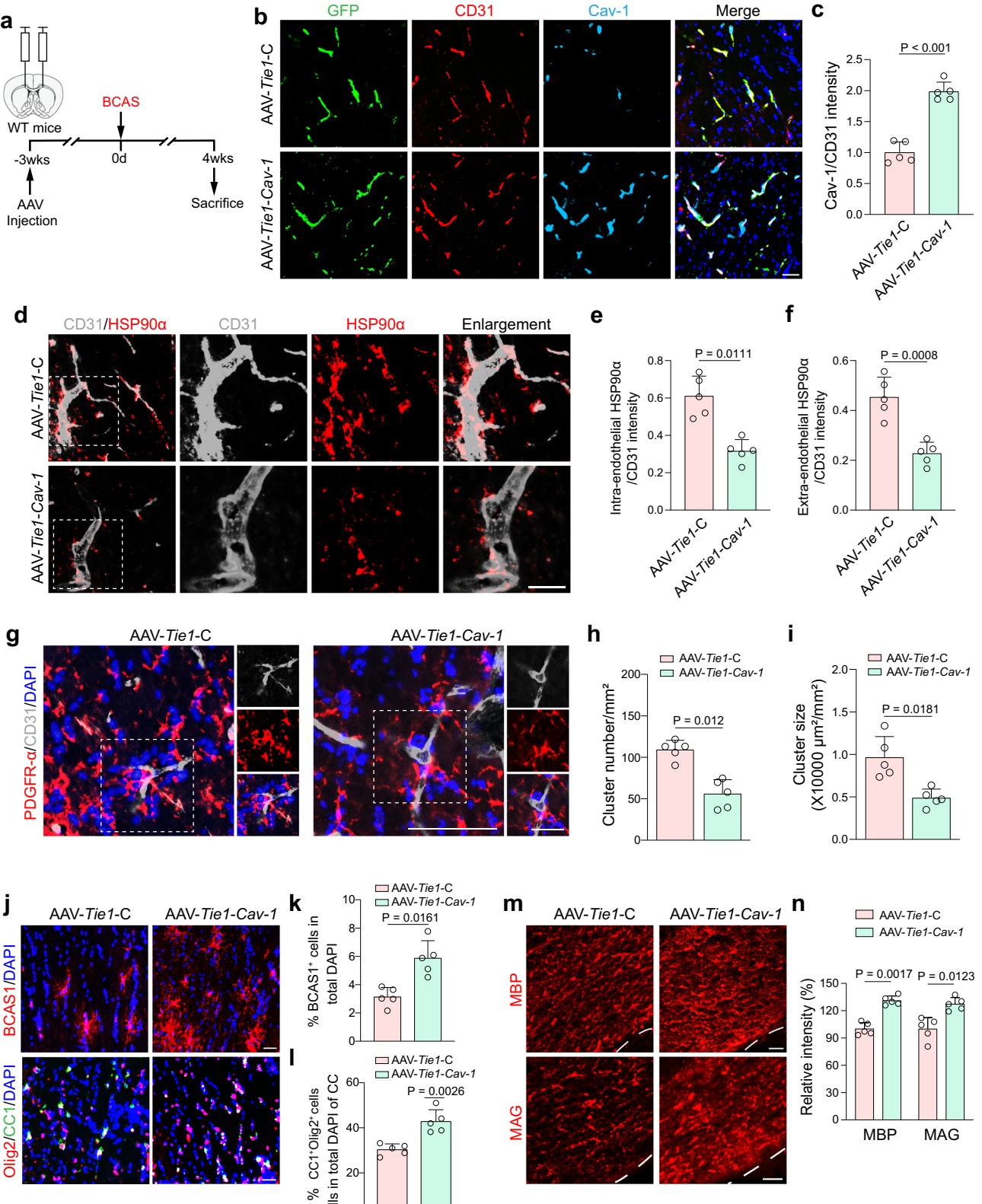

**Fig. 4 | Over-expression of endothelial Cav-1 inhibits HSP90α secretion and attenuates ischemic demyelination. a** Experimental flow chart.
**b, c** Representative fluorescent images and qualification of the transfection of GFP (green) reporter AAV into microvessels (CD31⁺, red) with enforced Cav-1 (light blue) expression in the CC of BCAS mice (*n* = 5 mice/group). Scale bar, 20 μm. **d–f** Double immunostaining and qualifications of CD31 and HSP90α showing the reduction of HSP90α induced by AAV-*Tie1-Cav-1* (*n* = 5 mice/group). Scale bar, 20 μm.
**g** Representative images showing the PDGFRα⁺ clusters contacting vessels (*n* = 5

mice/group). Scale bar, 20 μm. **h, i** Quantifications of cluster number and area in BCAS mice treated with control AAV or AAV-*Tie1-Cav-1* (*n* = 5 mice/group). **j–l** Representative immunofluorescence images and quantifications showing the changes in the percentage of BCAS1⁺ newly formed immature oligodendrocytes and CC1⁺Olig2⁺ mature oligodendrocytes (*n* = 5 mice/group). Scale bar, 20 μm. **m, n** Representative MBP and MAG staining and quantifications (*n* = 5 mice/group). Scale bar, 20 μm. All data are presented as the mean ± SD. The data were compared by paired *t*-test. Source data are provided as a Source Data file.

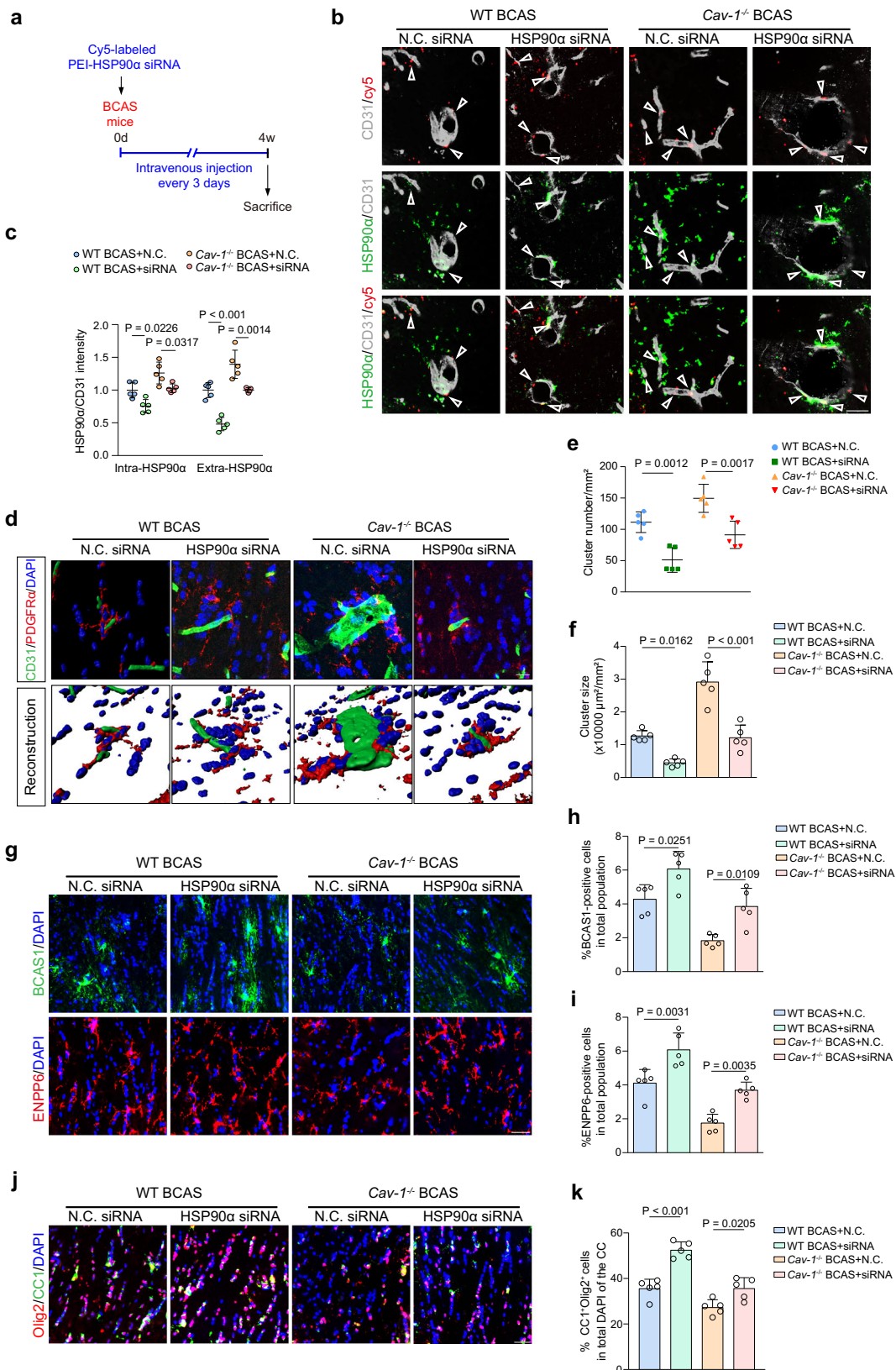

**Fig. 5 | HSP90α siRNA rescues white matter abnormality in wild type and Cav-1[−/−] BCAS mice. a** Flow chart. **b** Representative images of Cy5-siRNA (red) uptake in CD31[+] vessels (grey) with HSP90α (green) in the CC of wild type and *Cav-1[−/−]* mice (*n* = 5 mice/group). Scale bar, 20 μm. **c** Quantification of endothelial and secretory HSP90α intensity (normalized to CD31 intensity) (*n* = 5 mice/group). **d** Representative staining and reconstruction images of PDGFRα[+] OPC clusters attached on the vessel (*n* = 5 mice/group). Scale bar, 20 μm. **e, f** Quantification of

frequency and sizes of perivascular clusters (*n* = 5 mice/group). **g**–**i** Representative images and quantifications of newly-formed pre-myelinating cells (BCAS1[+] or ENPP6[+]). **j, k** Representative images, and quantifications showing the changes in mature oligodendrocytes (CC1[+]Olig2[+]) (*n* = 5 mice/group). Scale bar, 20 μm. All data are presented as the mean ± SD. The data were compared by one-way ANOVA with Tukey post hoc test. Source data are provided as a Source Data file.

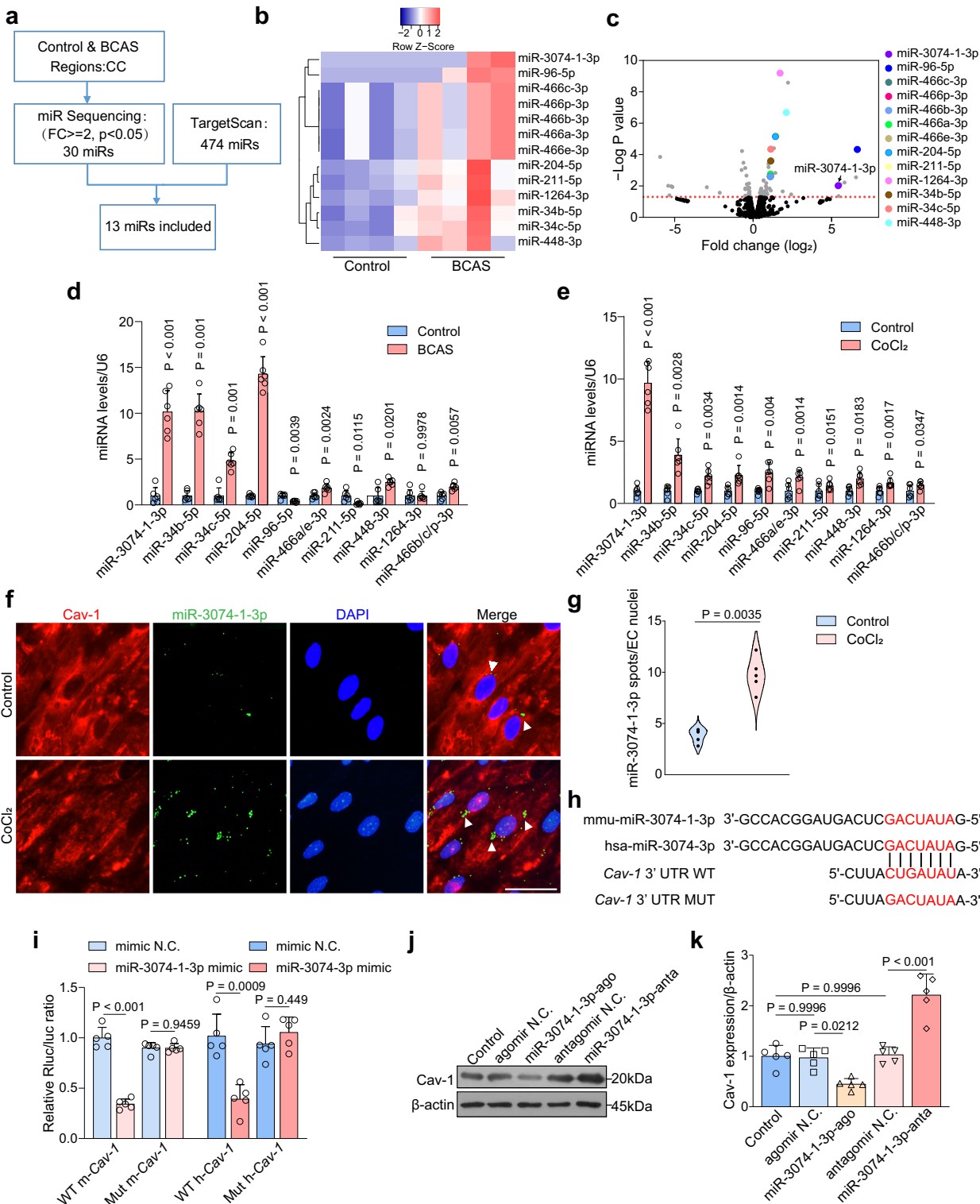

**Fig. 6 | Endothelial miR-3074-1-3p targeting Cav-1 is upregulated after BCAS.**
**a–c** Systematic screening of upregulated miRNAs targeting Cav-1 after 4 weeks of BCAS ($n = 4$ mice/group). **d, e** Verification of 13 miRNAs in the CC of BCAS mice ($n = 6$ mice/group) and BMECs under chronic hypoxia ($n = 6$ independent primary cell cultures/group). **f, g** Combined FISH and immunofluorescence labeling of miR-3074-1-3p and Cav-1 in BMECs of control and CoCl$_2$ groups ($n = 5$ independent primary cell cultures/group). Scale bar, 50 μm. **h, i** Dual luciferase reporter assay of the interaction of miR-3074-(1)-3p mimic and wild type mouse/human Cav-1 ($n = 5$

independent primary cell cultures/group). **j, k** Immunoblotting and quantification showing the Cav-1 level in BMECs transfected with agomir or antagomir. Results represent the mean for the relative band intensity of five replicates. For immunoblotting experiments, protein samples derived from the same experiment and gels/blots were processed in parallel. All data are presented as the mean ± SD. The data in **k** were compared by one-way ANOVA with Tukey post hoc test. The data in others were compared by paired $t$-test. Source data are provided as a Source Data file.

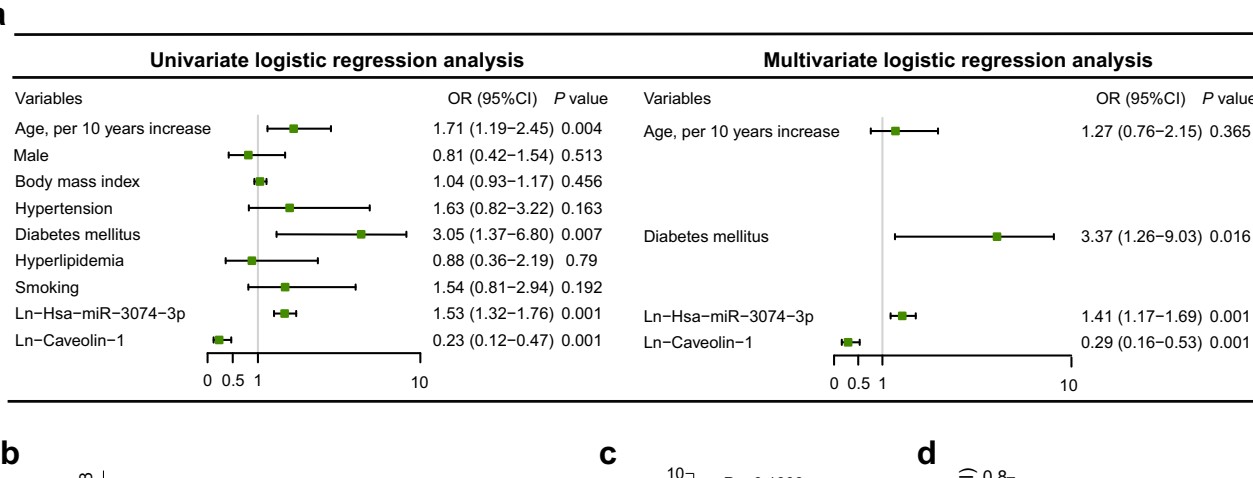

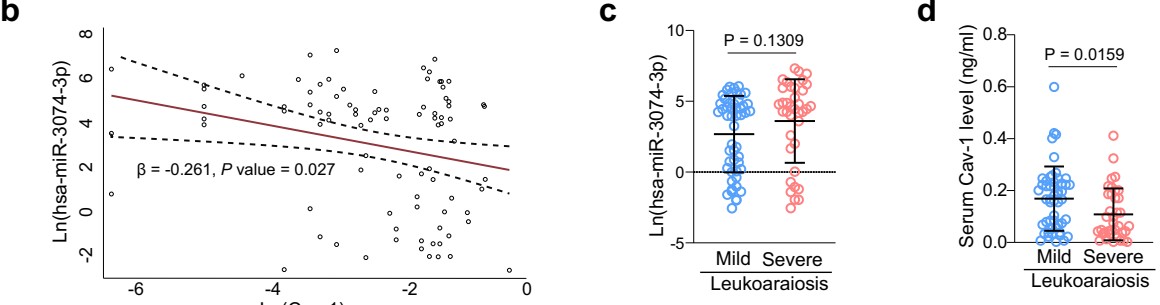

**Fig. 7 | Human homolog miR-3074-3p regulating Cav-1 is identified as a potential signature in leukoaraiosis. a** The univariate and multivariate logistic regression analysis with odds ratio (OR) values for leukoaraiosis. In the univariate analysis, the OR values, 95% confidence intervals (CIs), and their *P*-values represent the odds of leukoaraiosis of the covariates listed in the table. In the multivariate analysis, the OR values, 95% CIs and their *P*-values represent the odds of leukoaraiosis after adjusting for the covariates listed in the univariate analysis. The green dots indicate the OR value for the listed variable, and the lines indicate 95% CI. OR to the right of the grey midline (where OR = 1) indicate higher odds of leukoaraiosis while OR to the left of the midline indicate lower odds of leukoaraiosis. *n* = 66 in control and *n* = 88 in case. **b** Linear Regression assessing the association between hsa-miR-3074-3p and Cav-1 in patients with leukoaraiosis. The solid line represents the regression line and dashed lines represent the 95% CI. *n* = 66 in control and *n* = 88 in case. **c, d** Real-time PCR showing the serum hsa-miR-3074-3p and Cav-1 level in patients with mild or severe leukoaraiosis (*n* = 66 in control and *n* = 88 in case; mean ± SD.; unpaired *t*-test). Dots are all the data points including outliers. Source data are provided as a Source Data file.

attributed this phenomenon to an endogenous protective mechanism in chronic ischemia, whose reservation on OPC differentiation surpassed the inhibition from dysfunctional endothelial cells in the early phase. However, along with the hypoperfusion, proliferated OPCs were increasingly gathered on the blood vessels, more like in a "traffic jam", with growing cluster frequency and size. The overwhelmed differentiation of endogenous OPCs may act in concert with increased oligodendrocyte death[7], leading to a significant decline in pre-myelinating and mature oligodendrocytes at 4 weeks after BCAS. The OPC clustering not only suggested a substantial defect in single-cell perivascular interaction but also indicated vascular detachment failure, which may disturb OPC dispersal in the area of demyelination. Hence, it led us to investigate the modulatory mechanism behind oligovascular crosstalk after BCAS.

Cav-1 has been reported to express in different cells in the CNS[37], including endothelial cells, pericytes and OPCs themselves. Through co-immunostaining, we found that Cav-1 was mostly expressed in the endothelial cells. Thus, the changes observed in *Cav-1⁻/⁻* mice were mainly due to the knockout of endothelial *Cav-1*. However, limitations existed, as it was quite difficult to exclude the effects of other Cav-1-expressing cells in the global *Cav-1⁻/⁻* mice. To this end, the genetic delivery of AAV-*Tie1-Cav-1* was introduced before BCAS to enhance Cav-1 expression exclusively in endothelial cells. The findings that aberrant vascular-OPC association and demyelination were remarkably attenuated by AAV-*Tie1-Cav-1* suggest that the function of Cav-1 linking endothelial damage and ischemic demyelination may be cell-autonomous in endothelial cells. Cav-1 can maintain cellular

structure and normal permeability through the regulation of junction proteins expression and assembly[38]. Loss of Cav-1 could trigger BBB hyper-permeability and inflammatory injury[39], confirming our data that BCAS induced simultaneous reduction of endothelial Cav-1 and junction proteins. Cav-1 is an important regulator in cell signal transduction[40]. Mounting evidence has shown that HSP90 can be accumulated in the Cav-1 microdomain[41,42]. Competitive association with Cav-1 could facilitate HSP90 release[43]. We found endothelial Cav-1 could co-localize with HSP90α in the membrane. However, this interaction was reduced by hypoxia/ischemia. The mechanism underlying HSP90α secretion by BMECs remains unclear. We learned that Cav-1 could bind to the protein kinase A (PKA), impeding the PKA signaling[44]. PKA has been reported to phosphorylate HSP90α and promote the translocation and exocytosis of HSP90α[45,46]. We are herewith proposing that hypoxia/ischemia promoted HSP90α release through the interference of Cav-1. Moreover, it revealed that Cav-1 overexpression by AAV could restrain intra- and extra-endothelial HSP90α, while HSP90α siRNA did not change Cav-1 level. Thus, we identified HSP90α as a downstream secretory factor of endothelial Cav-1. Our data suggest that hypoxia-treated BMECs released high levels of HSP90α, leading to the cessation of OPC differentiation. In line with this, the BMECs of stroke-prone spontaneously hypertensive rats could also secrete HSP90α, which caused a reduction in OPC maturation[35]. Though we have validated that the *Hsp90α* mRNA was mainly enriched in endothelial cells after BCAS, the results should be interpreted with caution, as the expression or secretion of HSP90α from other cells cannot be excluded. The

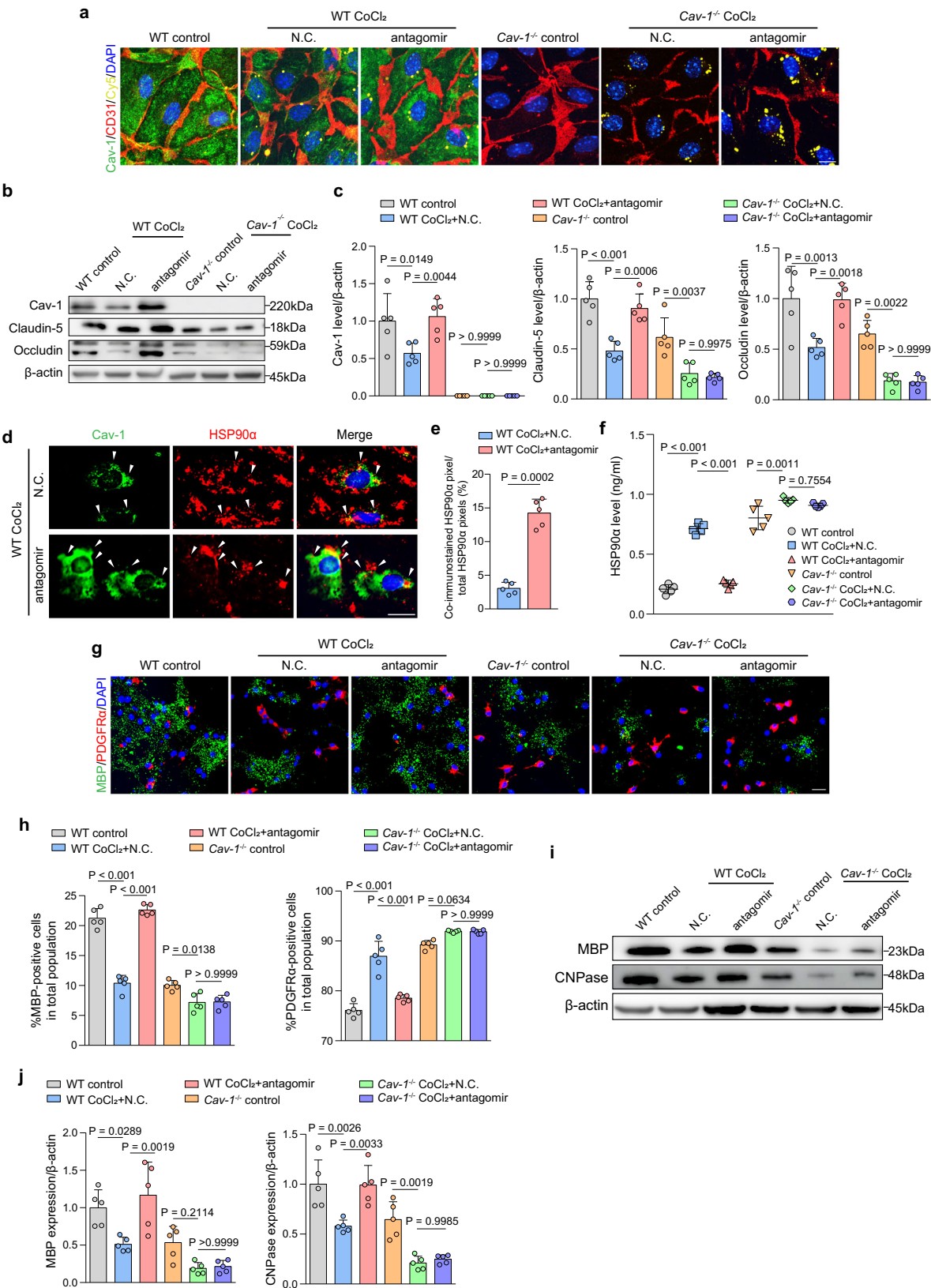

mechanism was unknown regarding the secreted HSP90α on OPCs. It is reported that secreted HSP90α is capable of binding to low-density lipoprotein receptor-related protein 1 (LRP1)[47,48], which is highly expressed in OPCs and has been defined as a negative regulator of OPC maturation[49]. Moreover, as a chaperone of Cxcr4[50], increased HSP90α may induce Cxcr4 activation and mediate OPC

attraction to the endothelium, preventing OPC recruitment and differentiation[11].

We identified hypoxia-responsive miR-3074-1-3p in the BMECs and validated that miR-3074-1-3p was an upstream regulator of Cav-1 by specifically binding with its 3′ UTR. Under the context of chronic ischemia, PEI-antagomir-induced inhibition of miR-3074-1-3p

**Fig. 8 | Antagomir treatment restores BMECs function and OPC differentiation via Cav-1 in vitro. a** BMECs transfected with Cy5-N.C. or Cy5-antagomir (yellow) were double immunostained by Cav-1 (green) and CD31 (red) (*n* = 5 independent primary cell cultures/group). Scale bar, 20 μm. **b, c** Immunoblotting and quantitative analyses of Cav-1, ZO-1, Occludin, and Claudin-5 levels. Results represent the mean for the relative band intensity of five replicates. **d, e** Representative co-immunostaining images and quantification of Cav-1 (green) and HSP90α (red) in control- or antagomir-treated endothelial cells in hypoxic conditions (*n* = 5 independent primary cell cultures/group). Scale bar, 20 μm. **f** ELISA for detection of HSP90α in CM from different groups (*n* = 5 independent primary cell cultures/

group). **g, h** Representative immunofluorescent images and quantifications showing the differentiated cells (MBP⁺) cultured in CM from BMECs of different groups (*n* = 5 independent primary cell cultures/group). Scale bar, 20 μm.
**i, j** Immunoblotting and quantifications of myelin proteins. Results represent the mean for the relative band intensity of five replicates. For immunoblotting experiments, protein samples derived from the same experiment and gels/blots were processed in parallel. All data are presented as the mean ± SD. The data in **e** were compared by paired t-test. The data in others were compared by one-way ANOVA with Tukey post hoc test. Source data are provided as a Source Data file.

---

attenuated physical coupling of endothelial cells and OPCs and mitigated demyelination dependent on Cav-1, further emphasizing the core value of Cav-1 in therapeutics for ischemic white matter disease. Considering that the abnormal OPC-endothelial interactions were reversible later in the disease when vascular dysfunction was already present, miR-3074-1-3p inhibition may be of great clinical value in ischemic demyelination, or other white matter diseases sharing a similar mechanism.

In summary, we show the mechanism engaging in the endothelial-oligodendroglial association in chronic ischemia, and regulation of this aberrant interaction could restore myelination. Chronic ischemia caused vascular dysfunction and OPC perivascular clustering, in which endothelial Cav-1 occupied a central place. A significant decline of endothelial Cav-1 was responsible for HSP90α secretion, blocking oligodendrogenesis. Specific overexpression of endothelial Cav-1 could reverse ischemic myelin damage. Intervention by antagomir preserved endothelial Cav-1 and suppressed the vascular HSP90α release, thereby attenuating OPC-vascular interactions and demyelination. Therefore, our findings may shed light on the therapeutic implications of endothelial Cav-1 in diseases involving white matter pathologies.

## Methods

All experiments were performed following the National Institutes of Health Guide and approved by Jinling Hospital Animal Care Committee. The participants or legal representatives who acknowledged the use of blood samples signed an informed consent form before being included in the study. The protocol was approved by the Ethics Committee of Jinling Hospital.

### Antibodies

Antibodies against CNPase (ab6319), GFAP (ab53554), Ki67 (ab15580), MAG (ab89780), MBP (ab40390), PLP (ab28486), Cav-1 (ab2910), Collagen IV (ab6586), HSP90α (ab79849), NeuN (ab104224) were purchased from Abcam, UK; antibody against BCAS1 (bs-11462R) was purchased from Bioss, China; antibody against CD31 (550274) was purchased from BD Biosciences, USA; antibodies against β-actin (#8457), Cav-1 (#3267) were purchased from Cell Signaling Technology, USA; antibody against HSP90α (ADI-SPS-771-D) was purchased from Enzo Life Sciences, USA;antibody against Occludin (R1510-33) was purchased from Huabio, China; antibodies against ENPP6 (PA5-25140), Claudin-5 (35-2500), Occludin (71-1500) were purchased from Invitrogen, USA; antibodies against CC1 (OP80), Olig2 (AB9610) were purchased from Millipore, USA; antibody against O4 (MAB1326-SP) was purchased from R&D Systems, USA; antibodies against PDGFR-α (sc-398206), ZO-1 (sc-33725), MBP (sc271524), Iba-1 (sc-32725), Cav-1(sc-53564) were purchased from Santa Cruz Biotechnology, USA; All antibodies were used at a dilution of 1:50–1:1000 for immunofluorescence, 1:500–1:1000 for immunoblotting analysis unless otherwise specified. Secondary antibodies were donkey-anti-mouse or anti-rabbit or anti-rat conjugated with either Alexa 488 or Alexa 594 or Alexa 647 (Jackson, USA; 1:400), goat anti-mouse or rabbit or rat IgG HRP (Cell Signaling Technology, USA; 1:5000). Mouse anti-goat IgG-HRP (sc-2354, Santa Cruz Biotechnology) was used for co-immunoprecipitation.

### Mice

Adult male (24–29 g; for BCAS surgery) and C57BL/6J mice (6–8 weeks old; for BMECs primary culture) were purchased from Gempharmatech co., Ltd (Nanjing, Jiangsu, China). *Cav-1* knock-out mice (Cav-1^tmIMls/J, C57BL/6 background, 007083) were purchased from the Jackson Laboratory (Bar Harbor, Maine, USA). All animals were housed in a 12 h light/dark cycle at approximately 25 °C and provided with free access to food and water.

### BCAS surgery

For BCAS surgery[51], mice were anesthetized with 5% isoflurane and maintained with 2% isoflurane in oxygen (RWD Life Science Co., LTD). Both common carotid arteries (CCAs) were carefully dissected and exposed through the midline incision. A microcoil with an internal diameter of 0.18 mm (Sawane Spring Co. Japan) was twined around the CCA on one side, followed by another twinning on the other side 30 minutes later. Rectal temperature was maintained between 36.5 °C and 37.5 °C during surgery. After the operation, the mice were taken care of and provided with ad libitum access to water and food. To study the disease progression, mice were divided into four time-point groups: Control, BCAS_1 week, BCAS_2 weeks and BCAS_4 weeks. Mice with sham operation served as controls. Control mice were given a skin incision and their CCAs were exposed without inserting the microcoil.

### CBF measurements

To confirm the BCAS modeling, CBF was measured before, 1 week, 2 weeks, and 4 weeks after surgery by laser speckle contrast imaging (RWD Life Science Co., LTD). Anesthetized by 2% isoflurane in oxygen, mice were placed in the prone position. The skull was exposed by a midline scalp incision and cleaned with sterile normal saline. Color-coded blood flow images obtained in high-resolution mode (2048 ×2048 pixels; 1 image/sec) were captured by a CMOS camera positioned above the head and transferred to a computer for analysis. By the color image program incorporated in the flowmetry system, images were analyzed to obtain the average value of blood flow. The mean CBF of 5 mice in each group was determined. The value of blood flow was expressed as a percentage of the baseline blood flow.

### Cognitive tests

Spatial learning and memory were evaluated by MWM[52]. Briefly, mice were trained for five consecutive days to find a hidden platform. On the sixth day, the platform was removed and mice were placed to swim for 1 min in probe trial. The escape latency to find the platform, the swim path length, and the percentage time in the platform quadrant and platform crossovers were recorded and analyzed by the ANY-maze video tracking software (Stoelting, USA). The NOR test was conducted to assess novelty preference[53]. In brief, mice were habituated in a box for testing and got familiarization with 2 identical objects in the box. Then memory function was evaluated either 1 h or 24 h later by replacing one of the familiar objects with a novel one that was different in shape and appearance. The time spent on exploring the familiar object (F) and the new object (N) was respectively recorded in a 5 minute-trial. The index of discrimination defined as (N-F)/(N + F) was calculated[54,55].

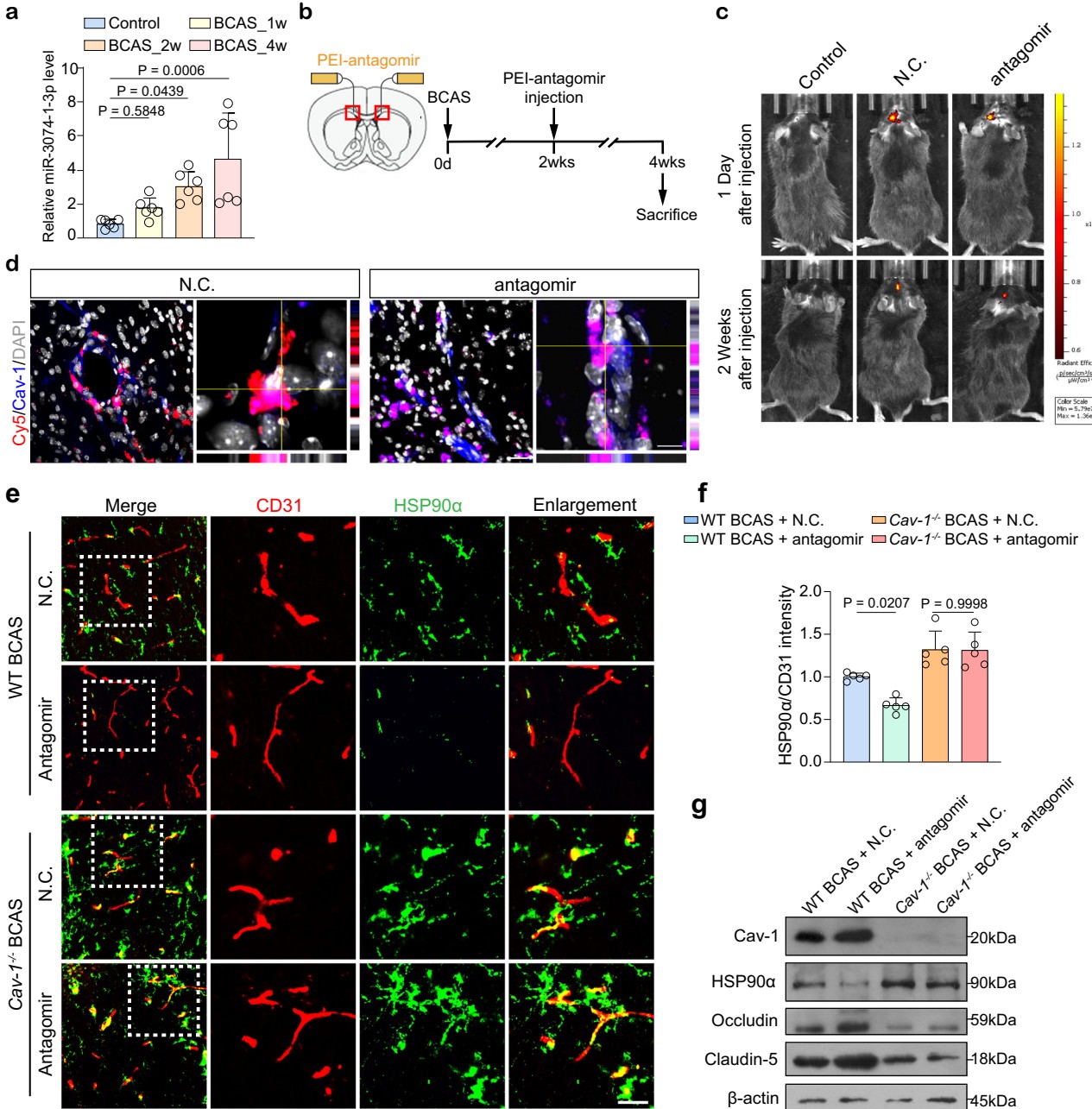

**Fig. 9 | The endothelial Cav-1 and HSP90α level after miR-3074-1-3p antagomir treatment in mice of 2 genotypes. a** Levels of miR-3074-1-3p in the CC at different time points after BCAS ($n = 6$ mice/group). **b** Experimental flow chart. **c** In vivo fluorescence images showing the fluorescent signal of N.C. or antagomir delivered in the CC 1 day and 2 weeks following BCAS ($n = 5$ mice/group). **d** Representative orthogonal images with *x-y, x-z,* and *y-z* views showing the uptake of Cy5-labeled nanoparticle-antagomir (red) in vessels with enforced Cav-1 expression (blue) in the CC of BCAS mice ($n = 5$ mice/group). Scale bar, 20 μm. **e, f** Representative images, and qualification showing the changes of HSP90α secretion ($n = 5$ mice/group). Scale bar, 20 μm. **g** Immunoblotting for Cav-1, HSP90α, and TJ proteins. Results represent the mean for the relative band intensity of five replicates. For immunoblotting experiments, protein samples derived from the same experiment and gels/blots were processed in parallel. All data are presented as the mean ± SD. The data were compared by one-way ANOVA with Tukey post hoc test. Source data are provided as a Source Data file.

## Cell culture and treatment

As modified from a published method[56], brains were obtained from wild-type or *Cav-1⁻/⁻* mice at the age of 6–8 weeks with cerebellum and meninges removed. After homogenization, the homogenate of brains was centrifuged at $150 \times g$ for 5 min. The pallet was then resuspended in a 15% dextran solution (mol wt 60,000–76,000, Sigma-Aldrich) and centrifuged at $400 \times g$ to isolate blood vessel fragments. The pallet of fragments was digested in collagenase/dispase (1 mg/ml, Roche) and DNase I (10 μg/ml, Roche) at 37 °C. The dissociated BMECs were washed and seeded onto coated plates in DMEM/F12 media with 20% FBS, 1% PS, 1% endothelial cell growth supplement (SclenCell), 1% l-glutamine, 1% heparin, and 2 ng ml⁻¹ bFGF (Biolegend). In vitro hypoxia was induced by either chronic $CoCl_2$ (10 μM, Sigma-Aldrich) for 5 consecutive days or OGD treatment for 4 h. Primary mouse OPCs were isolated from pups of postnatal days 0–2 as reported[57]. In brief, mixed glial cells were cultured in DMEM/F12 with 10% FBS and 1% PS for 7 days. Then the flasks were shaken on the orbital shaker at $200 \times g$ for 1 h. After washed, the flasks continued to shake for another 18–20 h at

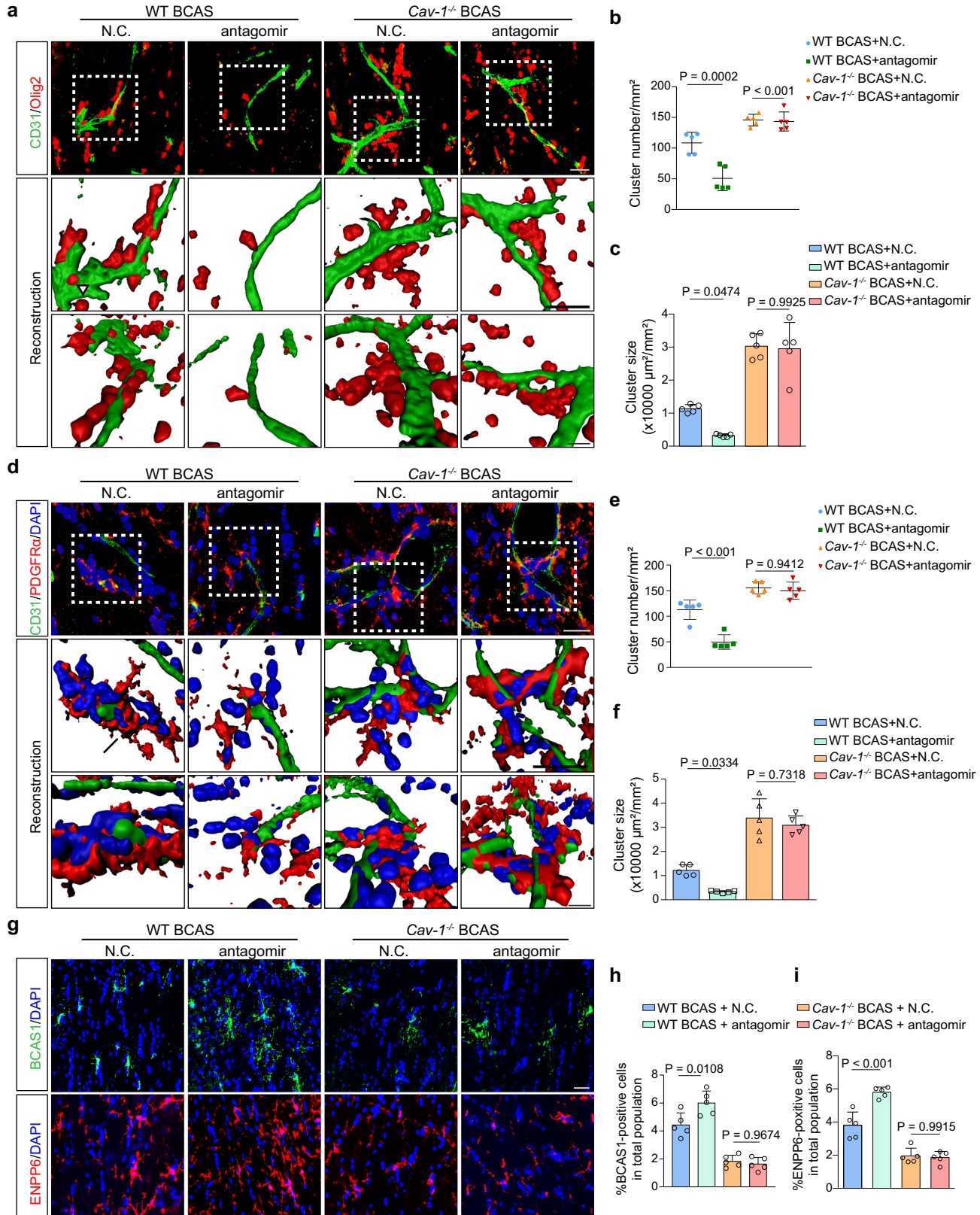

**Fig. 10 | PEI-antagomir ameliorates OPC perivascular clustering and promotes remyelination via Cav-1. a** Representative immunofluorescent images and reconstruction showing the attenuation in perivascular Olig2$^+$ cell clusters in wild type and *Cav-1*$^{-/-}$ mice treated with antagomir (n = 5 mice/group). Scale bar, 20 μm. **b**, **c** Quantitative analyses of cluster number and size around CD31$^+$ vessels (*n* = 5 mice/group). **d** Perivascular clustering of PDGFRα$^+$ OPCs in mice of 2 genotypes subjected to N.C. or antagomir (*n* = 5 mice/group). Scale bar, 20 μm. **e**, **f** Quantifications of the cluster number and size (*n* = 5 mice/group). **g**–**i** Representative staining of newly-formed pre-myelinating BCAS1$^+$ or ENPP6$^+$ cells (*n* = 5 mice/group). Scale bar, 20 μm. All data are presented as the mean ± SD. The data were compared by one-way ANOVA with Tukey post hoc test. Source data are provided as a Source Data file.

$4 \times g$. OPCs were obtained and plated onto coated plates in DMEM/F12 media containing 1% BSA (Gibco), 20 ng ml$^{-1}$ ITSS (Roche), 20 ng ml$^{-1}$ PDGF-BB (Biolegend) and 20 ng ml$^{-1}$ bFGF (Biolegend).

## CM transfer
After exposure to CoCl$_2$/OGD or antagomir/agomir, the medium was removed and BMECs were washed by PBS. Then BMECs were cultured in serum-free medium, including DMEM/F12 media with 1% PS, 1% endothelial cell growth supplement (SclenCell), 1% l-glutamine, 1% heparin, and 2 ng ml$^{-1}$ bFGF, for another 2 days. CM was collected and stored at −80 °C until use. Before applying to OPCs, the CM was centrifuged at $230 \times g$ for 5 min to remove the cell debris. The normal culture media was replaced by CM 1 day after OPCs were seeded onto the plates. OPCs were cultured under CM for 2 days before being analyzed. For HSP90α experiments[35,58], recombinant HSP90α (1 µg/ml, #ADI-SPP-776-D, Enzo Life Sciences) or HSP90α blocking antibody (1.7 µg/ml, #ADI-SPS-771-D, Enzo Life Sciences) was added into the CM before transferring to OPCs.

## Study participants and data collection
From January 2021 to April 2021, 154 patients who were more than 45 years old and referred to Jinling Hospital for further assessment due to dizziness and nonspecific headaches without migraines were recruited. All subjects performed an MRI examination for assessing the presence and severity of leukoaraiosis. Baseline data including age, gender, body mass index, and vascular risk factors (including hypertension, diabetes, hyperlipemia, and smoking) were recorded. The exclusion criteria of this study were as follows: (1) leukoaraiosis which was not due to vascular origins, such as multiple sclerosis and leukodystrophy; (2) signs of neurological deficit; (3) history of stroke, brain trauma, intracranial tumor, central nervous system infection, active malignancy, thyroid diseases, autoimmune diseases, and active or chronic inflammatory diseases; (4) contraindication for MRI examination. The patients gave written informed consent before entering the study. The patients were not financially compensated.

## Brain MRI examination and analysis
MRI scan of whole brain was performed with a 3.0 T system (TIM Trio; Siemens Healthineers, Erlangen, Germany) in all patients within 7 days after admission. Leukoaraiosis was defined on T2-fluid-attenuated inversion recovery (FLAIR) imaging (T2-FLAIR parameters: number of slices = 25, slice matrix = 512 × 512, field of view (FOV) = 220 × 220 mm2, repetition time (TR) = 8000 ms, echo time (TE) = 93 ms, echo train length = 24, slice thickness = 5 mm, spacing between slices = 7 mm, flip angle = 130°, inversion time = 2500 ms.) and graded using the Fazekas scale[59]. According to previous studies, we dichotomized leukoaraiosis according to its severity as mild (Fazekas scores of 0 or 1) and severe (Fazekas scores ≥ 2) in periventricular and deep subcortical region[60,61]. All images were analyzed by two experienced MRI-specialized neuroradiologists who were blinded to the patients' clinical information. In case of disagreement, lesions were ascertained by consensus. An intra-rater reliability test was performed on 50 subjects, the kappa values for the presence and severity of leukoaraiosis were 0.91 and 0.87, respectively.

## Antibody array
BMECs in the control and CoCl$_2$ groups were washed and cultured in the serum-free media for 2 days. CM was then collected and concentrated for the array test. The antibody array covering more than 1300 antibodies (#SET100, Full Moon BioSystems) was performed according to the manufacturer's manual.

## ELISA for HSP90α and Cav-1
ELISA was conducted as specified in the protocol of the HSP90AA1 ELISA Kit (AVIVA SYSTEMS BIOLOGY, USA) and Human Cav-1 ELISA Kit

(Sabbiotech, USA). Briefly, 100 µl of CM or patient serum was added into each well of the microplate and incubated at 37 °C for 2 h. The liquid was then removed and a 100 µl Detector Antibody was added to the well for 1 h of incubation. After washing, 100 µl avidin-HRP conjugate was added and incubated for another 1 h. With wells washed, 90 µl TMB substrate was added, followed by the addition of a stop solution 30 minutes later. O.D. absorbance was obtained at the wavelength of 450 nm.

## miRNA sequencing, miRNA extraction, and real-time PCR
The RNA was extracted from the CC tissue of control and BCAS_4w mice ($n = 4$ in each group). miRNA sequencing was performed by Illumina HiSeqTM 2500 platform and data analysis was conducted by Ribo Bio (Co., Ltd, Guangdong, China). The quantity and integrity of RNA yield were evaluated by the K5500 (Beijing Kaiao, China) and Agilent 2200 TapeStation (Agilent Technologies, USA). Total RNA (1 µg) of each sample was used to prepare small RNA libraries by NEBNext® Multiplex Small RNA Library Prep Set for Illumina (NEB, USA) according to manufacturer's instructions. The libraries were sequenced by HiSeq 2500 (Illumina,USA) with single-end 50 bp at Ribobio Co. Ltd. The raw reads were processed by filtering out containing adapter, poly 'N', low quality, smaller than 17nt reads by FASTQC to get clean reads. Mapping reads were obtained by mapping clean reads to reference genome of by Burrows-Wheeler-Alignment Tool. The miRNA expression was calculated by Reads Per Million (RPM) values (RPM = (number of reads mapping to miRNA/ number of reads in clean data) × 10$^6$). The expression levels were normalized by RPM, which equals to (number of reads mapping to miRNA/number of reads in clean data) × 10$^6$. Differentially expressed miRNAs were then screened by an adjusted $P$ value of <0.05 and at least a two-fold change of expression.

Serum miRNAs were extracted from patients' samples according to the manufacturer's manual of miRNeasy Serum/Plasma kit (QIAGEN, Germany). Total RNA was extracted from cells and tissues using TRIzol (Invitrogen, USA), followed by reverse transcription using RevertAid First Strand cDNA Synthesis Kit (Thermo Scientific, USA). Real-time PCR based on SYBR Green was performed by Stratagene Mx3000P QPCR system (Agilent Technologies, USA). Primers for reverse transcription were listed in Table S2. The primer pairs used for real-time PCR were listed in Table S3.

## miRNA transfection and dual-luciferase reporter assay
With Lipofectamine 3000 Reagent (Thermo Fisher Scientific, USA), BMECs were transfected with 50 nM miR-3074-1-3p N.C./agomir/ antagomir (Ribo Bio, Co., Ltd, China). The 293 T cells were seeded in the 96-well plates and co-transfected with a 1 µg vector containing 3' UTR of mouse/human Cav-1 and 100 nM mimic N.C./mmu-miR-3074-1-3p/hsa-miR-3074-3p mimic (Ribo Bio, Co., Ltd, China) using Lipofectamine 3000. After 24 h of transfection, the luciferase activity was measured by the Dual-Luciferase® Reporter Assay system (Cat# E1910, Promega, USA).

## Formulation, injection, and tracking of siRNA, antagomir, and AAV
HSP90α-siRNA or miR-3074-1-3p antagomir was diluted with 10% glucose solution and RNAse/DNAse free water. Separately, in vivo-PEI™ was diluted to the same volume by 10% glucose. The solutions were then mixed and incubated at room temperature for 15 min to form complexes with an N/P ratio of 6. For Cav-1$^{-/-}$ Mice, 20 µg PEI-N.C. siRNA or PEI-HSP90α siRNA was injected into mice via tail vein every 3 days for a month.

To over-express endothelial Cav-1 in BCAS mice, a cDNA encoding Cav-1 sequence, the enhanced GFP reporter gene, and endothelial-specific Tie1 promoter were produced and inserted into AAV packaging vectors. AAV containing control and Cav-1 vectors was purchased

from GeneChem co., Ltd (Shanghai, China) and injected in the CC (0.5 mm anterior-posterior, 1.0 mm medial-lateral, −2.1 mm dorsal-ventral relative to bregma) using stereotaxic injection 3 weeks before BCAS surgery. All the animals were injected bilaterally with 1 μl of AAV-*Tie1*-C (1.13 × E13 v.g/ml) or AAV-*Tie1-Cav-1* (1.95 × E13 v.g/ ml).

For continuous in vivo delivery of antagomir, osmotic infusion pumps (model 1002, Alzet) loaded with PEI-antagomir N.C. or PEI-antagomir and cannulas (Plastics One, USA) were connected and stereotactically inserted in the CC on the 15th-day post BCAS. Approximately a total of 2 μg PEI-antagomir was infused daily for a consecutive 14 days[62]. For antagomir tracking in mice, fluorescence whole-animal imaging was performed on the IVIS imaging system 1 day and 2 weeks after the implantation of pumps loaded with Cy5-labeled antagomir.

## TEM and immuno-TEM

For TEM, the CC tissue was fixed in 2.5% glutaraldehyde[63]. After dehydration and embedding, samples were cut to 60–80 nm slices and scanned by an H7500 Transmission Electron Microscope (Hitachi, Japan). The TJ length, gap area, the percentage of myelinated fibers, and myelin thickness were measured and analyzed. For immuno-TEM, samples were cut into 70–80 nm slices after fixed and embedding. By immunogold labeling of Olig2, samples were observed and images were captured with H7500 Transmission Electron Microscope (Hitachi, Japan)[15].

## Tissue processing and immunohistochemistry

Mice were anesthetized and transcardially perfused with PBS and 4% PFA. Brains were dissected, postfixed in 4% PFA, and dehydrated in gradient sucrose. Embedded in Tissue-Tek O.C.T compound (Sakura Finetek, USA), frozen brains were cut into 20 μm-thick sections. For immunostaining, sections or cell coverslips were fixed in PFA, followed by blocking with 5% goat serum, 1% BSA, and 0.1% Triton X-100 for 1 h. After incubation with primary antibodies at 4 °C overnight, the sections were incubated with secondary antibodies. Finally, images were captured using a BX51 microscope (Olympus, Japan) or a laser-scanning confocal microscope (FV3000, Olympus) and were prepared using Adobe Photoshop (version 21.0.2). The 3D reconstruction was analyzed by IMARIS software (Bitplane).

White matter LFB staining was carried out with the degree of white matter damage scored[7,51] and Black-gold II staining (Sigma-Aldrich, USA) was performed with the calculation of myelinated area[64].

For histochemical evaluation of BBB integrity, mice were injected with FITC-dextran (3 kDa, Sigma-Aldrich)[65] via CCA under deep anesthesia. After circulation for 90 min, brains were isolated and immediately fixed in 4% PFA, followed by dehydration in gradient sucrose. Dextran was visualized with a laser-scanning confocal microscope (FV3000, Olympus). TUNEL staining (Beyotime, China) and EdU proliferation assay (Ribo Bio, Co., Ltd, China) were performed on cell coverslips following the manufacturer's protocol[66,67]. For TUNEL staining, cell coverslips were fixed in 4% PFA and stained with TUNEL reagent. For EdU proliferation assay, cells were incubated with EdU buffer and fixed with 4% PFA. EdU solution was then added to coverslips followed by the staining of DAPI. Finally, the slides were visualized by fluorescence microscopy. The positively labeled cells were calculated by Image J.

Combined FISH for miR-3074-1-3p and HSP90α with immunostaining was conducted. The FAM-labeled miR-3074-1-3p probe was used to detect miR-3074-1-3p in Cav-1-stained endothelial cells. The FAM-labeled HSP90α probe was used to detect HSP90α in CD31-labeled endothelial cells, PDGFRβ-labeled pericytes, GFAP-labeled astrocytes, and Olig2-labeled oligodendroglial lineage cells. Briefly, the frozen sections were fixed in 4% PFA and incubated with probe hybridization solution overnight. The slides were then washed and blocked for the immunostaining. Nuclei were stained with DAPI and

the slides were mounted in an anti-fade reagent with DAPI (Vector Laboratories).

## Co-immunoprecipitation and Immunoblotting

Lysates were extracted from the CC brain tissue, microvascular segments, and cultured BMECs. Briefly, as for microvascular segment extraction, a pool of two mouse brains was defined as one group. Collected brains were homogenized and centrifuged at 150 × g. The pellet was then resuspended in 15% dextran solution for twice layered centrifugation at 400 × g for 10 min. The supernatant and upper myelin debris were discarded and the microvascular segment at the bottom was resuspended in RIPA lysis buffer (Cell Signaling Technology, USA) with 1% PMSF. Also, brain and cell lysates were harvested using RIPA lysis buffer with 1% PMSF. The concentration of protein was quantified by BCA Protein Assay Kit (Beyotime, PR China). For co-immunoprecipitation, protein extracted from endothelial cells was incubated with Cav-1, HSP90α or negative control IgG overnight in a 4 °C shaker. Protein A/G agarose beads were then added to the complex and incubated for 4 h at 4 °C. After centrifugation, the agarose beads were collected and protein was eluted. For immunoblotting, an equal amount of protein was loaded, separated by SDS-PAGE, and incubated with primary antibodies. After incubation of the secondary antibodies for 1 h, the special protein signals were detected by Immobilon Western Chemiluminescent HRP substrate (Millipore, USA).

## Quantifications

**OPC cluster quantification.** CD31/PDGFRα/Olig2 triple staining was used to detect the association of OPCs with blood vessels in the corpus callosum. An OPC cluster is defined as a perivascular aggregation of OPCs with more than 4 cell bodies in direct contact with blood vessels[36]. The total area of the cluster by PDGFRα⁺ or Olig2⁺ signals was outlined and measured by Image J, which was further normalized to the area of the image.

**Vessel density and length quantifications.** The vessel density was quantified by dividing the CD31⁺ vessel number by the area of the selected region. CD31⁺ vessel length was measured by Image J, which was normalized to the area of the image.

**BBB leakage quantification.** The extravasated dextran was outlined and its fluorescence intensity was measured by Image J, which was then normalized to the fluorescence intensity of the vessels.

**Quantifications for HSP90α.** The extra- and intra-endothelial HSP90α were chosen respectively. And the HSP90α level changes were measured by dividing the HSP90α fluorescence intensity by the CD31 fluorescence intensity.

**White matter damage score quantification.** The severity of demyelinated lesions was graded by LFB staining as follows: grade 0 (normal), grade 1 (disarrangement of the nerve fibers), grade 2 (marked vacuoles), and grade 3 (the disappearance of myelinated fibers)[68].

**Quantifications for TEM.** (1) We chose 4 capillaries and 4 arterioles for each sample. The area of tight junction gap between endothelial cells was measured and caveolae number was counted by Image J. (2) The percentage of myelinated axons and myelin thickness were calculated and analyzed.

**The fluorescent intensity and area quantifications.** Cell counting and fluorescence intensity analyses were conducted on five randomly chosen fields for each sample using Image J. The results were normalized to the areas of interest or the total cell population in the selected region.

## Statistics

Results were analyzed by GraphPad Prism software, SPSS version 24.0 (SPSS Inc., Chicago, IL, USA), and Stata version 16.0 (StataCorp, College Station, TX). All data were expressed as mean ± S.D. Continuous variables were assessed by paired t-test for two groups and one-way ANOVA followed by Tukey post hoc test for multiple comparisons. Escape latency and swimming path length in the MWM test were analyzed by two-way repeated-measures ANOVA, followed by Tukey's post hoc test. We compared patients with and without leukoaraiosis using unpaired t-test or the Mann-Whitney U tests for continuous variables and Pearson $\chi^2$ and Fisher exact tests for categorical variables. We constructed a logistic regression analysis to evaluate the risk factors of leukoaraiosis. A multivariable regression model was adjusted for variables with a $P$-value <0.1 in univariate analysis. Results were presented as odds ratio with a 95% CI. We also performed the linear regression to assess the association between levels of hsa-miR-3074-3p and Cav-1 in patients with leukoaraiosis. Two-sided P value < 0.05 was considered statistically significant.

## Reporting summary

Further information on research design is available in the Nature Research Reporting Summary linked to this article.

## Data availability

All relevant data are available within the manuscript and the supplementary materials. The miRNA sequencing data generated in this study have been deposited in the Sequence Read Archive (SRA) under bioprojects PRJNA886415. Source data are provided with this paper.

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

## Acknowledgements

This work was supported by the National Natural Science Foundation of China (No. 81701180 and 82171331 to Y.X., U20A20357 and 81870946 to X.L., No. 81901248 to X.Z., No. 82101406 to Y.L.), Jiangsu key research and development program (BE2020700 to X.L.), Fundamental Research Funds for the Central Universities (WK9110000056 to X.L.) and China Postdoctoral Science Foundation (Nos. 2019T120968 and 2019M664011 to Y.X.).

## Author contributions

X.L. and Y.X. conceived and supervised the study. Y.X. and R.Y. designed the research and wrote the manuscript. Y.Z., T.W., and Z.H. carried out animal experiments. Y.Z., T.W., and P.X. performed the cellular experiments. T.W. and P.X. assisted with imaging experiments and image reconstruction. Y.Z., X.Z., and Y.X. recruited the patients, collected clinical samples, and verified serum miRNA. W.Z., Y.L., and K.H. who were blinded to the group information, analyzed the data.

## Competing interests

The authors declare no competing interests.
