## [Peer Review File · Nature Communications]

Vascular endothelium deploys caveolin-1 to regulate oligodendrogenesis after chronic cerebral ischemia in miceREVIEWER COMMENTS

Reviewer #1 (Remarks to the Author):

Zhao and colleagues use a number of contemporary techniques to confirm the important link between vascular abnormalities and cerebral ischemic demyelination. They identify endothelial caveolin-1 as a mediator oligodendrocyte differentiation via the release of HSP90 α . Further, they identify a microRNA that acts upstream to influence the levels of caveolin-1. Overall, these findings are potentially important given the lack of understanding of ischemic demyelination. The main findings are clear and straightforward, however, some aspects of the manuscript need clarification and further explanation to ensure the validity of the findings.

Major comments

The rationale behind many aspects of the manuscript is poorly explained and can be difficult to understand. In the introduction the justification behind the study is unclear e.g. the fact that chronic cerebral ischemia causes demyelination is not stated. The precise meaning and quantification of “cluster number” and “cluster size” are not explained within the main text or the materials and methods. The use of CoCl₂ as a means to induce hypoxia is not explained. The use or reasoning of CSD supplementation is not explained. The authors should improve the text to ensure that the manuscript is easy to follow for the wide audience of Nature Communications.

Zhao et al. make the important observation that BCAS leads to an aggregation of OPCs clustered on blood vessels. The authors describe this finding as like a “traffic jam” that can be alleviated by knock down of HSP90 α or an antagomir for miR-3074-1-3p. To provide support to this hypothesis it would be beneficial to assess OPC proliferation around vessels following BCAS and if HSP90 α treatment influences OPC migration. Further, similar vascular clustering has been overserved during development in mice with overactive wnt-signaling through. This is believed to be a result of increased attraction to the vasculature through Cxcl12 – Cxcr4 signaling. Does the mechanism defined within the manuscript overlap with signaling required during development or is this mechanism distinct to adult injury?

In Figures 4K and 5J the authors quantify the number of mature oligodendrocytes and oligodendrocyte-lineage cells in the corpus callosum. As one of the major myelinated tracts in the brain, the number of oligodendrocyte-lineage cells is expected to be much higher than the ~30% stated in the manuscript. Can the authors explain why the number of oligodendrocyte-lineage cells is low?

The authors find a reduced level of myelination in the Caveolin-1 KO mice. Do these mice also show a reduction in oligodendrocyte numbers? In figure S4 the effect on vessels and myelination is described as “pathological”. Do the authors have evidence to suggest that Caveolin-1 KO mice is pathological?

Minor comments

A link between HSP90 α released by dysfunctional endothelial cells and oligodendrocyte generation is not a novel finding. Rajani et al 2018 should be referenced in regard to this. The language throughout should be improved for clarity e.g. frequent use of the word “apparently”, line 107 “to the opposite”, line 199 “retarded”

Line 76 – Yuen et al 2014 Cell (Oligodendrocyte-encoded HIF function couples postnatal myelination and white matter angiogenesis) should also be referenced.

Line 109 – “higher expression of PDGFR α indicated increased mobilisation of OPCs” – no

reference is provided to support this statement.

Line 580 – incorrect reference.

Immunohistochemistry of ENPP6 in vivo and O4 in vitro look odd. Both are usually expressed in membranes however in the images appear nuclear. High-magnification inserts would be beneficial.

MBP usually has multiple bands similar to that demonstrated in fig S9 however the rest of the immunoblots throughout the manuscript only show 1 band.

Line 378 – no quantification or evidence is provided to support the statement of reduced oligodendrocyte complexity.

Figure S1 – n and o - it is not stated which graph refers to which quantification.

Reviewer #2 (Remarks to the Author):

In this study, Zhao et al. examined the mechanisms of cell-cell interaction between endothelium and oligodendroglia under cerebral hypoperfusion. Using in vivo (a mouse model of chronic cerebral hypoperfusion) and in vitro (endothelial cells and OPCs) system, the authors reported that under the conditions of chronic ischemic conditions, Cav-1 reduction would increase the vascular secretion of HSP90a, which induces aberrant OPC accumulation in the perivascular region. In addition, the decreased Cav-1 level seemed correlated with the leukoaraiosis severity in patients. The finding about the role of Cav-1/HSP90a signaling in the compensatory oligodendrogenesis after cerebral hypoperfusion is novel, and this study provides a proof-of-concept that this pathway could be a therapeutic target for promoting white matter repair and recovery after white matter damage.

Major points:

1. Cav-1/HSP90a interaction: in the authors' summary diagram, Cav-1 binds HSP90a under the normoxic conditions, but after hypoxia, because of degradation/reduction of Cav-1, HSP90a is released from the cellular membrane. But it seems that this point was not proved in this study. Therefore, it would be required to show (i) Cav-1 keeps HSP90a around cell membrane under normoxic conditions and (ii) after hypoxic stress, Cav-1/HSP90a binding would be disrupted in endothelial cells, and (iii) the antagomir would suppress the changes by hypoxia.

2. cell source of HSP90a: although this study focused on the roles of endothelial-derived HSP90a in oligodendrogenesis after white matter damage, it seems that the increase of HSP90a level after cerebral hypoperfusion was not only from endothelial cells but also from other types of cells (Fig 3). Therefore, it would be necessary how much endothelial-derived HSP90a by Cav-1 reduction in endothelial cells would contribute to the aberrant OPC accumulation under chronic hypoxic conditions. In addition, in Fig 4, it may be also needed to provide some western blot data about how much Cav-1 upregulation in endothelial cells would increase the HSP90a level in the brain (or in the perivascular region of white matter).

3. BCAS hypoperfusion model and Leukoaraiosis: in this study, the authors prepared a mouse model of chronic cerebral hypoperfusion by BCAS operation. This model is now relatively well-accepted as a model of SIVD. However, in the clinic, leukoaraiosis could be observed in patients who are not diagnosed with SIVD. This discrepancy may need to be taken into consideration in the manuscript. In addition, the authors included both patients who had an MRI examination with a 3.0T system and a 1.5T system. But the detection sensitivity for white matter hyperintensity in the 3.0T system would be higher than the one in

the 1.5T system (PMID: 19369605), and this limitation may need to be noted.

4. HSP90a & OPC: Although the authors discussed how HSP90a negatively regulates OPC function in this manuscript, it would be recommended to try some mechanistic study into this point. In addition, the authors may need to cite previous papers about HSP90 and OPCs (e.g. PMID: 31017387, etc.).

Minor points:

5. Methods (mistakes and lack of explanation):

- Line 534: did the authors use 6-8 weeks old mice, whose body weights were between 24g and 29g? Please clarify.
- OPC isolation and culture: there are many unnecessary sentences. Please check.
- CM transfer: it is unclear what culture medium was used for the CM transfer experiments.
- Antibody array: it is unclear how many "N" were prepared for each group.
- BCAS surgery: ref37 did not use the BCAS model.
- Cognitive tests (novel object recognition test): the index of discrimination was defined as $(N-F)/(N+F)$, but this does not seem a standard method. Please cite references that used this method.
- Line 585: PDGF-AA? PDGF-BB? please clarify.

6. Line240-241: Fig S7a did not show that Cav^{-/-} BMECs induced OPC maturation defects.

7. BBB & BCAS model: BBB damage by BCAS was already reported by Seo et al. in 2013 (PMID: 23281396). The study showed some data about EC-oligodendrocyte interaction, so it would be recommended to cite this paper as well.

8. Cognitive function test: It would be highly recommended to provide data from sham-operated mice to show how much the intervention suppressed the cognitive decline by cerebral hypoperfusion (e.g. almost back to normal?, etc.).

9. Novel object recognition test: previous studies have shown that the hippocampal region would play an important role in the task of this cognitive function test (PMID: 25169255). It may be nice to provide some discussions on this point.

10. Fig 3e: it seems a bit unclear what the authors wanted to show from the Y-axis. Maybe it is better to use the volcano plot style for Fig 3e. Please consider.

Reviewer #3 (Remarks to the Author):

General comments:

This manuscript addresses a very interesting aspect of white matter injury upon cerebral ischemia: the role of the vasculature in regulating the generation and maturation of oligodendrocyte precursor cells upon an ischemic insult. Using a variety of in vivo and in vitro techniques the authors proposed that vascular dysfunction, in this case by the downregulation of Caveolin-1 expression in endothelial cells, impacts on oligodendrogenesis, by preventing oligodendrocyte precursor cells to dissociate from the vasculature and fully differentiate and mature in order to remyelinate.

In general the manuscript is well written, experiments are well planned and the results are interesting. However, certain aspects require further analysis, controls and clarification.

Specific comments:

1. While demyelination is already showed at 2 weeks after BCAS, vascular parameters such as tight junctions, BBB permeability, vessel density etc, are only shown at 4 weeks after BCAS (Figure S2). To support the proposed model where endothelial cell (EC) dysfunction is an early event that leads to impaired remyelination and thus increased demyelination, it is important to show that the vasculature is already affected at 2 weeks or even 1 week after BCAS.

2. Information about the different quantitative image analysis that the authors perform is missing. A detail description of those analysis for all the different immunostainings that were quantitatively analyzed should be provided. For example: how do the authors quantify vessel density in Figure S2? Do they quantify it based on CD31 staining, on Collagen IV, on both? In the images shown in S2d, it seems that in the BCAS condition there are more vessel fragments that are collagen IV+/CD31- compared to control conditions. Have the authors quantified vessel regression (for example by quantify Collagen IV empty sleeves)? As those images suggest, together with the in vitro experiment shown in Fig. 2c, it might be that BCAS leads to vessel regression. Thus, it is important that this quantification is also done and shown.

3. The number of BCAS1+ and ENPP6+ newly-formed oligodendrocytes are shown after 2 and 4 weeks after BCAS. How is the number of mature oligodendrocytes changing in this time frame?

4. Perivascular clusters, and their size, of OPCs increase from 2weeks after BCAS to 4 weeks after BCAS. Is this due to the proliferation of OPCs already present in perivascular clusters? Or to the newly clustering of OPCs into those existing clusters?

5. The authors use CoCl₂ treatment as a stimuli that would “mimic” hypoxia in vitro. However, it is described that CoCl₂ treatment only mimics HIF1a accumulation and not all aspects of hypoxia. In this scenario, it would be good if the authors could show some controls to better characterize the effect of CoCl₂ in BMECs. For example: are the classical hypoxia regulated genes changed upon CoCl₂?

6. It would be nice if the author could give more rationale on why they focused on Cav-1 as a factor that could be deregulated in ECs upon BCAS. Cav1 is a key component of caveolae in ECs. Are caveolae altered in hypoxia conditions?

7. In this study CoCl₂ leads to downregulation of Cav1. In contrast, it is described in several published studies (i.e. <https://doi.org/10.1073/pnas.1112129109>; <https://www.nature.com/articles/ncomms11371>) that Cav1 is upregulated by HIF1a in hypoxic conditions. This seem to be contradictory if one considers that the only effect of CoCl₂ would be “mimicking hypoxia” and would require further controls and discussion.

8. In this respect, the authors hypothesize that the down-regulation of Cav-1 is caused by miR-3074-1-3p, and that it is triggered by hypoxic conditions. It would be important show the presence or level of hypoxia in vivo in the mouse models they are using, with specific markers.

9. The authors use Cav1 $-/-$ in their study and correlate the results to a direct role of Cav1 in ECs. However, in Tabula muris (compendium of single cell transcriptome data from the model organism *Mus musculus*: <https://tabula-muris.ds.czbiohub.org/>), single cell sequencing expression of adult mouse brain shows that Cav1 expression is detected in ECs, pericytes and OPCs. This data indicates that perhaps some of the effects in OPCs and oligodendrogenesis that the authors are seeing in Cav1 $-/-$ mice are due to the lack of Cav1 in other cell populations, namely pericytes and OPCs themselves. Therefore, in order to claim a specific role of EC-Cav1 the authors should target Cav1 specifically in ECs in vivo.

10. To increase Cav-1 expression in ECs the authors use AAV-TIE. What does TIE stand for? Tie1? Tie2? Is there any change in the vasculature in basal conditions just due to the overexpression of Cav-1?

11. Fig 3g, 3i, 4d, 4e and 9e, 9f. As HSP90a is secreted from BVs, one can find the signal from the protein close or associated to BVs but cannot rule out that the protein comes from other cell source. A method as in situ hybridization would contribute to visualize and confirm the signal in BVs from Control, BCAS, AAV-Tie-Cav1 and Cav1 $-/-$ white matter. Particularly as Cav1 $-/-$ is a global knock out which can affect the other cell population also expressing Cav1.

12. Fig 5a-b. HSP90a siRNA significantly reduces HSP90 intensity in CD31+ endothelium. However, this reduction is mild having profound effects in OPC clustering and later differentiation. In the images of Fig. 5a, it is shown that HSP90a is reduced but there is still protein in BVs. However, in the surrounding of the BVs, seems that HSP9a is strongly reduced. Is HSP90 only secreted only by BVs? On the other hand, authors show that the Cy5 signal coming from the AAV mainly infects CD31+ cells, but also looks like there are some off-targets of this AAV in the lower magnification pictures. Could down regulation of HSP90 in off-targets contribute to the dramatic effect observed?

13. In Fig 5. The authors just show the role of HSP90a in Cav-1 $-/-$ mice, but is the downregulation of HSP90a also affecting those parameters in wt mice? Those controls should be shown.

14. Fig 8b,c. Authors show that antagomir restore BMECS function in vitro and promotes differentiation of OPCs in vitro. However the authors do not include control conditions in which the cells have not been exposed to any treatment and in normoxia conditions. To really state that there is a restoration to basal conditions, this control should be included in all experiments.

15. Fig 9g. The authors show that antagomir restore tight junction expression by western blot and structure by EM. However, does this means that functionally BBB permeability is restored in vivo? Moreover, does it restore functionality relieving the tissue from hypoxia, which is the primary cause of OPC blocked differentiation?

16. It would be interesting if they would further discuss the possible role of Cav-1 in other demyelating disorders involving BBB breakdown, whether this mechanism would also be important. Previous publications point Cav-1 to be important in EAE pathogenesis as is involved in immune cell trafficking.

17. In the proposed model (graphical abstract) the authors depict Cav-1 associated to

HSP90a, as if Cav-1 would be binding to HSP90a and preventing its secretion. However, no association of Cav-1 and HSP90a is shown. It is recommended that the authors revised their model and adapt it so that it does not contain overstatements.

Minor comments:

- Fig.S1 d: how was white matter damage quantified?
- Figure S1 n and o: it is recommended to indicate in the y axis the cell type that is being quantified there
- Row 328: living imaging should be live imaging
- The methods in general should be more detailed. For overexpression of Cav-1 by AAV, please include information about the volume injected and the amount of virus particles. How were the microvascular segments used for lysates extracted? Please also describe the layered centrifugation.
- Fig 1. Describe in methods how are the Olig2+ and Pdgfra+ clusters defined for quantification.
- Fig1e. To make sure that Pdgfra OPCs are clustering around the BVs, it would be nice that the images include co-staining with Olig2.
- Fig2.h. Proliferation is showed for isolated OPCs using Ki67 marker, it would be better that in addition to Hoechst as nuclear marker, Olig2 and Pdgfra is included for controlling purity and OPC proliferation.
- Fig 2j-k. In the manuscript only is described that MBP+ OLs were affected by the EC-CM, however in the images were provided co-stainings for MBP with O4 with respective quantifications. Please describe for what exactly was used O4 staining and if it would add information regarding the differentiation status of the cells.
- Fig 2. Could it be shown in vivo that the clustering, proliferation and failure of differentiation of OPC occurs in hypoxic areas? Co-staining with PIMO for example?
- Fig.2 m: Bars for control groups should be presented before CoCl₂ CM as in the group information
- Several fluorescent images with Cav-1 or HSP90a staining are too dim. To clearly show the difference in Cav-1 expression in BCAS mice, images with Cav-1 staining should improved
- Fig.3 g: Staining for Cav-1 in blue is too dim to see. Enhance the signal or switch to another colour. Show also a higher magnification image, in which HSP90a can be seen extra vascular.
- Fig.4 b: Cav-1 staining again too dim
- Fig.4 j: What staining/cells are shown in this graph?

- Fig.4 k: Graph is missing information for groups
- Fig.5 a-b: HSP90a staining is too dim to see. Please change it.

Reviewer #4 (Remarks to the Author):

The manuscript # NCOMMS-21-33332 entitled "Vascular endothelium deploys caveolin-1 to regulate oligodendrogenesis in chronic cerebral ischemia" addresses a critical question regarding the pathogenesis of Leukoaraiosis, a pathological appearance of the brain white matter, which has long been believed to be caused by perfusion disturbances within the arterioles perforating through the deep brain structures and has been associated with risk of stroke, VCID and dementia. Despite extensive pathological documentation of the disease, there are significant unanswered questions regarding the pathogenesis of these very common lesions, which are strongly associated with cognitive dysfunction and dementia. This study represents a step forward in our understanding of the mechanisms of disease. The authors provide a strong evidence that endothelial Cav-1 is required for this process by regulating the release of HSP90a. Moreover, the authors demonstrate that a microRNA (miR 3074-1-3p) functions upstream of Cav-1 in brain endothelial cells (BECs) and treatment with the anti-miR ameliorates deficits in Cav-1. The study provides a nice series of experiments to link miR 3074-1-3p to reduction of Cav-1 and release of HSp90-a as a potential mechanism to drive white matter pathogenesis by reducing maturation of OPCs to oligodendrocytes. However, some of the experimental approaches performed by the authors and the analysis of some of the data requires further clarification to strengthen the rigor of the study.

Major concerns:

- 1) One of the pathological hallmarks of the human disease, is a pathological appearance of the brain white matter, which has long been believed to be caused by perfusion disturbances within the arterioles perforating through the deep brain structures. The in vivo mouse model (BCAS) in Fig S1. shows nicely the progression of the demyelination. However, the analysis of the BBB in Fig S2. needs further details.
 - a) The TEM images for TJs are not convincing. The authors need to provide better images to convince the reviewer that there are TJ abnormalities.
 - b) The authors need to assess BBB dysfunction with smaller MW tracers (cadaverine, dextran 3kDa) that diffuse through disrupted tight junctions.
 - c) What happens to the caveolae by TEM? Is there an increase in the caveolae in capillaries? What happens to the caveolae in arteries and arterioles? This analysis is critical for the paper.
 - d) Collagen IV deposition seems abnormal. The authors need to quantify this aspect since Collagen IV deficits cause BBB leakage and are characteristic of small vessel disease.

2) The authors use the CoCl₂ treatment in BMECs (Fig. 2) to mirror hypoxia. What is the rationale for this approach? Typically, the OGD (oxygen and glucose deprivation) is typically used to mirror the hypoxic environment. I have concerns that CoCl₂ is killing cells and therefore I have serious concerns about the validity of the data and the CM used from these cells. I do not think that this mirrors the disease.

3) The authors show that Cav-1 is reduced in BCAS mouse model. Cav-1 is highly expressed in arteries and arterioles where it regulates neurovascular coupling, but the levels are lower in capillaries. Which vessel subtype identity lose Cav-1 expression in BCAS mouse model and induces release of HSP90a?

4) What happens to blood flow in Cav1^{-/-} BCAS at 2 or 4 weeks? Are there differences from the wild type?

5) Where is Cav-1 expressed in the AAV approach, arterioles or capillaries? What happens to the blood flow in Cav1 overexpression at BCAS at 2 or 4 weeks? The blood flow analysis is critical since if there are changes in blood flow that would change the interpretation of the findings.

6) What happens to white matter abnormalities in wild-type mice when the authors overexpress siRNA for HSP90a? Are the phenotypes rescued in WT mice? These data need to be compared to Cav1^{-/-} data shown in Fig. 5.

7) What happens to the TEER in wt or Cav1^{-/-} BMECs treated with CoCl₂ and antagomir in Fig 8?

8) Several studies have shown that tight junction abnormalities in stroke are still present in Cav-1^{-/-} brains suggesting that tight junction abnormalities occur independently of Cav-1. Although the data provided by the authors with the antagomir treatment would suggest a similar mechanisms at play in BCAS mouse model, they do not consider this model. What happens to tight junctions or Collagen IV in vivo in Cav-1^{-/-} BCAS 2 or 4 weeks with or without antagomer treatment? The authors seem to favor one model but do not consider any alternative models.

Minor:

1) Cav-1 staining in BMECs in Figure 3 look strange. Have the authors validated the antibody in Cav-1^{-/-}?

2) The abstract is too long.

Response to reviewers

Reviewer #1:

Zhao and colleagues use a number of contemporary techniques to confirm the important link between vascular abnormalities and cerebral ischemic demyelination. They identify endothelial caveolin-1 as a mediator oligodendrocyte differentiation via the release of HSP90 α . Further, they identify a microRNA that acts upstream to influence the levels of caveolin-1. Overall, these findings are potentially important given the lack of understanding of ischemic demyelination. The main findings are clear and straight forward, however, some aspects of the manuscript need clarification and further explanation to ensure the validity of the findings.

Major comments

1. The rationale behind many aspects of the manuscript is poorly explained and can be difficult to understand. In the introduction the justification behind the study is unclear e.g. the fact that chronic cerebral ischemia causes demyelination is not stated.

Reply: Thank you for the advice. We have re-written the introduction in the revised manuscript.

2. The precise meaning and quantification of “cluster number” and “cluster size” are not explained within the main text or the materials and methods. The use of CoCl₂ as a means to induce hypoxia is not explained. The use or reasoning of CSD supplementation is not explained. The authors should improve the text to ensure that the manuscript is easy to follow for the wide audience of Nature Communications.

Reply: Thanks for the suggestion. We have added the detailed contents in the revised manuscript highlighted in Red on **Page 32, 6, and 15.**

3. Zhao et al. make the important observation that BCAS leads to an aggregation of OPCs clustered on blood vessels. The authors describe this finding as a “traffic jam” that can be alleviated by knockdown of HSP90 α or an antagomir for miR-3074-1-3p. To provide support to this hypothesis it would be beneficial to assess OPC proliferation around vessels following BCAS and if HSP90 α treatment influences OPC migration.

Reply: This is a good point. As suggested, 1) we firstly assessed OPC proliferation both in the corpus callosum and subventricular zone through co-immunostaining of PDGFR α and Ki67. We found that BCAS mice yielded high numbers of proliferating OPCs in the subventricular zone rather than in the corpus callosum (**Fig. R1a–d**). However, these proliferating OPCs did not accumulate around the vessels. 2) We further detected the migratory capacity of OPCs in the control conditioned medium (CM) or CoCl₂-CM in the Transwell migration assay. OPCs were cultured in the upper chamber for 24 hours. Then the culture medium was changed to control CM or CoCl₂-CM. After co-cultured for 24 hours, the migratory OPCs clinging to the bottom side were fixed and stained with crystal violet. Compared with control CM, CoCl₂-CM caused a significant increase in OPCs migration through filters (**Fig. R1e**). Therefore, these data suggest that the endogenous OPCs mobilization in chronic ischemia included increased proliferation in the subventricular zone and enhanced migration around neighboring vessels. However, increased HSP90 α may mediate OPC attraction to the endothelium, causing accumulation of these migrated OPC on the vessels.

Figure R1. (a, c) Representative fluorescent images of proliferated OPCs (Ki67⁺PDGFR α ⁺) around blood vessels (CD31⁺) in the corpus callosum and subventricular zone [n = 5 in each group; mean \pm S.D.; ** P < 0.01 vs. control; one-way ANOVA, Tukey post hoc test, quantified in (b, d)]. (e, f) Representative images and the quantification of migratory OPCs (purple) clinging to the bottom side of the Transwell (n = 5 experiments in each group; mean \pm S.D.; ** P < 0.01 vs. controls; paired t-test). Scale bar: 20 μm .

4. Further, similar vascular clustering has been overserved during development in mice with overactive *wnt*-signaling through. This is believed to be a result of increased attraction to the vasculature through *Cxcl12-Cxcr4* signaling. Does the mechanism defined within the manuscript overlap with signaling required during development or is this mechanism distinct to adult injury?

Reply: As suggested, we evaluated the expression of CXCL12-CXCR4 signaling during chronic cerebral ischemia. Few CXCL12 was expressed in the microvessels of

the corpus callosum both in control and BCAS mice (**Fig. R2a**), suggesting other possible mechanisms. However, CXCR4-positive cells were frequently detected in the perivascular cluster under hypoxia (**Fig. R2b**). Interestingly, it was reported that CXCR4, as a chaperone of HSP90 α ¹, could be activated by HSP90 α and mediate OPC attraction to the endothelium². Thus, further investigation is needed regarding the effects of secreted HSP90 α on OPC accumulation and the function of CXCR4 during this process.

Figure R2. The expression of vascular CXCL12-CXCR4 signaling after BCAS. Scale bar: 20 μ m.

5. In Figures 4K and 5J the authors quantify the number of mature oligodendrocytes and oligodendrocyte-lineage cells in the corpus callosum. As one of the major myelinated tracts in the brain, the number of oligodendrocyte-lineage cells is expected to be much higher than the ~30% stated in the manuscript. Can the authors explain why the number of oligodendrocyte-lineage cells is low?

Reply: Thanks for this question. The cell statistics were not based on sham mice but on BCAS mice with myelin defects (**Fig. 4l and 5j in the revised manuscript**). In Figure 4, the BCAS surgery was conducted in both groups 3 weeks after the stereotaxic transduction of AAV-control or AAV-Tie1-Cav-1. In Figure 5 of the revised manuscript, wild-type mice and Cav-1^{-/-} mice both received BCAS surgery. Therefore, the reduction of oligodendrocytes number was attributed to BCAS-induced ischemia with or without Cav-1 knockout. To avoid misunderstanding, we added a flow chart

accordingly.

6. The authors find a reduced level of myelination in the Caveolin-1 KO mice. Do these mice also show a reduction in oligodendrocyte numbers? In figure S4 the effect on vessels and myelination is described as “pathological”. Do the authors have evidence to suggest that Caveolin-1 KO mice is pathological?

Reply: According to your advice, we calculated oligodendrocyte numbers in *Cav-1*^{-/-} mice via immunostaining of Olig2 and CC1. The result showed a significant reduction of mature oligodendrocytes in *Cav-1*^{-/-} mice (**Fig. R3, a and b**). In the revised manuscript, we also observed a reduction of CBF (**Fig. R3c**) as well as impaired cognitive function by novel object recognition of 1-h retention compared to wild-type mice (**Fig. xx**), indicating a pathological phenotype of *Cav-1*^{-/-} mice, which was consistent with previous publications^{3, 4}.

Figure R3. The (a, b) mature oligodendrocytes number, (c) CBF value, and (d) NOR performance of *Cav-1*^{-/-} mice. n = 5/group in a, b; n = 10/group in c; mean ± S.D.; ***P* < 0.01 vs. wild type mice, paired t-test. Scale bar: 20 μm.

Minor comments

1. A link between HSP90α released by dysfunctional endothelial cells and oligodendrocyte generation is not a novel finding. Rajani et al 2018 should be referenced in regard to this.

Reply: Thanks for this notice. We have added this reference in the discussion on Page 20.

2. The language throughout should be improved for clarity.

e.g. frequent use of the word “apparently”, line 107 “to the opposite”, line 199 “retarded”.
Line 76 – Yuen et al 2014 Cell (Oligodendrocyte-encoded HIF function couples postnatal myelination and white matter angiogenesis) should also be referenced.

Line 109 – “higher expression of PDGFR α indicated increased mobilisation of OPCs”
– no reference is provided to support this statement.

Line 580 – incorrect reference.

Reply: Thanks for the suggestion. Related contents have been corrected.

3. Immunohistochemistry of ENPP6 *in vivo* and O4 *in vitro* look odd. Both are usually expressed in membranes however the images appear nuclear. High-magnification inserts would be beneficial.

Reply: Thanks for the advice. The ENPP6 staining has been retried and new images have been provided (**Fig. R4**). The O4 has been changed to PDGFR α to provide better information regarding the differentiation status of the cells (**Fig. R5**).

Figure R4. The images of ENPP6 staining all through the manuscript.

Figure R5. The images of co-staining for MBP and PDGFR α all through the manuscript.

4. MBP usually has multiple bands similar to that demonstrated in fig S9 however the rest of the immunoblots throughout the manuscript only show 1 band.

Line 378 – no quantification or evidence is provided to support the statement of reduced oligodendrocyte complexity.

Figure S1 – n and o - it is not stated which graph refers to which quantification.

Reply: Thanks for the advice. The immunoblots for MBP with 1 band were protein samples from primary cultured cells, while MBP with multiple bands were protein samples from mice brains. The pattern of band display was consistent and repeatable throughout our study. Other changes have been made accordingly.

Reviewer #2:

In this study, Zhao et al. examined the mechanisms of cell-cell interaction between endothelium and oligodendroglia under cerebral hypoperfusion. Using in vivo (a mouse model of chronic cerebral hypoperfusion) and in vitro (endothelial cells and OPCs) system, the authors reported that under of chronic ischemic conditions, Cav-1 reduction would increase the vascular secretion of HSP90a, which induces aberrant OPC accumulation in the perivascular region. In addition, the decreased Cav-1 level seemed correlated with the leukoaraiosis severity in patients. The finding about the role of Cav-1/HSP90a signaling in the compensatory oligodendrogenesis after cerebral hypoperfusion is novel, and this study provides a proof-of-concept that this pathway could be a therapeutic target for promoting white matter repair and recovery after white matter damage.

Major points

1. Cav-1/HSP90a interaction: in the authors' summary diagram, Cav-1 binds HSP90a under the normoxic conditions, but after hypoxia, because of degradation/reduction of Cav-1, HSP90a is released from the cellular membrane. But it seems that this point was not proved in this study. Therefore, it would be required to show (i) Cav-1 keeps HSP90a around cell membrane under normoxic conditions and (ii) after hypoxic stress, Cav-1/HSP90a binding would be disrupted in endothelial cells, and (iii) the antagomir would suppress the changes by hypoxia.

Reply: Thanks for the advice. We have performed co-immunostaining and co-immunoprecipitation in the revised manuscript. From Cav-1 and HSP90 α staining, under normoxic status, Cav-1 was co-localized with HSP90 α at the endothelial membrane (**Fig. R6a**). However, during chronic hypoxia, the co-localization of Cav-1 and HSP90 α was destroyed by decreased Cav-1. Consistently, endothelial Cav-1 could be co-immunoprecipitated with HSP90 α (**Fig. R6b**). This interaction was disrupted by hypoxic stress (**Fig. R6b**), which was further enhanced by antagomir ((**Fig. R6c**)).

Figure R6. The co-immunostaining and co-immunoprecipitation of Cav-1 and HSP90 α . White arrows indicated the loss of co-localization of Cav-1 and HSP90 α . Scale bar: 20 μ m.

2. cell source of HSP90 α : although this study focused on the roles of endothelial-derived HSP90 α in oligodendrogenesis after white matter damage, it seems that the increase of HSP90 α level after cerebral hypoperfusion was not only from endothelial cells but also from other types of cells (Fig 3). Therefore, it would be necessary how much endothelial-derived HSP90 α by Cav-1 reduction in endothelial cells would contribute to the aberrant OPC accumulation under chronic hypoxic conditions. In addition, in Fig 4, it may be also needed to provide some western blot data about how much Cav-1 upregulation in endothelial cells would increase the HSP90 α level in the brain (or in the perivascular region of white matter).

Reply: Thanks for the advice. 1) To identify the origin of secreted HSP90 α , we performed combined FISH and immunofluorescence. We used the double-FAM-labeled probes to detect HSP90 α mRNA and antibodies of cell markers to locate different cells, including CD31 for endothelial cells, PDGFR β for pericytes, GFAP for astrocytes, and Olig2 for oligodendroglial lineage cells. The results showed that HSP90 α mRNA was mainly expressed within endothelial cells and significantly increased under chronic ischemic conditions (**Fig. S6**). Thus, the majority of secreted HSP90 α inducing aberrant OPC accumulation with inhibited differentiation was derived from vascular endothelial cells. 2) Unfortunately, we thought the sample preparation for immunoblotting might not be able to distinguish the perivascular region from the vascular region in white matter. Therefore, to answer the relationship between Cav-1 upregulation and HSP90 α downregulation, we separately quantified the endothelial

HSP90 α and secreted HSP90 α based on the immunostaining (**Fig. 4, a-f**). The results displayed that about 1.98-fold upregulation of endothelial Cav-1 led to 46.7% downregulation of endothelial HSP90 α and 48.9% downregulation of secreted HSP90 α in the perivascular region.

Figure S6 from the manuscript. Fluorescent FISH of HSP90 α with different cell markers in the corpus callosum. Scale bar: 20 μ m.

Figure 4 a to f from the manuscript. (a) Experimental flow chart. (b) Representative fluorescent images of the transfection of GFP (green) reporter AAV into microvessels (CD31⁺, red) with enforced Cav-1 (light blue) expression in the CC of BCAS mice. (c) Qualification of

Cav-1 intensity normalized to CD31. (d–f) Double immunostaining and qualifications of CD31 and HSP90 α showing the reduction of HSP90 α induced by AAV-*Tie1-Cav-1*. n = 5 in each group; mean \pm S.D.; **P < 0.001, *P < 0.05 vs. control; paired t-test. Scale bar: 20 μ m.

3. BCAS hypoperfusion model and Leukoaraiosis: in this study, the authors prepared a mouse model of chronic cerebral hypoperfusion by BCAS operation. This model is now relatively well-accepted as a model of SIVD. However, in the clinic, leukoaraiosis could be observed in patients who are not diagnosed with SIVD. This discrepancy may need to be taken into consideration in the manuscript. In addition, the authors included both patients who had an MRI examination with a 3.0T system and a 1.5T system. But the detection sensitivity for white matter hyperintensity in the 3.0T system would be higher than the one in the 1.5T system (PMID: 19369605), and this limitation may need to be noted.

Reply: Thank you for your concern. 1) Sorry for this unclarity. In the present study, we included patients with leukoaraiosis who were mainly of vascular origin, because those, such as multiple sclerosis and leukodystrophy, were already excluded. We have added this information to the exclusion criteria in the revised paper (Page 26). 2) As for the MRI examination with the 3.0T or 1.5T system, we apologized for this typo. In recent years, the 1.5T system in our hospital has been gradually abandoned in neuroimaging. The author who wrote the manuscript was not familiar with the clinical part. We have re-checked the patients' imaging data and confirmed that no patients received the 1.5T MRI examination. Sorry about this mistake again.

4. HSP90 α & OPC: Although the authors discussed how HSP90 α negatively regulates OPC function in this manuscript, it would be recommended to try some mechanistic study into this point. In addition, the authors may need to cite previous papers about HSP90 and OPCs (e.g. PMID: 31017387, etc.).

Reply: It is reported that secreted HSP90 α is capable of binding to low-density lipoprotein receptor-related protein 1 (LRP1)^{5, 6}, which is highly expressed in OPCs and has been defined as a negative regulator of OPC maturation⁷. Therefore, we asked whether the knockdown of LRP1 in OPCs would abolish the effects of HSP90 α on OPC differentiation. We introduced LV-C and LV-*Lrp1*-RNAi to silence LRP1 expression in

OPCs, followed by recombinant HSP90 α (rHSP90 α) co-cultured with OPCs (**Fig. R7a**). Immunostaining and immunoblotting analysis revealed that rHSP90 α inhibited myelin protein expression by OPCs only in the LV-C-transfected group, suggesting that LRP1 may be necessary for the inhibition of rHSP90 α on OPC maturation (**Fig. R7b–e**). We are conducting further studies to address the underlying mechanism between secretory HSP90 α and myelination.

Figure R7. The role of LRP1 in the effects of HSP90 α on OPC differentiation. **(a)** Flow chart. **(b, c)** Representative immunofluorescent images and quantifications showing the successful silencing of LRP1 in OPCs. **(d, e)** Immunoblotting analysis showing the effects of oligodendrocyte LRP1 under rHSP90 α treatment. $n = 4$ experiments; mean \pm S.D.; $^{***}P < 0.001$ vs. LV-C; paired t-test. Scale bar, 20 μ m.

Minor points:

5. Methods (mistakes and lack of explanation):

- Line 534: did the authors use 6-8 weeks old mice, whose body weights were between 24g and 29g? Please clarify.
- OPC isolation and culture: there are many unnecessary sentences. Please check.
- CM transfer: it is unclear what culture medium was used for the CM transfer experiments.

- Antibody array: it is unclear how many "N" were prepared for each group.
- BCAS surgery: ref37 did not use the BCAS model.
- Cognitive tests (novel object recognition test): the index of discrimination was defined as $(N-F)/(N+F)$, but this does not seem a standard method. Please cite references that used this method.
- Line 585: PDGF-AA? PDGF-BB? please clarify.

Reply: Thanks for all suggestions. Related contents have been corrected.

6. Line240-241: Fig S7a did not show that Cav^{-/-} BMECs induced OPC maturation defects.

Reply: Thanks for this notice. Changes have been made accordingly.

7. BBB & BCAS model: BBB damage by BCAS was already reported by Seo et al. in 2013 (PMID: 23281396). The study showed some data about EC-oligodendrocyte interaction, so it would be recommended to cite this paper as well.

Reply: Thanks for the suggestion. The study by Seo has been added in the Introduction.

8. Cognitive function test: It would be highly recommended to provide data from sham-operated mice to show how much the intervention suppressed the cognitive decline by cerebral hypoperfusion (e.g. almost back to normal?, etc.).

Reply: In our study, we have performed NOR in wild-type sham as well as in Cav^{-/-} sham mice. We then included them in the revised manuscript. The results showed that antagomir-induced attenuation of cognitive decline in wild-type BCAS mice could attain a level close to the basal condition (**Fig. S10, f and g**).

Figure S10 f and g from the manuscript. The NOR test results of antagomir-treated BCAS mice of two genotypes. n = 10 in each group; mean \pm S.D.; * $P < 0.05$, ** $P < 0.01$ vs. WT mice; ## $P < 0.01$ vs. WT BCAS + antagomir; one-way ANOVA, Tukey post hoc test.

9. Novel object recognition test: previous studies have shown that the hippocampal region would play an important role in the task of this cognitive function test (PMID: 25169255). It may be nice to provide some discussions on this point.

Reply: Thanks for the advice. We have added some discussions on Page 17.

10. Fig 3e: it seems a bit unclear what the authors wanted to show from the Y-axis. Maybe it is better to use the volcano plot style for Fig 3e. Please consider.

Reply: Thanks for the suggestion. Fig 3e has been changed into a volcano plot style (Fig. 3e).

Figure 3e from the manuscript. Volcano plot showing highly expressed secretory factors by endothelial cells treated with chronic hypoxia. HSP90 α (red spot) is shown as a highly expressed secretory protein compared to control.

Reviewer #3:**General comments:**

This manuscript addresses a very interesting aspect of white matter injury upon cerebral ischemia: the role of the vasculature in regulating the generation and maturation of oligodendrocyte precursor cells upon an ischemic insult. Using a variety of in vivo and in vitro techniques the authors proposed that vascular dysfunction, in this case by the downregulation of Caveolin-1 expression in endothelial cells, impacts on oligodendrogenesis, by preventing oligodendrocyte precursor cells to dissociate from the vasculature and fully differentiate and mature in order to remyelinate.

In general the manuscript is well written, experiments are well planned and the results are interesting. However, certain aspects require further analysis, controls and clarification.

Specific comments:

1. While demyelination is already showed at 2 weeks after BCAS, vascular parameters such as tight junctions, BBB permeability, vessel density etc, are only shown at 4 weeks after BCAS (Figure S2). To support the proposed model where endothelial cell (EC) dysfunction is an early event that leads to impaired remyelination and thus increased demyelination, it is important to show that the vasculature is already affected at 2 weeks or even 1 week after BCAS.

2. Information about the different quantitative image analysis that the authors perform is missing. A detail description of those analysis for all the different immunostainings that were quantitatively analyzed should be provided. For example: how do the authors quantify vessel density in Figure S2? Do they quantify it based on CD31 staining, on Collagen IV, on both? In the images shown in S2d, it seems that in the BCAS condition there are more vessel fragments that are collagen IV⁺/CD31⁻ compared to control conditions. Have the authors quantified vessel regression (for example by quantify Collagen IV empty sleeves)? As those images suggest, together with the in vitro experiment shown in Fig. 2c, it might be that BCAS leads to vessel regression. Thus, it is important that this quantification is also done and shown.

Reply: Thanks for the advice. We have provided a paragraph for quantifications in the

revised manuscript on **Page 32 to 33**. To validate the time course of vasculature damage with demyelination, we probed the structure of tight junction, BBB permeability, and myelin integrity at 1 week, 2 weeks, and 4 weeks after BCAS. According to TEM, the tight junction degenerated and a gap between endothelial cells was generated from 1 week to 4 weeks after BCAS (**Fig. S2, d and e**). Significant BBB breakdown represented by 3-kD dextran leakage from ZO-1 was observed at 1 and 2 weeks after BCAS, which was minimized at 4 weeks post-BCAS (**Fig. S2, f and g**). We evaluated the collagen IV “empty sleeves” comprising a collagen IV basement membrane without endothelial cells in BCAS mice. Briefly, the number of collagen IV empty sleeves increased at 1 week after BCAS with an attenuation thereafter (**Fig. S2, h and i**), suggesting sustained vessel instability during hypoperfusion. As for the pathological changes in myelin integrity, the immunostaining of myelin proteins showed that the white matter lesions could not be found until 2 weeks after BCAS (**Fig. S1**), suggesting that vasculature damage preceded myelin loss in the corpus callosum.

Figure S2 d to j from the manuscript. (d) Representative TEM images showing tight junction gaps between endothelial cells (pink arrows) in sham, BCAS_1w, BCAS_2w, and BCAS_4w mice. The gap area was quantified in (e). (f, g) Representative images and the quantification for ZO-1/dextran co-staining. White arrows indicated the extravasated dextran. (h)

Representative of collagen IV and CD31 staining images. (i, j) Quantifications for the number of basement membrane sleeves devoid of endothelial cells and the number of basement membranes devoid of collagen IV. $n = 5$ in each group; mean \pm S.D.; $**P < 0.01$, $*P < 0.05$ vs. sham; one-way ANOVA, Tukey post hoc test. Scale bar: 500 nm in (d) and 20 μ m in (f, h).

3. The number of BCAS1⁺ and ENPP6⁺ newly-formed oligodendrocytes are shown after 2 and 4 weeks after BCAS. How is the number of mature oligodendrocytes changing in this time frame?

Reply: We stained mature oligodendrocytes with CC1 and Olig2, which were gradually decreased at 2 and 4 weeks after BCAS, though BCAS1⁺ and ENPP6⁺ newly-formed immature oligodendrocytes were increased (Fig. 1, e and h). The data suggest that OPCs tended to differentiate to generate new myelin sheaths. However, this compensation was not durable and failed to restore myelination.

Figure 1 e and h from the manuscript. (e, h) Immunofluorescence images and quantification of the numbers of CC1⁺Olig2⁺ cells in the CC of BCAS mice. $n = 5$ in each group; mean \pm S.D.; $**P < 0.01$, $*P < 0.05$ vs. sham; one-way ANOVA, Tukey post hoc test. Scale bar: 20 μ m.

4. Perivascular clusters, and their size, of OPCs increase from 2 weeks after BCAS to 4 weeks after BCAS. Is this due to the proliferation of OPCs already present in perivascular clusters? Or to the newly clustering of OPCs into those existing clusters?

Reply: This is a good point. 1) We assessed OPC proliferation both in the corpus callosum and subventricular zone through co-immunostaining of PDGFR- α and Ki67. We found that BCAS mice yielded high numbers of proliferating OPCs in the subventricular zone rather than in the corpus callosum (Fig. R10, a–d). However, these proliferating OPCs did not accumulate around the vessels. 2) We further detected the migratory capacity of OPCs within the control conditioned medium (CM) or CoCl₂-CM in the Transwell migration assay. OPCs were cultured in the upper chamber for 24 hours. Then the culture medium was changed to control CM or CoCl₂-CM. After co-

cultured for 24 hours, the migratory OPCs clinging to the bottom side were fixed and stained with crystal violet. Compared with control CM, CoCl₂-CM caused a significant increase in OPCs migration through filters (**Fig. R8, e and f**). 3) Interestingly, the perivascular cluster expressed a high level of CXCR4, whose activation could mediate OPCs attraction to the endothelium (**Fig. R8g**). Therefore, these data suggest that the endogenous OPCs mobilization in chronic ischemia may include increased proliferation in the subventricular zone, enhanced migration around neighboring vessels, and increased attraction to the blood vessels.

Figure R8. (a, c) Representative fluorescent images of proliferated OPCs (Ki67⁺PDGFRα⁺) around blood vessels (CD31⁺) in the corpus callosum and subventricular zone [n = 5 in each group; mean ± S.D.; **P < 0.01 vs. control; one-way ANOVA, Tukey post hoc test, quantified in (b, d)]. (e, f) Representative images and the quantification of migratory OPCs (purple) clinging to the bottom side of the Transwell (n = 5 experiments in each group; mean ± S.D.; **P < 0.01 vs. controls; paired t-test). (g) The expression of CXCR4 in the perivascular cluster after BCAS (n = 5 in each group). Scale bar: 20 μm.

5. The authors use CoCl_2 treatment as a stimuli that would “mimic” hypoxia *in vitro*. However, it is described that CoCl_2 treatment only mimics HIF1a accumulation and not all aspects of hypoxia. In this scenario, it would be good if the authors could show some controls to better characterize the effect of CoCl_2 in BMECs. For example: are the classical hypoxia regulated genes changed upon CoCl_2 ?

Reply: Thanks for this question. CoCl_2 is a fast and efficient hypoxia-mimetic agent, which is commonly used to induce chemical hypoxia in endothelial cells^{8, 9}. It can stabilize HIF complexes more stable than the hypoxic chambers¹⁰ and activate HIF signaling with downstream erythropoietin, which is similar to the true hypoxia¹¹. In the revised manuscript, we have added the OGD model as an additional control to further justify the rationale for CoCl_2 treatment. We performed PCR for the classic hypoxia-related genes in endothelial cells cultured within CoCl_2 or OGD (**Fig. R9a**). OGD, as the optimal *in vitro* model for stroke, induced a remarkable increment in the mRNA level of hypoxic markers, including HIF-1 α , HIF-2 α , PDK-1, BNIP3, and VEGF. CoCl_2 treatment also increased these markers, though milder than OGD, representing significant hypoxia. After exposure to OGD, endothelial tight junctions were also disintegrated (**Fig. R9, b–d**). Importantly, OGD decreased endothelial Cav-1 level and impaired OPC differentiation (**Fig. R9, e–k**), which was similar to chronic CoCl_2 incubation. Thus, CoCl_2 treatment may be an appropriate way to mimic chronic hypoxia in our study.

Figure R9. (a) The mRNA level of classic hypoxia-related genes of control, CoCl₂- and OGD-treated BMECs. (b–d) Representative staining images and immunoblotting of tight junctions and quantifications. (e, f) Representative staining images for CD31 and Cav-1. (g–j) Representative staining images and immunoblotting for OPC maturation by PDGFRα, CNPase, and MBP levels. n = 5 experiments in each group; mean ± S.D.; **P < 0.01, *P < 0.05 vs. control; one-way ANOVA, Tukey post hoc test. Scale bar: 20 μm.

6. It would be nice if the author could give more rationale on why they focused on Cav-1 as a factor that could be deregulated in ECs upon BCAS. Cav1 is a key component of caveolae in ECs. Are caveolae altered in hypoxia conditions?

Reply: 1) A recent paper has indicated that arteriolar endothelial Cav-1, an essential component of caveolae, has an active role in mediating neurovascular coupling, including neural activity and vascular dynamics¹². However, some publications have

pointed out that non-caveolar Cav-1 also exists^{13, 14}. Therefore, we aim to investigate whether endothelial Cav-1 could participate in oligovascular coupling and whether this process was dependent on caveolar Cav-1. Thanks for your suggestion. We have added this rationale in the Introduction (**Page 3**). 2) In the revised manuscript, we used TEM to directly assess the change of caveolae number and found no alteration in capillary and arteriolar caveolae (**Fig. S5, a–c**). Co-immunostaining of vascular markers, such as MFSD2A for capillary and α -SMA for arteriole, with Cavin-1, another caveolae marker, consistently showed no differences in vascular Cavin-1 expression in sham and BCAS mice (**Fig. S5, d–f**). In *in vitro* experiments, the immunostaining demonstrated that chronic hypoxia only suppressed the Cav-1 in the cytoplasm compartment without affecting the co-localization of Cav-1 and Cavin1 in the membrane compartment (**Fig. S5g**), further suggesting that hypoxia did not affect caveolar Cav-1.

Figure S5 from the manuscript. The alterations of capillary and arteriolar caveolae in hypoxia. (a–c) Representative TEM images and statistical analyses of vascular caveolae number (pink arrows; $n = 5$ mice, 4 capillaries or arterioles in each sample). (d–f) Representative Cavin-1 immunostaining with capillary and arteriolar marker, MFSD2A and α -SMA, and their pixel ratios ($n = 5$ in each group). (g) *In vitro* staining of Cav-1 and Cavin-1 in cultured BMECs. The white

arrows indicated the co-localization of these two proteins (n = 5 experiments in each group). mean \pm S.D.; N.S. vs. control; paired t-test. Scale bar: 1 μ m in (a) and 20 μ m in others.

7. In this study CoCl_2 leads to downregulation of Cav1. In contrast, it is described in several published studies (i.e. <https://doi.org/10.1073/pnas.1112129109>; <https://www.nature.com/articles/ncomms11371>) that Cav1 is upregulated by HIF1 α in hypoxic conditions. This seems to be contradictory if one considers that the only effect of CoCl_2 would be “mimicking hypoxia” and would require further controls and discussion.

Reply: Thanks for this question. 1) In the revised manuscript, compared to the OGD model with truly reduced oxygen content, we found chronic CoCl_2 exposure could mimic hypoxia by promoting mRNA expression of several hypoxia-inducible genes *in vitro* (**Fig. R10**). 2) The upregulated Cav-1 by HIF observed in hypoxic tumor cells was total Cav-1, including caveolar and non-caveolar Cav-1. However, in the present study, we found unchanged caveolar Cav-1 in CoCl_2 -treated BMECs (**Fig. S5**), thereby making non-caveolar Cav-1 the significantly reduced Cav-1 in hypoxia. We attributed this discrepancy to two possible reasons. Firstly, it may be due to a different time of hypoxia incubation. We introduced CoCl_2 exposure for 5 consecutive days, while in tumor cells, hypoxic treatment was with a much shorter time within 48 hours. Secondly, the expression pattern of Cav-1 in hypoxia may vary a lot depending on the cell types. The BMECs in our study were primary cultured cells, which are quite different from tumor cell lines. We learned that in adipocytes identically incubated in hypoxia for 48 hours, Cav-1 mRNA level and Cav-1 protein expression were both downregulated by HIF-1^{15, 16}.

Figure R10. The mRNA level of classic hypoxia-related genes of control, CoCl_2 - and OGD-treated BMECs. n = 5 experiments in each group; mean \pm S.D.; * P < 0.05, ** P < 0.01 vs.

controls; one-way ANOVA with Tukey's post hoc test

Figure S5 from the manuscript. The alterations of capillary and arteriolar caveolae in hypoxia. (a–c) Representative TEM images and statistical analyses of vascular caveolae number (pink arrows; n = 5 mice, 4 capillaries or arterioles in each sample). (d–f) Representative Cav-1 immunostaining with capillary and arteriolar marker, MFSD2A and α-SMA, and their pixel ratios (n = 5 in each group). (g) In vitro staining of Cav-1 and Cavin-1 in cultured BMECs. The white arrows indicated the co-localization of these two proteins (n = 5 experiments in each group). mean ± S.D.; N.S. vs. control; paired t-test. Scale bar: 1 μm in (a) and 20 μm in others.

8. In this respect, the authors hypothesize that the down-regulation of Cav-1 is caused by miR-3074-1-3p, and that it is triggered by hypoxic conditions. It would be important show the presence or level of hypoxia in vivo in the mouse models they are using, with specific markers.

Reply: As suggested, we used the pimonidazole (PIMO) metabolite staining kit (HP1-100Kit, Hypoxyprobe, Inc., USA) to mark the hypoxic microenvironment. We co-stained PIMO with Olig/CD31 and BCAS1 *in vivo* (Fig. R11). PIMO-positive regions recognized hypoxia in the CC, which was sustained with the BCAS modeling. The co-immunostaining confirmed that the OPC clustering and inhibited OPC differentiation were observed in hypoxic areas.

Figure R11. The OPC clustering (a) and inhibited OPC differentiation (b) in the hypoxic microenvironment in the CC of BCAS mice. Scale bar: 20 μ m.

9. The authors use *Cav1*^{-/-} in their study and correlate the results to a direct role of *Cav1* in ECs. However, in *Tabula Muris* (compendium of single cell transcriptome data from the model organism *Mus musculus*: <https://tabula-muris.ds.czbiohub.org/>), single cell sequencing expression of adult mouse brain shows that *Cav1* expression is detected in ECs, pericytes and OPCs. This data indicates that perhaps some of the effects in OPCs and oligodendrogenesis that the authors are seeing in *Cav1*^{-/-} mice are due to the lack of *Cav1* in other cell populations, namely pericytes and OPCs themselves. Therefore, in order to claim a specific role of EC-*Cav1* the authors should target *Cav1* specifically in ECs *in vivo*.

Reply: Thanks for this reminder. 1) As you suggested, on the one hand, we supplemented the co-immunostaining for *Cav-1* and PDGFR β (markers for pericytes), *Cav-1* and PDGFR α (markers for OPCs). We evaluated *Cav-1* expression in microvascular endothelial cells, pericytes, oligodendroglial lineage cells (OPCs and mature oligodendrocytes), and astrocytes. The results showed that about 84% of

endothelial cells (CD31⁺) expressed Cav-1, whose level was at least 2.5 times higher than other neural cells (**Fig. S4**). Therefore, as endothelial cells were the most important source of Cav-1, we hypothesized that the effects observed in *Cav-1*^{-/-} mice were mainly due to the knockout of endothelial Cav-1. 2) On the other hand, we used genetic delivery of AAV-*Tie1-Cav-1* to bilateral corpus callosum before BCAS surgery. Tie1, which is one of the Tie family of receptors, is expressed almost exclusively by endothelial cells^{17, 18}. Thus, the AAV with *Tie1* promoter could specifically enhance endothelial Cav-1 expression. We have supplemented some discussions on this point in the revised manuscript (**Page 19**).

Figure S4 from the manuscript. Cav-1 is predominantly expressed in microvascular endothelial cells. (a) Double immunostaining and the quantification of Cav-1 with CD31 (endothelial marker), PDGFRβ (pericyte marker), PDGFRα (OPC marker), CC1 (mature oligodendrocyte marker), and GFAP (astrocyte marker). n = 5 in each group; mean ± S.D.; **P < 0.01 vs. CD31-Cav-1; one-way ANOVA, Tukey post hoc test. Scale bar: 20 μm.

10. To increase Cav-1 expression in ECs the authors use AAV-TIE. What does TIE stand for? Tie1? Tie2? Is there any change in the vasculature in basal conditions just due to the overexpression of Cav-1?

Reply: TIE stands for Tie1. Sorry for this unclarity. We have changed AAV-TIE to AAV-Tie1 all through the revised manuscript. To determine whether Cav-1 overexpression can affect vasculature in normoxia animals, we injected AAV-Tie1-C and AAV-Tie1-Cav-1 into the corpus callosum of wild-type mice respectively. Mice were sacrificed 3 weeks later. We examined the vessel density, vessel length, and tight junctions'

expression in the corpus callosum (**Fig. R12**). AAV-*Tie1-Cav-1* induced a 1.9-fold elevation of Cav-1 in the extracted microvascular segments of the corpus callosum. However, it did not affect the expression of ZO-1, Occludin, and Claudin-5. By co-immunostaining of CD31 and collagen IV, control and Cav-1-overexpression viruses did not change the vasculature system. Collectively, the observation suggests that enhanced expression of endothelial Cav-1 in the corpus callosum of normal mice cannot alter vasculature *in vivo*.

Figure R12. AAV transfection cannot change the vasculature system in normal mice. (**a–d**) Immunoblotting and quantification for the levels of Cav-1 and tight junction proteins. (**e–g**) Representative images for collagen IV and CD31 and quantifications for vessel density and vessel length. $n = 5$ in each group; mean \pm S.D.; ** $P < 0.01$ vs. AAV-control; paired t-test). Scale bar: 20 μ m.

11. Fig 3g, 3i, 4d, 4e and 9e, 9f. As HSP90 α is secreted from BVs, one can find the signal from the protein close to or associated with BVs but cannot rule out that the protein comes from other cell source. A method as *in situ* hybridization would contribute to visualize and confirm the signal in BVs from Control, BCAS, AAV-Tie-Cav1 and Cav1^{-/-} white matter. Particularly as Cav1^{-/-} is a global knock out which can affect the other cell population also expressing Cav1.

Reply: Thanks for this professional advice. To identify the origin of secreted HSP90 α ,

we performed combined FISH and immunofluorescence as suggested. We used the double-FAM-labeled probes to detect HSP90 α mRNA and antibodies of cell markers to locate different cells, including CD31 for endothelial cells, PDGFR β for pericytes, GFAP for astrocytes, and Olig2 for oligodendroglial lineage cells. We compared the HSP90 α mRNA level in wild-type BCAS mice to that in *Cav-1*^{-/-} BCAS mice. The results showed that in the mice of two genotypes HSP90 α mRNA was mainly expressed within endothelial cells (**Fig. R13**). And the signal of HSP90 α mRNA was more frequent in *Cav-1*^{-/-} endothelial cells after BCAS. Thus, the secreted HSP90 α close to or associated with the vessel in chronic hypoxic-ischemic circumstances was mostly derived from vascular endothelial cells. This point has been further discussed (**Page 20**).

Figure R13. Fluorescent FISH of HSP90 α with different cell markers in the corpus callosum. Scale bar: 20 μ m.

12. Fig 5a-b. HSP90a siRNA significantly reduces HSP90 intensity in CD31+ endothelium. However, this reduction is mild but has profound effects on OPC clustering and later differentiation. In the images of Fig. 5a, it is shown that HSP90a is reduced but there is still protein in BVs. However, in the surrounding of the BVs, it seems that HSP9a is strongly reduced. Is HSP90 only secreted only by BVs? On the other hand, authors show that the Cy5 signal coming from the AAV mainly infects CD31+ cells, but also looks like there are some off-targets of this AAV in the lower

magnification pictures. Could down regulation of HSP90 in off-targets contribute to the dramatic effect observed?

13. In Fig 5. The authors just show the role of HSP90a in Cav-1^{-/-} mice, but is the downregulation of HSP90a also affecting those parameters in wt mice? Those controls should be shown.

Reply: This is a good point. We re-arranged the experiments in the revised manuscript regarding this point. We compared the effects of control and HSP90 α siRNA in wild-type BCAS as well as in Cav-1^{-/-} BCAS mice. The function of HSP90 α siRNA observed in Cav-1^{-/-} BCAS mice could be repeatable in wild-type BCAS mice. The N.C. and HSP90 α siRNA packed with PEI were intravenously injected at intervals of three days for a month (**Fig. 5a**). The PEI nanoparticles with repeated injections could enhance siRNA delivery to endothelial cells¹⁹. We separated the statistical quantification into two parts, endothelial HSP90 α level and secreted HSP90 α level, respectively. It turned out that HSP90 α siRNA induced a 24.6% reduction of endothelial HSP90 α , and a 51.9% reduction in secreted HSP90 α in the wild-type mice (**Fig. 5, b and c**). It was also the case for Cav-1^{-/-} mice that the siRNA-induced inhibition on secretory HSP90 α was 1.5-fold higher than endothelial HSP90 α (**Fig. 5, b and c**). It may be due to the probability that the residual intracellular HSP90 α , which was significantly interfered with by HSP90 α siRNA, could only be sufficient to function as a molecular chaperone that regulates substrate proteins rather than be secreted extracellularly. We have verified that it was the secreted HSP90 α that controlled OPC aggregation and maturation failure under chronic cerebral ischemia (**Fig. 5d–j**). Thus, with remarkable suppression of secreted HSP90 α , it was not difficult to recognize the salutary effects of HSP90 α siRNA on OPC clustering and later differentiation.

As inhibition of endothelial HSP90 α remarkably blocked secretory HSP90 α in the surroundings, it was also supported that HSP90 α close to or associated with blood vessels mainly came from endothelial cells, which agreed with the results from combined FISH and immunofluorescence (**Fig. S6 and R13**). The off-target ratio of the PEI-siRNA was only 7.4% in average among the four groups. However, as HSP90 α was mainly derived from endothelial cells, the confounding effects of the siRNA escaping through the endothelial barrier and being endocytosed into other neural cells

could be minimized under the context of aberrant oligovascular coupling.

Figure 5 from the manuscript. HSP90 α siRNA rescues white matter abnormality in wild type and *Cav-1*^{-/-} BCAS mice. (a) Flow chart. (b) Representative images of Cy5-siRNA (red) uptake in CD31⁺ vessels (grey) with HSP90 α (green) in the CC of wild type and *Cav-1*^{-/-} mice. (c) Quantification of endothelial and secretory HSP90 α intensity (normalized to CD31 intensity). (d) Representative staining and reconstruction images of PDGFR α ⁺ OPC clusters attached on the vessel. (e, f) Quantification of frequency and sizes of perivascular clusters. (g–j) Representative images and quantifications of newly-formed pre-myelinating cells (BCAS1⁺ or ENPP6⁺) and mature oligodendrocytes (CC1⁺Olig2⁺). n = 5 in each group; mean \pm S.D.; ***P* < 0.01, **P* < 0.05 vs. N.C.-treated WT BCAS mice; ##*P* < 0.01, #*P* < 0.05 vs. N.C.-treated *Cav-1*^{-/-} BCAS mice; one-way ANOVA, Tukey post hoc test. Scale bar: 20 μ m.

14. *Fig 8b,c. Authors show that antagomir restore BMECS function in vitro and promotes differentiation of OPCs in vitro. However, the authors do not include control conditions in which the cells have not been exposed to any treatment and in normoxia conditions. To really state that there is a restoration to basal conditions, this control should be included in all experiments.*

Reply: Thanks for this advice. We have supplemented wild-type and *Cav-1*^{-/-} controls in normoxia conditions respectively (**Fig. 8**). It displayed that the antagomir could restore the endothelial Cav-1 expression and OPC maturation almost to basal conditions.

Figure 8 from the manuscript. Treatment with antagomir restores BMECs function and OPC differentiation via *Cav-1* *in vitro*. **(a)** BMECs transfected with Cy5-N.C. or Cy5-antagomir (yellow) were double immunostained by *Cav-1* (green) and CD31 (red). **(b,c)** Immunoblotting and quantitative analyses of *Cav-1*, ZO-1, Occludin, and Claudin-5 levels. **(d)** BMEC lysates of control or antagomir treatment were immunoprecipitated with anti-LCN2 or anti-LRP1

antibodies and then analyzed by immunoblotting with anti-HSP90 α . (e) ELISA for detection of HSP90 α in CM from different groups. (f, g) Representative immunofluorescent images and quantifications showing the differentiated cells (MBP⁺) cultured in CM from BMECs of different groups. (h, i) Immunoblotting and quantifications of myelin proteins. n = 5 experiments in each group; mean \pm S.D.; **P* < 0.05, ***P* < 0.01 vs. WT control; ##*P* < 0.01, #*P* < 0.05 vs. WT CoCl₂+antagomir; one-way ANOVA, Tukey post hoc test. Scale bars: 20 μ m.

15. Fig 9g. The authors show that antagomir restore tight junction expression by western blot and structure by EM. However, does this means that functionally BBB permeability is restored in vivo? Moreover, does it restore functionality relieving the tissue from hypoxia, which is the primary cause of OPC blocked differentiation?

Reply: 1) The BBB permeability, represented by dextran staining, was not improved by antagomir no matter in wild-type BCAS mice or *Cav-1*^{-/-} BCAS mice (**Fig. R14**). The observation that BBB leakage was gradually improved after BCAS (**Fig. S2, f and g**) might make the detection of protection from antagomir on BBB difficult. 2) To answer whether antagomir could protect tissue from hypoxia, we evaluated the regional hypoxia-related genes expression in the CC. We found no differences between control-treated and antagomir-treated groups, suggesting that tissue hypoxia was not substantially relieved (**Fig. R15**). Thus, the attenuation of OPC clustering and differentiation may largely be attributed to the signal transduction of the miR/*Cav-1*/HSP90 α pathway.

Figure R14. The dextran extravasation in control- and antagomir-treated BCAS mice of two genotypes. The dotted line outlined the blood vessels. The white arrow indicated dextran leakage. Scale bar: 20 μ m.

Figure S2, f and g from the manuscript. (f, g) Representative images and the quantification for ZO-1/dextran co-staining. White arrows indicated the extravasated dextran. n = 5 in each group; mean ± S.D.; ** $P < 0.01$, * $P < 0.05$ vs. sham; one-way ANOVA, Tukey post hoc test. Scale bar: 20 μm .

Figure R15. The hypoxia-related gene expression in the corpus callosum of control-and antagonist-treated BCAS mice of two genotypes. Scale bar: 20 μm .

16. It would be interesting if they would further discuss the possible role of Cav-1 in other demyelinating disorders involving BBB breakdown, whether this mechanism would also be important. Previous publications point Cav-1 to be important in EAE pathogenesis as is involved in immune cell trafficking.

Reply: We used an alternative model of focal ischemic demyelination by the injection of vasoconstrictor Endothelin-1 to examine whether this OPC perivascular clustering mechanism was involved. In the demyelinated lesions, PDGFR α ⁺ OPCs significantly gathered around blood vessels (**Fig. R16a**). Besides, reduced endothelial Cav-1 and increased endothelial HSP90 α were both observed (**Fig. R16b**). Therefore, endothelial

Cav-1/HSP90 α disruption may be important in ischemic white matter injury. As demyelination in EAE pathology is more probably related to autoimmunity, the role of Cav-1 can be different even quite the opposite. Further investigations and discussion are needed regarding the function of Cav-1 in the EAE model.

Figure R16. (a) Representative images of co-staining of PDGFR α and CD31 in the corpus callosum. (b) Representative images of co-staining of Cav-1, HSP90 α and CD31. Scale bar: 20 μ m.

17. In the proposed model (graphical abstract) the authors depict Cav-1 associated to HSP90 α , as if Cav-1 would be binding to HSP90 α and preventing its secretion. However, no association of Cav-1 and HSP90 α is shown. It is recommended that the authors revised their model and adapt it so that it does not contain overstatements.

Reply: Thanks for the advice. We have performed co-immunostaining and co-immunoprecipitation in the revised manuscript. From Cav-1 and HSP90 α staining, under normoxic status, Cav-1 was co-localized with HSP90 α at the endothelial membrane (**Fig. S3a**). However, during chronic hypoxia, the co-localization of Cav-1 and HSP90 α was destroyed with decreased Cav-1. Consistently, endothelial Cav-1 could be co-immunoprecipitated with HSP90 α (**Fig. S3b**). This interaction was disrupted by hypoxic stress, which was further ameliorated by antagomir (**Fig. S3c**).

However, we removed the graphical abstract in the revised manuscript, because it was not allowed according to the Journal requirements.

Figure S3. The co-immunostaining and co-immunoprecipitation of Cav-1 and HSP90 α . White arrows indicated the loss of co-localization of Cav-1 and HSP90 α . Scale bar: 20 μ m.

Minor comments:

- Fig.S1 d: how was white matter damage quantified?

Reply: The severity of demyelinated lesions was graded by LFB staining as follows: grade 0 (normal), grade 1 (disarrangement of the nerve fibers), grade 2 (marked vacuoles), and grade 3 (the disappearance of myelinated fibers) as described elsewhere²⁰. The related contents were supplemented in the revised paper (**Page 33**).

- Figure S1 n and o: it is recommended to indicate in the y axis the cell type that is being quantified there.

Reply: Thanks for the notice. The information on Y-axis has been modified.

- Row 328: living imaging should be live imaging.

Reply: Thanks for the notice. The related contents have been corrected.

- The methods in general should be more detailed. For overexpression of Cav-1 by AAV, please include information about the volume injected and the amount of virus particles. How were the microvascular segments used for lysates extracted? Please also describe the layered centrifugation.

Reply: Thanks for the advice. Detailed information has been provided accordingly

(Page 29).

- Fig 1. Describe in methods how are the Olig2+ and Pdgfra+ clusters defined for quantification.

Reply: We have provided a paragraph for quantifications in the revised manuscript on **Page 32**.

- Fig1e. To make sure that Pdgfra OPCs are clustering around the BVs, it would be nice that the images include co-staining with Olig2.

Reply: Thanks for the suggestion. We have provided co-staining of PDGFR α , Olig2, and CD31 in **Fig. 1a** in the revised manuscript.

Figure 1, a to c from the manuscript. (a) Representative confocal images of OPC clustering (PDGFR α ⁺Olig2⁺, unfilled arrow) on CD31⁺ blood vessels. (b, c) Quantitative analyses of cluster numbers and sizes, n = 5 in each group; mean \pm S.D.; ** P < 0.01 vs. control; ## P < 0.01, # P < 0.05 vs BCAS_2w; one-way ANOVA, Tukey post hoc test. Scale bar: 20 μ m.

- Fig2.h. Proliferation is showed for isolated OPCs using Ki67 marker, it would be better that in addition to Hoechst as nuclear marker, Olig2 and Pdgfra is included for controlling purity and OPC proliferation.

Reply: Thanks for the advice. The changes have been made accordingly in **Fig. 2g** in the revised manuscript.

Figure 2g from the manuscript. Representative confocal images of Ki67 (proliferating marker), PDGFR α (OPC marker), and Olig2 co-staining. n = 5 experiments in each group. Scale bar: 20 μ m.

- Fig 2j-k. In the manuscript only is described that MBP+ OLs were affected by the EC-CM, however in the images were provided co-staining for MBP with O4 with respective quantifications. Please describe for what exactly was used O4 staining and if it would add information regarding the differentiation status of the cells.

Reply: The marker O4 is an antigen on the surface of pre-oligodendrocytes²¹, which could also be found in mature oligodendrocytes²². Therefore, it could label the oligodendroglial lineage in the process of myelin sheath formation. However, as the staining pattern of marker O4 was concerned by another reviewer, we have changed it to PDGFR α in the revised paper.

Figure R5. The images of co-staining for MBP and PDGFR α all through the manuscript.

- Fig 2. Could it be shown *in vivo* that the clustering, proliferation and failure of differentiation of OPC occurs in hypoxic areas? Co-staining with PIMO for example?

Reply: Thanks for the suggestion. We added co-staining of Olig2/CD31 and BCAS1 with PIMO respectively (**Fig. R11**). Olig2⁺ cells clustering around blood vessels were mainly observed in the PIMO⁺ hypoxic areas. Also, co-staining of BCAS1 with PIMO showed that the hypoxia-induced OPC differentiation was strengthened but increasingly inhibited in the PIMO⁺ hypoxic areas.

Figure R11. The OPC clustering (a) and inhibited OPC differentiation (b) in the hypoxic microenvironment in the CC of BCAS mice. Scale bar: 20 μ m.

- Fig.2 m: Bars for control groups should be presented before CoCl₂ CM as in the group information.

- Several fluorescent images with Cav-1 or HSP90a staining are too dim. To clearly show the difference in Cav-1 expression in BCAS mice, images with Cav-1 staining should be improved.

- Fig.3 g: Staining for Cav-1 in blue is too dim to see. Enhance the signal or switch to another colour. Show also a higher magnification image, in which HSP90a can be seen extra vascular.

- Fig.4 b: Cav-1 staining again too dim.

- Fig.5 a-b: HSP90a staining is too dim to see. Please change it.

Reply: Thanks for the advice. We have changed the related contents accordingly.

- Fig.4 j: What staining/cells are shown in this graph?

- Fig.4 k: Graph is missing information for groups

Reply: The information has been supplemented.

Reviewer #4:

The manuscript # NCOMMS-21-33332 entitled "Vascular endothelium deploys caveolin-1 to regulate oligodendrogenesis in chronic cerebral ischemia" addresses a critical question regarding the pathogenesis of Leukoaraiosis, a pathological appearance of the brain white matter, which has long been believed to be caused by perfusion disturbances within the arterioles perforating through the deep brain structures and has been associated with risk of stroke, VCID and dementia. Despite extensive pathological documentation of the disease, there are significant unanswered questions regarding the pathogenesis of these very common lesions, which are strongly associated with cognitive dysfunction and dementia. This study represents a step forward in our understanding of the mechanisms of disease. The authors provide a strong evidence that endothelial Cav-1 is required for this process by regulating the release of HSP90a. Moreover, the authors demonstrate that a microRNA (miR 3074-1-3p) functions upstream of Cav-1 in brain endothelial cells (BECs) and treatment with the anti-miR ameliorates deficits in Cav-1. The study provides a nice series of experiments to link miR 3074-1-3p to reduction of Cav-1 and release of HSp90-a as a potential mechanism to drive white matter pathogenesis by reducing maturation of OPCs to oligodendrocytes. However, some of the experimental approaches performed by the authors and the analysis of some of the data requires further clarification to strengthen the rigor of the study.

Major concerns:

*1) One of the pathological hallmarks of the human disease, is a pathological appearance of the brain white matter, which has long been believed to be caused by perfusion disturbances within the arterioles perforating through the deep brain structures. The in vivo mouse model (BCAS) in Fig S1. shows nicely the progression of the demyelination. However, the analysis of the BBB in Fig S2. needs further details.
a) The TEM images for TJs are not convincing. The authors need to provide better images to convince the reviewer that there are TJ abnormalities.*

Reply: Thanks for the suggestion. The TEM images have been changed (**Fig. R17**).

Fig. S2d:

Fig. S8e:

Figure R17. The TEM images of endothelial tight junction gap (pink arrows) all through the manuscript. Scale bar: 500 nm.

b) The authors need to assess BBB dysfunction with smaller MW tracers (cadaverine, dextran 3kDa) that diffuse through disrupted tight junctions.

Reply: We probed the structure of tight junction and BBB permeability at 1 week, 2 weeks, and 4 weeks after BCAS to validate the time course of vasculature damage. According to TEM, the tight junction between endothelial cells degenerated from 1 week to 4 weeks after BCAS (**Fig. S2, d and e**). Significant BBB breakdown represented by 3-kD dextran leakage was observed at 1 week after BCAS, which was minimized at 2 and 4 weeks post-BCAS (**Fig. S2, f and g**), suggesting a mechanism independent of tight junction integrity.

Figure S2 d to g from the manuscript. (d) Representative TEM images showing tight junction gaps (pink arrows) in sham, BCAS_1w, BCAS_2w, and BCAS_4w mice. The gap area was quantified in (e). (f, g) Representative images and the quantification for ZO-1/dextran co-

staining. White arrows indicated the extravasated dextran. n = 5 in each group; mean \pm S.D.; ** $P < 0.01$, * $P < 0.05$ vs. sham; one-way ANOVA, Tukey post hoc test. Scale bar: 500 nm in (d) and 20 μ m in (f).

c) *What happens to the caveolae by TEM? Is there an increase in the caveolae in capillaries? What happens to the caveolae in arteries and arterioles? This analysis is critical for the paper.*

Reply: Thanks for this question. We quantified the number of caveolae attached to the luminal and abluminal plasma membranes of endothelial cells by TEM in sham and BCAS_4w mice. We found that there was no significant difference in caveolae number in capillaries as well as in arterioles of the corpus callosum (**Fig. S5, a and b**). We co-immunostained vascular markers, such as MFSD2A for capillary and α -SMA for arteriole, with Cavin-1 to evaluate vascular caveolae expression. Consistently, we found that arteriolar and capillary Cavin-1 expression was not altered after BCAS (**Fig. S5, c–e**). On the one hand, sudden and acute shear stress caused by BCAS may lead to the disassembly of caveolae to buffer membrane tension¹⁴. The disappearance of caveolae may persist because other processes are involved, such as actin dynamics¹⁴. On the other hand, chronic and repetitive shear stress tensions would result in an increment of caveolae number through the mobilization of the Cav-1 pool²³. Therefore, we estimated that the unchanged caveolae at BCAS_4w may represent a temporary balance as the increase in caveolae number in the later phase of BCAS may overcome caveolae flattening at the acute/subacute stage after the surgery. In *in vitro* experiments, the immunostaining demonstrated that chronic hypoxia only suppressed the Cav-1 in the cytoplasm compartment without affecting the co-localization of Cav-1 and Cavin1 in the membrane compartment (**Fig. S5f**), further suggesting that hypoxia did not affect caveolar Cav-1.

Figure S5 from the manuscript. The alterations of capillary and arteriolar caveolae in hypoxia. (a–c) Representative TEM images and statistical analyses of vascular caveolae number (pink arrows; $n = 5$ mice, 4 capillaries or arterioles in each sample). (d–f) Representative Cavin-1 immunostaining with capillary and arteriolar marker, MFSD2A and α -SMA, and their pixel ratios ($n = 5$ in each group). (g) In vitro staining of Cav-1 and Cavin-1 in cultured BMECs. The white arrows indicated the co-localization of these two proteins ($n = 5$ experiments in each group). mean \pm S.D.; N.S. vs. control; paired t-test. Scale bar: 1 μ m in (a) and 20 μ m in others.

d) Collagen IV deposition seems abnormal. The authors need to quantify this aspect since Collagen IV deficits cause BBB leakage and are characteristic of small vessel disease.

Reply: Thanks for the advice. We learned that collagen IV mutations limit perivascular deposition of collagen IV and disrupt collagen IV bioavailability during blood vessel development, which becomes one of the bases of small vessel disease²⁴. In the revised manuscript, the collagen IV staining was supplemented at 1 and 2 weeks after BCAS. We found the collagen IV was degraded from 1 week to 4 weeks post-BCAS (Fig. S2, h–j). It may be explained by the activation of matrix metalloproteinases (MMPs), such as MMP-2 and MMP-9, after BCAS^{25, 26}. MMPs could not only degrade

the basement membrane and TJs but also contribute more extensively to vascular and myelin remodeling²⁷.

Figure S2 h to j from the manuscript. (h) Representative of collagen IV and CD31 staining images. (i, j) Quantifications for the number of basement membrane sleeves devoid of endothelial cells and the number of basement membranes devoid of collagen IV. $n = 5$ in each group; mean \pm S.D.; $**P < 0.01$, $*P < 0.05$ vs. sham; one-way ANOVA, Tukey post hoc test. Scale bar: 20 μm .

2) The authors use the CoCl_2 treatment in BMECs (Fig. 2) to mirror hypoxia. What is the rationale for this approach? Typically, the OGD (oxygen and glucose deprivation) is typically used to mirror the hypoxic environment. I have concerns that CoCl_2 is killing cells and therefore I have serious concerns about the validity of the data and the CM used from these cells. I do not think that this mirrors the disease.

Reply: Thanks for your concern. The CoCl_2 treatment to model chronic hypoxia *in vitro* has been frequently used in papers regarding chronic cerebral hypoperfusion by the team of Lo, Eng H and Arai, K^{25, 28-31}. We treated the BMECs with CoCl_2 for 5 consecutive days with a low concentration of 10 μM . In the revised manuscript, we have added the OGD model (4 hours) as an additional control to further justify the rationale for CoCl_2 as suggested. MTT analysis showed that the viability of cells was equally decreased by CoCl_2 as well as OGD (**Fig. R18a**). We performed PCR for the classic hypoxia-related genes in endothelial cells cultured with CoCl_2 or OGD (**Fig. R18b**). OGD, as the optimal *in vitro* model for stroke, induced a remarkable increment in the mRNA level of hypoxic markers, including HIF-1 α , HIF-2 α , PDK-1, BNIP3, and VEGF. CoCl_2 treatment also increased these markers, though milder than OGD, representing significant hypoxia. Importantly, OGD decreased endothelial Cav-1 level and impaired OPC differentiation, which was similar to chronic CoCl_2 incubation (**Fig. R18c–l**). Thus, CoCl_2 treatment may be an appropriate way to mimic chronic hypoxia

in our study.

Figure R18. (a) MTT analysis of control, CoCl₂- and OGD-treated BMECs. (b) The mRNA level of classic hypoxia-related genes of control, CoCl₂- and OGD-treated BMECs. (c–e) Representative staining images and immunoblotting of tight junctions and quantifications. (f, g) Representative staining images for CD31 and Cav-1. (h–l) Representative staining images and immunoblotting for OPC maturation by PDGFRα, CNPase, and MBP levels. n = 5 experiments in each group; mean ± S.D.; **P < 0.01, *P < 0.05 vs. control; one-way ANOVA, Tukey post hoc test. Scale bar: 20 μm.

3) The authors show that Cav-1 is reduced in the BCAS mouse model. Cav-1 is highly expressed in arteries and arterioles where it regulates neurovascular coupling, but the

levels are lower in capillaries. Which vessel subtype identity lose Cav-1 expression in BCAS mouse model and induces release of HSP90a?

Reply: This is a good point. We used MFSD2A to mark capillaries and α -SMA to mark arterioles. Co-immunostaining of these markers with Cav-1 demonstrated that Cav-1 was reduced by 60.5% in capillaries and by 57.3% in arterioles (**Fig. R19, a and b**; data). The reduction of Cav-1 was not statistically different between capillaries and arterioles. It was also the case for the HSP90 α increment in capillaries and arterioles. HSP90 α in capillaries increased to 3.2 times, while HSP90 α in arterioles was raised to 2.9 times more than in control mice (**Fig. R19, c–e**). Thus, it indicated that though with different levels of caveolae, capillaries and arterioles endured a comparable level of Cav-1 reduction as well as HSP90 α increment.

Figure R19. (a) Representative images of MFSD2A (capillary marker), α -SMA (arteriole marker) and Cav-1 co-staining in wild-type sham and BCAS mice. (b) Quantifications for co-staining pixels of MFSD2A with Cav-1, and α -SMA with Cav-1, respectively. (c, d) Representative images of MFSD2A, α -SMA, and HSP90 α co-staining in wild-type sham and BCAS mice. (e) Quantifications for co-staining pixels of MFSD2A with Cav-1, and α -SMA with Cav-1, respectively. n = 5 in each group; mean \pm S.D.; **P < 0.01 vs. WT sham; one-way

ANOVA, Tukey post hoc test. Scale bar: 20 μ m.

4) *What happens to blood flow in Cav1^{-/-} BCAS at 2 or 4 weeks? Are there differences from the wild type?*

Reply: We assessed the CBF in Cav1^{-/-} BCAS mice at baseline, 2 and 4 weeks. The surgery caused CBF reduction to 49.8% of Cav1^{-/-} sham mice, which was significantly lower than wild-type BCAS mice (**Fig. R20**). However, there was no significant difference in CBF recovery at 4 weeks between wild-type and Cav1^{-/-} BCAS mice.

Figure R20. The time course of CBF changes in wild type and Cav1^{-/-} BCAS mice. n = 5 in each group; mean \pm S.D.; ** $P < 0.01$, * $P < 0.05$ vs WT baseline; # $P < 0.05$ vs WT BCAS_2w; and N.S. vs. WT BCAS_4w; one-way ANOVA, Tukey post hoc test.

5) *Where is Cav-1 expressed in the AAV approach, arterioles or capillaries? What happens to the blood flow in Cav1 overexpression at BCAS at 2 or 4 weeks? The blood flow analysis is critical since if there are changes in blood flow that would change the interpretation of the findings.*

Reply: Thanks for your suggestions. We found the efficiency and specificity of AAV transfection were not statistically different among different blood vessel types, as the GFP signal was both observed in arterioles and capillaries (**Fig. R21, a and b**). Also, no significant differences were found in the CBF value at 2 or 4 weeks after BCAS in the two AAV-transfected groups (**Fig. R21c**). We speculated that overexpression of Cav-1 did not change the caveolae number as reported elsewhere³². To further prove this hypothesis, we conducted cavin-1 staining and TEM in the arterioles to evaluate the change of caveolae number, which could regulate CBF via controlling arteriolar dilatation, after AAV transfection. All the results turned out that the AAV-Tie1-induced Cav-1 overexpression did not affect caveolae number in arterioles (**Fig. R21, d-h**).

Thus, it could also provide explanations for the unaltered CBF in mice receiving AAV-*Tie1-Cav-1*.

Figure R21. (a, b) GFP signals in capillaries (MFSD2A⁺) and arterioles (α -SMA⁺). (c) The time course of CBF value changes in mice transfected with AAV-C or AAV-Cav-1. (d, e) Representative images of GFP, α -SMA, and Cavin-1 co-staining and the quantification. (f-h) TEM images showing the caveolae number in capillaries and arterioles (pink arrows; n = 5 mice, 4 capillaries or arterioles in each sample). n = 5 in each group; mean \pm S.D.; paired t-test. Scale bar: 1 μ m in (f) and 20 μ m in others.

6) What happens to white matter abnormalities in wild-type mice when the authors overexpress siRNA for HSP90 α ? Are the phenotypes rescued in WT mice? These data need to be compared to Cav1^{-/-} data shown in Fig. 5.

Reply: Thanks for this question. We re-arranged the experiments in the revised manuscript regarding this point. We compared the effects of control with HSP90 α

siRNA in wild-type BCAS as well as in *Cav-1*^{-/-} BCAS mice (**Fig. 5**). It showed that the function of HSP90α siRNA observed in *Cav-1*^{-/-} BCAS mice could be repeatable in wild-type BCAS mice.

Figure 5 from the manuscript. HSP90 α siRNA rescues white matter abnormality in wild type and *Cav-1*^{-/-} BCAS mice. (a) Flow chart. (b) Representative images of Cy5-siRNA (red) uptake in CD31⁺ vessels (grey) with HSP90 α (green) in the CC of wild type and *Cav-1*^{-/-} mice. (c) Quantification of endothelial and secretory HSP90 α intensity (normalized to CD31 intensity). (d) Representative staining and reconstruction images of PDGFR α ⁺ OPC clusters attached on the vessel. (e, f) Quantification of frequency and sizes of perivascular clusters. (g–j) Representative images and quantifications of newly-formed pre-myelinating cells (BCAS1⁺ or ENPP6⁺) and mature oligodendrocytes (CC1⁺Olig2⁺). n = 5 in each group; mean \pm S.D.; ***P* < 0.01, **P* < 0.05 vs. N.C.-treated WT BCAS mice; ###*P* < 0.01, #*P* < 0.05 vs. N.C.-treated *Cav-1*^{-/-} BCAS mice; one-way ANOVA, Tukey post hoc test. Scale bar: 20 μ m.

7) What happens to the TEER in wt or *Cav1*^{-/-} BMECs treated with CoCl₂ and antagomir in Fig 8?

Reply: We have conducted the TEER experiments to determine the cell permeability *in vitro* (**Fig. R22**). The TEER value in wild-type cells treated with CoCl₂ and control antagomir was significantly decreased to about 79.4% of the control group. Antagomir could increase TEER value to approximately 1.2-fold compared to the N.C. group, suggesting that antagomir could attenuate endothelial hyper-permeability after hypoxic treatment. However, this protection was not detected in *Cav1*^{-/-} BMECs.

Figure R22. The TEER value of BMECs in different models of two genotypes. n = 5 experiments in each group; mean \pm S.D.; ***P* < 0.01, **P* < 0.05 vs. WT control; ###*P* < 0.01 vs. WT CoCl₂ BMECs + N.C.; one-way ANOVA, Tukey post hoc test.

8) Several studies have shown that tight junction abnormalities in stroke are still present in *Cav-1*^{-/-} brains suggesting that tight junction abnormalities occur independently of *Cav-1*. Although the data provided by the authors with the antagomir treatment would suggest a similar mechanism at play in BCAS mouse model, they do

not consider this model. What happens to tight junctions or Collagen IV *in vivo* in *Cav-1*^{-/-} BCAS 2 or 4 weeks with or without antagomir treatment? The authors seem to favor one model but do not consider any alternative models.

Reply: Thanks for this question. In our study, we found *Cav-1* was necessary for the antagomir-induced attenuation of tight junction abnormalities. We have quantified the area of the gaps between endothelial cells, at which intact tight junctions should have existed, in **Fig. S8**. The results showed that antagomir significantly reduced the area of tight junction gaps in wild-type BCAS mice. However, this protection was not observed in *Cav-1*^{-/-} BCAS mice, suggesting that the function of antagomir was dependent on *Cav-1*.

Figure S8 e and f from the manuscript. Representative TEM images and quantifications for the gaps between endothelial cells (pink arrows) of control- and antagomir-treated BCAS mice of two genotypes. $n = 5$ in each group; mean \pm S.D.; $**P < 0.01$ vs. N.C.-treated WT BCAS mice; $##P < 0.01$ vs. antagomir-treated WT BCAS mice; one-way ANOVA, Tukey post hoc test. Scale bar: 500 nm.

As for an alternative model, we introduced the Endothelin-1-injected focal demyelination model. In the demyelinated lesions, PDGFR α ⁺ OPCs significantly gathered around blood vessels (**Fig. R16a**). Besides, reduced endothelial *Cav-1* and increased endothelial HSP90 α were both observed (**Fig. R16b**). Therefore, endothelial *Cav-1*/HSP90 α disruption may be important in ischemic white matter injury. However, further investigations are needed to evaluate whether the *Cav-1*-dependent function of antagomir is effective in this model.

Figure R16. (a) Representative images of co-staining of PDGFR α and CD31 in the corpus callosum. (b) Representative images of co-staining of Cav-1, HSP90 α , and CD31. Scale bar: 20 μ m.

Minor:

1) Cav-1 staining in BMECs in Figure 3 look strange. Have the authors validated the antibody in Cav-1^{-/-}?

Reply: We retried the Cav-1 staining with two antibodies from Cell Signaling Technology (#3267) and Abcam (ab2910). The staining pattern could be repeatable. As this staining has been widely used in the manuscript, we further validated these two antibodies as suggested (**Fig. R23**).

Figure 3a from the manuscript. Representative images of CD31 and Cav-1 co-staining in control and CoCl₂-treated endothelial cells. Scale bar: 20 μm.

Figure R23. The specificity of Cav-1 antibodies. **(a, b)** Wild type and Cav-1^{-/-} BCAS mice were stained with Cav-1 (green; #3267 from Cell Signaling Technology, USA; ab2910 from Abcam, UK) and CD31 (red) antibodies (n = 5 in each group). **(c, d)** Representative immunostaining and immunoblotting images from cultured primary wild type and Cav-1^{-/-} endothelial cells. n = 5 independent cell cultures. Scale bar, 20 μm.

2) The abstract is too long.

Reply: Thanks for the notice. We have corrected the Abstract according to the Journal Requirements.

Reference

- 1 Mandawat A, Fiskus W, Buckley KM *et al.* Pan-histone deacetylase inhibitor panobinostat depletes CXCR4 levels and signaling and exerts synergistic antimyeloid activity in combination with CXCR4 antagonists. *Blood* 2010; **116**:5306-5315.
- 2 Tsai H-H, Niu J, Munji R *et al.* Oligodendrocyte precursors migrate along vasculature in the developing nervous system. *Science* 2016; **351**:379-384.
- 3 Schubert W, Frank PG, Woodman SE *et al.* Microvascular hyperpermeability in caveolin-1 (-/-) knock-out mice: Treatment with a specific nitric-oxide synthase inhibitor, L-NAME, restores normal microvascular permeability in Cav-1 null mice. *Journal of Biological Chemistry* 2002; **277**:40091-40098.
- 4 Gioiosa L, Raggi C, Ricceri L *et al.* Altered emotionality, spatial memory and cholinergic function in caveolin-1 knock-out mice. *Behavioural brain research* 2008; **188**:255-262.
- 5 Tian Y, Wang C, Chen S, Liu J, Fu Y, Luo Y. Extracellular Hsp90alpha and clusterin synergistically promote breast cancer epithelial-to-mesenchymal transition and metastasis via LRP1. *J Cell Sci* 2019; **132**:jcs228213.
- 6 Woodley DT, Fan J, Cheng CF *et al.* Participation of the lipoprotein receptor LRP1 in hypoxia-HSP90alpha autocrine signaling to promote keratinocyte migration. *J Cell Sci* 2009; **122**:1495-1498.
- 7 Auderset L, Pitman KA, Cullen CL *et al.* Low-Density Lipoprotein Receptor-Related Protein 1 (LRP1) Is a Negative Regulator of Oligodendrocyte Progenitor Cell Differentiation in the Adult Mouse Brain. *Frontiers in Cell and Developmental Biology* 2020; **8**.
- 8 Yang Z, Lin P, Chen B *et al.* Autophagy alleviates hypoxia-induced blood-brain barrier injury via regulation of CLDN5 (claudin 5). *Autophagy* 2021; **17**:3048-3067.
- 9 Yu W, Jin H, Sun W *et al.* Connexin43 promotes angiogenesis through activating the HIF-1 α /VEGF signaling pathway under chronic cerebral hypoperfusion. *Journal of Cerebral Blood Flow & Metabolism* 2021; **41**:2656-2675.
- 10 Lee HR, Leslie F, Azarin SM. A facile in vitro platform to study cancer cell dormancy under hypoxic microenvironments using CoCl₂. *Journal of biological engineering* 2018; **12**:1-15.
- 11 Goldberg MA, Dunning SP, Bunn HF. Regulation of the erythropoietin gene: evidence that the oxygen sensor is a heme protein. *Science* 1988; **242**:1412-1415.
- 12 Chow BW, Nunez V, Kaplan L *et al.* Caveolae in CNS arterioles mediate neurovascular coupling. *Nature* 2020; **579**:106-110.
- 13 Pol A, Morales-Paytuyvi F, Bosch M, Parton RG. Non-caveolar caveolins - duties outside the caves. *J Cell Sci* 2020; **133**.
- 14 Sinha B, Koster D, Ruez R *et al.* Cells respond to mechanical stress by rapid disassembly of caveolae. *Cell* 2011; **144**:402-413.
- 15 Varela-Guruceaga M, Milagro FI, Martinez JA, de Miguel C. Effect of hypoxia on caveolae-related protein expression and insulin signaling in adipocytes. *Mol Cell Endocrinol* 2018; **473**:257-267.
- 16 Van Regemorter E, Joris V, Van Regemorter V *et al.* Downregulation of Caveolin-1 and Upregulation of Deiodinase 3, Associated with Hypoxia-Inducible Factor-1alpha Increase, Are Involved in the Oxidative Stress of Graves' Orbital Adipocytes. *Thyroid* 2021; **31**:627-637.
- 17 Korhonen J, Partanen J, Armstrong E *et al.* Enhanced expression of the tie receptor tyrosine kinase in endothelial cells during neovascularization. *Blood* 1992; **80**:2548-2555.
- 18 Porat RM, Grunewald M, Globerman A *et al.* Specific induction of tie1 promoter by disturbed flow in atherosclerosis-prone vascular niches and flow-obstructing pathologies. *Circulation Research* 2004; **94**:394-401.

- 19 Dahlman JE, Barnes C, Khan O *et al.* In vivo endothelial siRNA delivery using polymeric nanoparticles with low molecular weight. *Nat Nanotechnol* 2014; **9**:648-655.
- 20 Wakita H, Tomimoto H, Akiguchi I, Kimura JJAn. Glial activation and white matter changes in the rat brain induced by chronic cerebral hypoperfusion: an immunohistochemical study. 1994; **87**:484-492.
- 21 Armstrong R, Dorn H, Kufta C, Friedman E, Dubois-Dalcq M. Pre-oligodendrocytes from adult human CNS. *Journal of Neuroscience* 1992; **12**:1538-1547.
- 22 Marton RM, Miura Y, Sloan SA *et al.* Differentiation and maturation of oligodendrocytes in human three-dimensional neural cultures. *Nature neuroscience* 2019; **22**:484-491.
- 23 Yu J, Bergaya S, Murata T *et al.* Direct evidence for the role of caveolin-1 and caveolae in mechanotransduction and remodeling of blood vessels. *J Clin Invest* 2006; **116**:1284-1291.
- 24 Gross SJ, Webb AM, Peterlin AD *et al.* Notch regulates vascular collagen IV basement membrane through modulation of lysyl hydroxylase 3 trafficking. *Angiogenesis* 2021; **24**:789-805.
- 25 Seo JH, Miyamoto N, Hayakawa K *et al.* Oligodendrocyte precursors induce early blood-brain barrier opening after white matter injury. 2013; **123**.
- 26 Ihara M, Tomimoto H, Kinoshita M *et al.* Chronic cerebral hypoperfusion induces MMP-2 but not MMP-9 expression in the microglia and vascular endothelium of white matter. *Journal of Cerebral Blood Flow Metabolism* 2001; **21**:828-834.
- 27 Rajeev V, Fann D, Dinh QN *et al.* Pathophysiology of blood brain barrier dysfunction during chronic cerebral hypoperfusion in vascular cognitive impairment. *Theranostics* 2022; **12**:1639-1658.
- 28 Miyamoto N, Maki T, Pham LD *et al.* Oxidative stress interferes with white matter renewal after prolonged cerebral hypoperfusion in mice. *Stroke* 2013; **44**:3516-3521.
- 29 Miyamoto N, Pham LD, Hayakawa K *et al.* Age-related decline in oligodendrogenesis retards white matter repair in mice. *Stroke* 2013; **44**:2573-2578.
- 30 Miyamoto N, Maki T, Shindo A *et al.* Astrocytes Promote Oligodendrogenesis after White Matter Damage via Brain-Derived Neurotrophic Factor. *Journal of Neuroscience* 2015; **35**:14002-14008.
- 31 Miyamoto N, Magami S, Inaba T *et al.* The effects of A1/A2 astrocytes on oligodendrocyte lineage cells against white matter injury under prolonged cerebral hypoperfusion. *Glia* 2020; **68**:1910-1924.
- 32 Bauer PM, Yu J, Chen Y *et al.* Endothelial-specific expression of caveolin-1 impairs microvascular permeability and angiogenesis. 2005; **102**:204-209.

REVIEWER COMMENTS

Reviewer #1 (Remarks to the Author):

The authors have addressed all concerns raised. The study is now much clearer and the data is greatly supported by the additional work.

Reviewer #2 (Remarks to the Author):

I have no more concerns.

Reviewer #3 (Remarks to the Author):

The authors have satisfactorily answer this reviewer concerns. There are just some minor points that should be considered:

1. The authors now analyze the number of mature oligodendrocytes and of newly-formed immature oligodendrocytes 2 and 4 weeks after BCAS and show that OPCs tend to differentiate to generate new myelin sheaths, but that this is not durable.

Another interpretation of the results is that OPCs do lead to immature OLs, but that the population of mature OLs still decrease either because of a defect in full maturation or in survival of mature OLs. This should be consider in the interpretation of the data.

Reviewer #4 (Remarks to the Author):

The authors have done a remarkable job to address multiple concerns raised by the reviewers in their revised manuscript (# NCOMMS-21-33332A). In particular, they have addressed the concerns that I raised regarding the vascular abnormalities in their BCAS model in both wild-type and Cav1^{-/-} mice. There are some issues that the authors need to address more carefully as follows:

1) Abstract and Introduction - I appreciate that the authors have done a major rewriting of these sections. However, both the abstract and the introduction do not address anymore the relevance to the human disease. We get "introduced" to the human disease on page 12 when the authors measure the serum levels of the hsa-miR-3074-3p in leukoaraiosis patients. This is a critical component of the manuscripts that needs to be presented in the Abstract and Introduction.

2) In response to my concerns, the authors show nicely that the number of caveolae in both arterioles and capillaries does not change between sham and BCAS mice (Fig. S5). The authors conclude that the reduced Cav-1 levels that they observe must be cytoplasmic (pg. 9). As a cell biologist who studies caveolae, I disagree with this conclusion. Cav-1 is a transmembrane protein that is found in the cell membrane (>95%). The rest is found in the

ER or Golgi since it traffics between these compartments in a unique manner (<https://rupress.org/jcb/article/148/1/17/53175/Multiple-Domains-in-Caveolin-1-Control-Its>). Therefore, there is no cytoplasmic Cav-1. I don't understand this statement. Is the trafficking of Cav-1 altered under these BCAS conditions? Fig S5g is oversaturated during image acquisition. If the authors reduce the pixel intensity of both Cav-1 and Cavin-1 staining and add an ER or Golgi marker, they may better understand the trafficking of Cav-1. Since HSP90a has been linked to the ER, this may represent an important point for the authors to consider in their model.

3. In Page 8 of the manuscript, the authors state: "...endothelial Cav-1 could be co-immunoprecipitated with HSP90 α (Fig. S3b). This interaction was disrupted by hypoxic stress. The data do not support this argument. Since the amount of input (Cav-1) is smaller under hypoxia, but the band of HSP90a is stronger that would indicate the opposite. The Co-IPs shown in Fig 8d with the antagomir treatment are equally not convincing. These experiments needs to be repeated and better quantified.

4. In Figure S8, the authors show the effects of antagomir treatment on endothelial tight junctions. The images do not support the data. The last panel (Cav1^{-/-} BCAS with antagomir treatment) shows a perfect tight junction. The pink arrow is not a gap in junctions, it is the lip at the end of tight junctions that is characteristic for endothelial cells. The authors need to find a better representative image or change the quantification in panel f.

5. The CSD experiment (Fig. S11) is a complicated experiment since this peptide can affect multiple scaffolding proteins and may be non-specific. I would advice the authors to consider removing it from the manuscript to improve the clarity and flow of the study.

6. Overall, I found the revised version in particular the text in red harder to read with many typos and incoherent sentences. I would advice the authors to integrate all the information better to make the story more coherent and appealing to the Nature readership.

Response to reviewers

Reviewer #3:

The authors have satisfactorily answered this reviewer' concerns. There are just some minor points that should be considered:

1. The authors now analyze the number of mature oligodendrocytes and of newly-formed immature oligodendrocytes 2 and 4 weeks after BCAS and show that OPCs tend to differentiate to generate new myelin sheaths, but that this is not durable. Another interpretation of the results is that OPCs do lead to immature OLs, but that the population of mature OLs still decrease either because of a defect in full maturation or in the survival of mature OLs. This should be considered in the interpretation of the data.

Reply: Thanks for this professional advice. We have supplemented the interpretation of mature oligodendrocytes and of newly-formed immature oligodendrocytes on **Page 6 and 19** in Red.

Reviewer #4:

The authors have done a remarkable job to address multiple concerns raised by the reviewers in their revised manuscript (# NCOMMS-21-33332A). In particular, they have addressed the concerns that I raised regarding the vascular abnormalities in their BCAS model in both wild-type and *Cav1*^{-/-} mice. There are some issues that the authors need to address more carefully as follows:

1) Abstract and Introduction - I appreciate that the authors have done a major rewriting of these sections. However, both the abstract and the introduction do not address anymore the relevance to the human disease. We get "introduced" to the human disease on page 12 when the authors measure the serum levels of the hsa-miR-3074-3p in leukoaraiosis patients. This is a critical component of the manuscripts that needs to be presented in the Abstract and Introduction.

Reply: Thanks for the suggestion. Related contents highlighted in Red have been added accordingly (**Page 2–4**).

2) In response to my concerns, the authors show nicely that the number of caveolae in both arterioles and capillaries does not change between sham and BCAS mice (Fig. S5). The authors conclude that the reduced Cav-1 levels that they observe must be cytoplasmic (pg. 9). As a cell biologist who studies caveolae, I disagree with this conclusion. Cav-1 is a transmembrane protein that is found in the cell membrane (>95%). The rest is found in the ER or Golgi since it traffics between these compartments in a unique manner (<https://rupress.org/jcb/article/148/1/17/53175/Multiple-Domains-in-Caveolin-1-Control-Its>). Therefore, there is no cytoplasmic Cav-1. I don't understand this statement. Is the trafficking of Cav-1 altered under these BCAS conditions? Fig S5g is oversaturated during image acquisition. If the authors reduce the pixel intensity of both Cav-1 and Cavin-1 staining and add an ER or Golgi marker, they may better understand the trafficking of Cav-1. Since HSP90a has been linked to the ER, this may represent an important point for the authors to consider in their model.

Reply: We really appreciate your professional guidance! And sorry for referring to an

unreliable hypothesis. We deleted the contents related to “cytoplasmic Cav-1” (Page 8–9) and wondered whether it would be better to conclude that the reduced Cav-1 levels observed in our disease model were more likely to be ascribed to non-caveolar Cav-1. In the revised version, we reduced the pixel intensity of both Cav-1 and Cavin-1 staining in Fig. S5g and supplemented the quantifications to show that the reduction of Cav-1 was not due to a loss of co-localization with Cavin-1. As for the Cav-1 trafficking, this is quite a good point which has broaden our views. We are planning a subsequent study based on this point.

Figure S5, g and h. (g, h) *In vitro* staining and quantification of co-localized Cav-1 and Cavin-1 in cultured BMECs. The white arrows indicated the co-localization of the two proteins (n = 5 independent primary cell cultures; mean \pm SD.; N.S. vs. controls; paired t-test). Scale bar, 20 μ m.

3) In Page 8 of the manuscript, the authors state: “...endothelial Cav-1 could be co-immunoprecipitated with HSP90 α (Fig. S3b). This interaction was disrupted by hypoxic stress.” The data do not support this argument. Since the amount of input (Cav-1) is smaller under hypoxia, but the band of HSP90 α is stronger that would indicate the opposite. The Co-IPs shown in Fig 8d with the antagomir treatment are equally not convincing. These experiments need to be repeated and better quantified.

Reply: Thanks for this question. In Fig. S3b, protein extracts from BMECs were incubated with IP antibody for HSP90 α . Therefore, the stronger band of HSP90 α did not reveal the interaction but the increased protein level of HSP90 α . It was the IB band of Cav-1 in the fourth column, which seemed smaller than control group, that could represent the interaction. We further performed a reverse co-immunoprecipitation

analysis in Fig. S3a. We incubated protein extracts with IP antibody for Cav-1. The IB band of HSP90 α in the fourth column, which also seemed smaller than control group, represented the interaction. However, co-IP could not be quantified. Thus, it could only support the conclusion that after hypoxic treatment, reduced Cav-1 was also co-localized with HSP90 α . Related descriptions have been modified (**Page 8–9**). We then supplemented a quantification for co-immunostaining of Cav-1 and HSP90 α . It showed that the percent of co-localized HSP90 α was significantly reduced. Therefore, we showed that as endothelial Cav-1 decreased, the co-localization of Cav-1 and HSP90 α was reduced, which might further cause increased HSP90 α secretion.

Figure S3. The co-immunoprecipitation and co-immunostaining of Cav-1 and HSP90 α . (**a, b**) After immunoprecipitation with anti-Cav-1 or anti-HSP90 α antibodies respectively, the immunoprecipitates were analyzed by immunoblotting with anti-Cav-1 and anti-HSP90 α antibodies ($n = 5$ independent primary cell cultures). (**c, d**) Representative co-immunostaining images and quantification of Cav-1 (green) and HSP90 α (red) in endothelial cells after CoCl₂ treatment. White arrows indicated the co-localization of Cav-1 and HSP90 α ($n = 5$ independent primary cell cultures; mean \pm SD.; ** $P < 0.01$ vs. controls; paired t-test). Scale bar, 20 μ m.

As co-IP could only reflect the presence of interaction but not a change of interaction, we have replaced it with a representative co-immunostaining of Cav-1 and HSP90 α in Fig. 8d. Quantification of co-localization was provided in Fig. 8e. Taken together, it showed that antagonism increased the percent of co-localized HSP90 α in CoCl₂-treated BMECs (**Page 13–14**).

Figure 8, d and e. (d, e) Representative co-immunostaining images and quantification of Cav-1 (green) and HSP90α (red) in control- or antagomir-treated endothelial cells in hypoxic conditions (n = 5 independent primary cell cultures; mean ± SD.; **P < 0.01 vs. WT control+CoCl₂; paired t-test). Scale bar, 20 μm.

4) In Figure S8, the authors show the effects of antagomir treatment on endothelial tight junctions. The images do not support the data. The last panel (Cav1^{-/-} BCAS with antagomir treatment) shows a perfect tight junction. The pink arrow is not a gap in junctions, it is the lip at the end of tight junctions that is characteristic for endothelial cells. The authors need to find a better representative image or change the quantification in panel f.

Reply: Sorry for this mistake. We have changed another representative image. Related quantifications have been changed accordingly.

Figure S8, e and f. (e, f) Representative ultrastructural images and quantifications showing the differences in tight junction gaps between endothelial cells (pink arrows) in CC from mice of 4 groups (n = 5 mice; mean ± SD.; **P < 0.01 vs. WT BCAS + N.C., ##P < 0.01; one-way ANOVA, Tukey post hoc test). Scale bar, 500 nm.

5. The CSD experiment (Fig. S11) is a complicated experiment since this peptide can affect multiple scaffolding proteins and may be non-specific. I would advise the authors to consider removing it from the manuscript to improve the clarity and flow of the study.

Reply: Thanks for the advice. This part has been removed as suggested.

6. Overall, I found the revised version in particular the text in red harder to read with many typos and incoherent sentences. I would advise the authors to integrate all the information better to make the story more coherent and appealing to the Nature readership.

Reply: Thanks for the suggestion. We have done our best in manuscript editing in this revised version. Sorry for those typos. Related contents have been corrected.

REVIEWERS' COMMENTS

Reviewer #3 (Remarks to the Author):

The authors have sufficiently addressed my final concerns.

Reviewer #4 (Remarks to the Author):

The authors have addressed the concerns that I raised with the previous submission adequately. I have no further concerns for the manuscript.

Response to reviewers

Reviewer #3

The authors have sufficiently addressed my final concerns.

Reply: Thanks for your valuable comments on my research. We have learned a lot from your professional advice.

Reviewer #4

The authors have addressed the concerns that I raised with the previous submission adequately. I have no further concerns for the manuscript.

Reply: Thank you for the professional comment. Your professional suggestions have made further improvements of this manuscript available.